# On the Convergence of Decentralized Stochastic Minimax Optimization Algorithm with Compressed Communication

**Yihan Zhang** [1]  **Xinghua Shi** [1]  **Meikang Qiu** [2]  **Yu Wang** [1]  **Hongchang Gao** [1]

## Abstract

The stochastic minimax optimization problem has widespread applications in machine learning. Recently, numerous distributed minimax optimization algorithms have been developed to handle distributed training data. However, most of these algorithms suffer from high communication costs. To address this issue, we develop a novel communication-efficient decentralized stochastic gradient descent ascent with momentum algorithm based on the error feedback mechanism. Importantly, our algorithm demonstrates how to balance the full-precision update and the compression residual with novel designs for coefficients regarding variables and gradients to guarantee convergence. However, compressing the primal and dual variables (and their gradients) of stochastic minimax optimization problems with the error feedback mechanism presents significant challenges for convergence analysis. In particular, it incurs the circle dependence among consensus errors and compression errors. To overcome this challenge, we propose novel strategies that enable the establishment of the convergence rate for our algorithm. Our theoretical results demonstrate how the compression operator influences the convergence rate. Finally, extensive experimental results confirm the efficacy of our proposed algorithm.

## 1. Introduction

Stochastic minimax optimization problems have been gaining significant attention recently, as many machine learning models can be expressed in the form of a minimax optimization problem. Representative examples include generative adversarial networks (Goodfellow et al., 2014a), adversarially robust machine learning models (Goodfellow et al., 2014b; Madry et al., 2018), AUC maximization (area under the curve) (Ying et al., 2016), distributionally robust learning (Gao & Kleywegt, 2023; Duchi et al., 2021), etc. To train those machine learning models on distributed data, in this paper, we aim to develop efficient optimization algorithms to solve the decentralized stochastic minimax optimization problem, which is defined as follows:

$$\min_{x \in \mathbb{R}^{d_x}} \max_{y \in \mathbb{R}^{d_y}} f(x,y) = \frac{1}{K} \sum_{k=1}^{K} f^{(k)}(x,y) , \qquad (1)$$

where $x \in \mathbb{R}^{d_x}$ denotes the primal variable, $y \in \mathbb{R}^{d_y}$ represents the dual variable, $K > 1$ denotes the total number of workers, $f^{(k)}(x,y) = \mathbb{E}[f(x,y;\xi^{(k)})]$ denotes the loss function on the $k$-th worker and $\xi^{(k)}$ denotes the training samples on the $k$-th worker. In addition, all workers conduct peer-to-peer communication.

To solve the decentralized stochastic minimax problem, a series of decentralized stochastic gradient descent ascent algorithms (Xian et al., 2021; Zhang et al., 2021; Gao, 2022; Chen et al., 2024; Zhang et al., 2024a; Xu, 2024; Beznosikov et al., 2022; Huang et al., 2024; Tsaknakis et al., 2020; Huang & Chen, 2023; Metelev et al., 2024) and some federated learning variants (Zhang et al., 2023a; 2024b; Zhang & Gao, 2025b; Zhang et al., 2025) have recently been developed. For example, (Xian et al., 2021) developed a decentralized stochastic variance-reduced gradient descent ascent algorithm based on the STORM (Cutkosky & Orabona, 2019) gradient estimator and established its convergence rate for nonconvex-strongly-concave problems. (Zhang et al., 2021) combined the SPIDER (Fang et al., 2018; Nguyen et al., 2017) gradient estimator with decentralized SGDA and provided theoretical convergence rates for nonconvex-strongly-concave problems. Unfortunately, these decentralized minimax optimization algorithms communicate full-precision variables, which can incur high communication costs when the size of the machine learning model is large, leading to a performance bottleneck.

To address the communication bottleneck issue, a feasi-

---

[1]Department of Computer and Information Sciences, Temple University, Philadelphia, USA [2]Department of Computer & Cyber Sciences, Augusta University, Augusta, USA. Correspondence to: Hongchang Gao <hongchang.gao@temple.edu>.

*Proceedings of the 43$^{rd}$ International Conference on Machine Learning*, Seoul, South Korea. PMLR 306, 2026. Copyright 2026 by the author(s).

ble approach is to compress the communicated variables or gradients with some compression operators, such as sparsification and quantization operators. For example, numerous communication-efficient optimization algorithms (Seide et al., 2014; Stich et al., 2018; Alistarh et al., 2017; Wen et al., 2017; Karimireddy et al., 2019; Richtárik et al., 2021; Stich & Karimireddy, 2020; Gorbunov et al., 2020; Gao et al., 2024; Fatkhullin et al., 2024; Koloskova et al., 2020; Tang et al., 2019; Zhao et al., 2022; Yau & Wai, 2023; Yan et al., 2023) have been developed for traditional minimization problems. Among them, the error feedback strategy (Seide et al., 2014; Stich et al., 2018; Karimireddy et al., 2019), which aims to reduce compression error, has shown great performance in various distributed minimization problems. For example, (Koloskova et al., 2020) applied the error feedback strategy to the gossip-based decentralized stochastic gradient descent (DSGD) algorithm, while (Zhao et al., 2022; Yau & Wai, 2023; Yan et al., 2023) combined it with the gradient-tracking-based DSGD algorithm. All these algorithms have shown improved performance after incorporating the error feedback strategy. However, these algorithms restrict their focus solely to the traditional minimization problem. We are only aware of a recent work (Zhang et al., 2023b) that applied the error feedback strategy to the centralized minimax optimization algorithm to handle the compression error. Therefore, all these existing works cannot be applied directly to Eq. (1) because they cannot address the interaction between the primal and dual variables in the decentralized setting.

In fact, due to the interaction between the primal and dual variables, incorporating error feedback into decentralized minimax optimization algorithms is much more challenging, both from the perspective of algorithmic design and convergence analysis, which is shown as below.

1) **Challenges in Algorithmic Design.** In minimax problems, due to the interaction between the primal and dual variables, the learning rate of these two variables should be carefully set to guarantee convergence (Lin et al., 2020). When incorporating error feedback, the update of these two variables is affected by both the full-precision gradient and the compensated errors. Then, *how to set the learning rate to balance the updates from the full-precision gradient on local workers and the compensated error from other workers to guarantee convergence is still unclear.* Furthermore, when using the gradient tracking communication strategy, the tracked gradient estimator is also communicated and then compensated by the error feedback strategy. Then, *it is unclear how to set the hyperparameter regarding the tracked gradient estimator to guarantee convergence.*

2) **Challenges in Convergence Analysis.** When investigating the theoretical convergence rate, the interdependence between the primal and dual variables makes the consensus error and compensation error of these two variables (and their gradients) depend on each other. In particular, *when the potential function is utilized to establish the convergence rate, the coefficient of different terms in this function could have a **circle dependence***. Then, it is challenging to determine these coefficients to establish the theoretical convergence rate.

To address aforementioned challenges, we developed a novel communication-efficient decentralized stochastic gradient descent ascent with momentum algorithm and established its convergence rate. To the best of our knowledge, this is the first time that the error feedback strategy has been applied to decentralized stochastic minimax optimization with theoretical guarantees. Specifically, on the algorithmic design side, our algorithm uses error feedback to handle the compression error with respect to both variables and gradient trackers. Importantly, *our algorithm demonstrates how to balance the full-precision update and the compression residual with novel designs to guarantee convergence. Meanwhile, our algorithm shows the difference in coordinating these factors when handling variables and gradient trackers.* On the convergence analysis side, we developed a novel potential function to handle various estimation errors to investigate the theoretical convergence rate of our algorithm. In particular, *our proof demonstrates how to determine the coefficient of different estimation errors to handle their **circle dependence** when establishing the theoretical convergence rate.* Furthermore, our convergence analysis discloses the explicit dependence between the number of iterations and the contraction constant in compression operators and the spectral gap of the communication topology. With these novelties in algorithmic design and convergence analysis, our algorithm can achieve superior empirical performance. Specifically, we applied our algorithm to the AUC maximization and fair classification tasks. The experimental results demonstrate that our algorithm can significantly reduce the communication cost while preserving the convergence performance as much as possible. In summary, our paper makes significant contributions to decentralized minimax optimization, which are summarized as follows:

- We developed a novel decentralized stochastic gradient descent ascent with momentum algorithm based on error feedback. Our algorithm demonstrates how to set the hyperparameter to balance the full-precision local updates and the compression residuals regarding the communicated variables and gradients to guarantee convergence. To our knowledge, this is the first time that the error feedback strategy has been applied to decentralized stochastic minimax optimization.

- We established the convergence rate of our algorithm by developing a novel potential function. Our proof demonstrates how to determine the coefficient of different es-

timation errors in the potential function to handle their circle dependence. Moreover, our convergence rates disclose the explicit dependence between the number of iterations and the contraction constant in compression operators and the spectral gap of the communication topology. To the best of our knowledge, this is the first work to provide theoretical guarantees for communication-efficient decentralized minimax optimization algorithms.

• We conducted extensive experiments for the AUC maximization problem and fair classification problem. The results confirm the effectiveness of our algorithm.

# 2. Related Works

## 2.1. Stochasitc Minimax Optimization Algorithms

Numerous machine learning models (Goodfellow et al., 2014a;b; Madry et al., 2018; Gao & Kleywegt, 2023; Duchi et al., 2021; Ying et al., 2016) can be represented in the form of a stochastic minimax optimization problem. For example, (Ying et al., 2016) formulated the AUC maximization problem as the minimax optimization problem, which is widely used in learning classifiers for imbalanced dataset. Given the widespread application of stochastic minimax optimization problems in machine learning, numerous stochastic optimization algorithms have been proposed in recent years. Specifically, a series of stochastic gradient descent ascent (SGDA) algorithms (Lin et al., 2020; Zhang et al., 2020; Qiu et al., 2020; Yang et al., 2020; Huang et al., 2022; Luo et al., 2020; Kovalev et al., 2022; Guo et al., 2023; Yang et al., 2022; Chen et al., 2022; Gao et al., 2021a; Zhang & Gao, 2025a) have been developed and their theoretical convergence rates have been investigated in different settings. For example, (Lin et al., 2020) proposed the mini-batch SGDA algorithm and investigated its convergence rate for nonconvex-strongly-concave and nonconvex-concave problems. However, this method is not suitable for real-world machine learning applications as it theoretically requires a very large batch size. (Yang et al., 2022) established the convergence rate of the alternating SGDA algorithm without requiring strong concavity and a large batch size. (Qiu et al., 2020; Huang et al., 2022; Luo et al., 2020) employed the momentum technique to accelerate the convergence rate of SGDA for nonconvex-strongly-concave problems. However, the aforementioned methods are primarily focused on the single-machine setting. In fact, real-world data are usually very large and distributed on many different devices. Therefore, it has become an urgent necessity to develop distributed optimization algorithms for minimax optimization problems. Among various distributed optimization algorithms, the decentralized method has attracted increasing attention, as each worker only conducts peer-to-peer communication with its neighboring workers, which helps avoid the communication bot-

tleneck in the central server. Thus, our focus is to develop efficient decentralized minimax optimization algorithms.

## 2.2. Decentralized Minimax Optimization Algorithms

To enable decentralized optimization for stochastic minimax optimization problems, numerous decentralized algorithms have been proposed in the past few years. For example, (Xian et al., 2021) proposed a decentralized algorithm for the nonconvex-strongly-concave minimax optimization problem using the variance reduced technique (Cutkosky & Orabona, 2019), which achieves the linear speedup with respect to the number of workers. Similarly, (Zhang et al., 2021) proposed two decentralized stochastic optimization algorithms, which are enhanced by the SPIDER (Fang et al., 2018; Nguyen et al., 2017) gradient estimator, and established the convergence rate for the nonconvex-strongly-concave finite-sum minimax problems. To further improve the convergence rate for finite-sum problems, (Gao, 2022) used a ZeroSARAH variance-reduced gradient estimator (Li & Richtárik, 2021) for the decentralized finite-sum minimax optimization algorithm and achieved a better convergence rate than that in (Zhang et al., 2021), while (Zhang et al., 2024a) established the convergence rate of the stochastic variant of (Zhang et al., 2021). More recently, (Chen et al., 2024) developed a decentralized recursive gradient descent ascent algorithm based on the PAGE (Li et al., 2021) gradient estimator, which can handle the constraint regarding the dual variable and achieve a better dependence on the condition number than existing methods for nonconvex-strongly-concave minimax problems. Unfortunately, all these algorithms focus on scenarios where the communication is full precision. If the model is large, it will lead to significant communication overhead. Therefore, it is still unclear how to design communication-efficient algorithms for the decentralized minimax optimization problem.

## 2.3. Communication-Efficient Decentralized Optimization Algorithms

In distributed optimization problems, each worker usually needs to communicate with a central server or with neighboring workers frequently, which can result in high communication costs. To address this issue, numerous efforts have been made, with gradient compression being one of the most common approaches (Seide et al., 2014; Stich et al., 2018; Jiang & Agrawal, 2018; Alistarh et al., 2017; Wen et al., 2017; Karimireddy et al., 2019; Richtárik et al., 2021; Stich & Karimireddy, 2020; Gorbunov et al., 2020; Gao & Huang, 2020; Gao et al., 2024; Fatkhullin et al., 2024; Koloskova et al., 2020; Gao et al., 2021b; 2023; Tang et al., 2019; Zhao et al., 2022; Yau & Wai, 2023; Yan et al., 2023). However, the standard compression approach can lead to the divergence issue because the com-

pression operation could cause large compression errors. An effective approach to address this issue is the error feedback strategy (Seide et al., 2014). Specifically, it compensates for the update of local variables or gradient estimators with the historical compression residual. As a result, the compressed optimization algorithm can achieve almost the same convergence performance as the full-precision counterpart. For example, (Koloskova et al., 2020) developed CHOCO-SGD, which applied the error feedback mechanism to the gossip-based DSGD algorithm. In CHOCO-SGD, only the variable is communicated and then compensated by error feedback because the gossip communication strategy only communicates variables. As a result, it cannot be applied to the other commonly used communication strategy, gradient tracking, in decentralized optimization because it communicates both variables and gradients. To address this issue, (Zhao et al., 2022; Yau & Wai, 2023; Yan et al., 2023) investigated error feedback for gradient-tracking-based DSGD. However, these existing methods are designed for minimization problems, and it is unknown whether they can be applied to minimax problems and achieve good performance. Additionally, establishing the theoretical convergence rate for minimax problems is also a significant challenge due to the interdependence between the minimization and maximization subproblems.

# 3. Decentralized Minimax Algorithm with Compressed Communication

## 3.1. Problem Setup

In this paper, the compression operator is assumed to satisfy the following condition, which is widely used in existing decentralized minimization methods (Zhao et al., 2022; Yau & Wai, 2023; Yan et al., 2023).

**Assumption 3.1.** The compression operator $\mathcal{C} : \mathbb{R}^d \to \mathbb{R}^d$ satisfies the $\delta$-contraction condition as follows:

$$\|x - \mathcal{C}(x)\|^2 \leq (1 - \delta)\|x\|^2 , \qquad (2)$$

where $\delta \in (0, 1]$.

A representative example that satisfies this condition is Top-K operator (Stich et al., 2018), which communicates the largest $K$ elements in a communicated vector based on their absolute value. More examples can be found in (Karimireddy et al., 2019; Beznosikov et al., 2023; Stich et al., 2018; Alistarh et al., 2017; Koloskova et al., 2020).

In a decentralized optimization system, all workers compose a communication graph. Its adjacency matrix is denoted by $W = [w_{ij}] \in \mathbb{R}_+^{K \times K}$. Specifically, $w_{ij} > 0$ indicates the $i$-th worker connects with the $j$-th worker. Otherwise, they are disconnected. This adjacency matrix satisfies the following assumption, which is commonly used in existing decentralized minimax optimization works (Xian et al., 2021; Zhang et al., 2021; Gao, 2022; Zhang et al., 2024a; Chen et al., 2024).

**Assumption 3.2.** The adjacency matrix $W = [w_{ij}] \in \mathbb{R}_+^{K \times K}$ of the communication graph is doubly stochastic and symmetric. Furthermore, the eigenvalues $\{\lambda_i\}_{i=1}^K$ of $W$ satisfy $|\lambda_K| \leq |\lambda_{K-1}| \leq \cdots \leq |\lambda_2| < |\lambda_1| = 1$.

Based on Assumption 3.2, the spectral gap of $W$ is defined as $1 - \lambda$ where $\lambda = |\lambda_2|$. Additionally, $\mathcal{N}_k = \{j | w_{kj} > 0\}$ denotes the neighbors of the $k$-th worker.

## 3.2. Algorithm

In Algorithm 1, we developed a communication-efficient decentralized stochastic gradient descent ascent with momentum algorithm: CE-DSGDAR. Specifically, our algorithm uses momentum-based gradient estimators to update local variables on each worker and utilize the gradient tracking strategy to conduct communication between workers. The detailed gradient estimators can be found in Step 10 and Step 14 in Algorithm 1. Since our algorithm employs the gradient tracking communication strategy, both variables and gradient estimators are communicated between different workers. Therefore, we apply the error feedback mechanism to both variables and gradient estimators. In the following, we will show how to balance the full-precision local update and the compensation from the error feedback mechanism.

**Error Feedback for Variables.** In the $t$-th iteration, the update regarding the local primal variable $x_t^{(k)}$ and dual variable $y_t^{(k)}$ on the $k$-th worker consists of two steps, respectively. Specifically, *the first step is to update local variables with the full-precision gradient estimators*, which is defined as follows:

$$\tilde{x}_{t+1}^{(k)} = x_t^{(k)} - \gamma_x \eta u_t^{(k)}, \qquad (3)$$

where $\eta > 0$ is the learning rate, $\gamma_x > 0$ is the coefficient attached to $\eta$ for the primal variable, and $u_t^{(k)}$ is the gradient estimator. It is worth noting that this step does not require communication. *The second step is to communicate the primal variable as required by the gradient tracking communication strategy.* To mitigate the large compression error, we use the error feedback mechanism in this step. Specifically, we first use a compression operator $\mathcal{C}(\cdot)$, such as Top-K operator, to do compression: $\mathcal{C}(\tilde{x}_{t+1}^{(k)} - \hat{x}_t^{(k)})$, where $\hat{x}_t^{(k)}$ is the accumulated compression residual, which is defined as follows:

$$\hat{x}_{t+1}^{(j)} = \hat{x}_t^{(j)} + \mathcal{C}(\tilde{x}_{t+1}^{(j)} - \hat{x}_t^{(j)}) , \quad j \in \mathcal{N}_k \cup \{k\}. \quad (4)$$

Note that the $k$-th worker maintains a compression residual $\hat{x}_{t+1}^{(j)}$ on itself for all its neighbors. It should be noted

---

**Algorithm 1** CE-DSGDAR

---

**Input:** $\eta > 0, \gamma_x > 0, \gamma_y > 0, \rho_x > 0, \rho_y > 0, \alpha_x > 0, \alpha_y > 0, \alpha_u > 0, \alpha_v > 0$.
  $\hat{x}_0^{(k)} = x_0^{(k)} = x_0, \hat{y}_0^{(k)} = y_0^{(k)} = y_0$,
  $\hat{u}_0^{(k)} = 0, \hat{v}_0^{(k)} = 0, u_0^{(k)} = g_0^{(k)} = \nabla_x f^{(k)}(x_0, y_0; \xi_0^{(k)}), v_0^{(k)} = h_0^{(k)} = \nabla_y f^{(k)}(x_0, y_0; \xi_0^{(k)})$ .

1: **for** $t = 0, \cdots, T-1$, each worker $k$ **do**
2:   Update the intermediate variable $\tilde{x}_{t+1}^{(k)}$: $\tilde{x}_{t+1}^{(k)} = x_t^{(k)} - \gamma_x \eta u_t^{(k)}$,
3:   Communicate the compressed variable: $\mathcal{C}(\tilde{x}_{t+1}^{(k)} - \hat{x}_t^{(k)})$,
4:   Maintain a copy $\hat{x}_{t+1}^{(j)}$ of its neighbors and itself: $\hat{x}_{t+1}^{(j)} = \hat{x}_t^{(j)} + \mathcal{C}(\tilde{x}_{t+1}^{(j)} - \hat{x}_t^{(j)})$ , $j \in \mathcal{N}_k \cup \{k\}$,
5:   Update the variable $x_{t+1}^{(k)}$: $x_{t+1}^{(k)} = \tilde{x}_{t+1}^{(k)} + \alpha_x \eta \sum_{j \in \mathcal{N}_k} w_{kj}(\hat{x}_{t+1}^{(j)} - \hat{x}_{t+1}^{(k)})$,
6:   Update the intermediate variable $\tilde{y}_{t+1}^{(k)}$: $\tilde{y}_{t+1}^{(k)} = y_t^{(k)} + \gamma_y \eta v_t^{(k)}$,
7:   Communicate the compressed variable: $\mathcal{C}(\tilde{y}_{t+1}^{(k)} - \hat{y}_t^{(k)})$,
8:   Maintain a copy $\hat{y}_{t+1}^{(j)}$ of its neighbors and itself: $\hat{y}_{t+1}^{(j)} = \hat{y}_t^{(j)} + \mathcal{C}(\tilde{y}_{t+1}^{(j)} - \hat{y}_t^{(j)}), j \in \mathcal{N}_k \cup \{k\}$,
9:   Update the variable $y_{t+1}^{(k)}$: $y_{t+1}^{(k)} = \tilde{y}_{t+1}^{(k)} + \alpha_y \eta \sum_{j \in \mathcal{N}_k} w_{kj}(\hat{y}_{t+1}^{(j)} - \hat{y}_{t+1}^{(k)})$,
10:  Update the gradient estimator for $x$:
  $g_{t+1}^{(k)} = (1 - \rho_x \eta^2)(g_t^{(k)} - \nabla_x f^{(k)}(x_t^{(k)}, y_t^{(k)}; \xi_{t+1}^{(k)})) + \nabla_x f^{(k)}(x_{t+1}^{(k)}, y_{t+1}^{(k)}; \xi_{t+1}^{(k)})$,
  $\tilde{u}_{t+1}^{(k)} = u_t^{(k)} + g_{t+1}^{(k)} - g_t^{(k)}$,
11:  Communicate compressed estimator: $\mathcal{C}(\tilde{u}_{t+1}^{(k)} - \hat{u}_t^{(k)})$,
12:  Maintain a copy $\hat{u}_{t+1}^{(j)}$ of its neighbors and itself: $\hat{u}_{t+1}^{(j)} = \hat{u}_t^{(j)} + \mathcal{C}(\tilde{u}_{t+1}^{(j)} - \hat{u}_t^{(j)}), j \in \mathcal{N}_k \cup \{k\}$,
13:  Update gradient estimator: $u_{t+1}^{(k)} = \tilde{u}_{t+1}^{(k)} + \alpha_u \sum_{j \in \mathcal{N}_k} w_{kj}(\hat{u}_{t+1}^{(j)} - \hat{u}_{t+1}^{(k)})$,
14:  Update the gradient estimator for $y$:
  $h_{t+1}^{(k)} = (1 - \rho_y \eta^2)(h_t^{(k)} - \nabla_y f^{(k)}(x_t^{(k)}, y_t^{(k)}; \xi_{t+1}^{(k)})) + \nabla_y f^{(k)}(x_{t+1}^{(k)}, y_{t+1}^{(k)}; \xi_{t+1}^{(k)})$,
  $\tilde{v}_{t+1}^{(k)} = v_t^{(k)} + h_{t+1}^{(k)} - h_t^{(k)}$,
15:  Communicate gradient estimator: $\mathcal{C}(\tilde{v}_{t+1}^{(k)} - \hat{v}_t^{(k)})$
16:  Maintain a copy $\hat{v}_{t+1}^{(j)}$ of its neighbors and itself: $\hat{v}_{t+1}^{(j)} = \hat{v}_t^{(j)} + \mathcal{C}(\tilde{v}_{t+1}^{(j)} - \hat{v}_t^{(j)}), j \in \mathcal{N}_k \cup \{k\}$,
17:  Update gradient estimator: $v_{t+1}^{(k)} = \tilde{v}_{t+1}^{(k)} + \alpha_v \sum_{j \in \mathcal{N}_k} w_{kj}(\hat{v}_{t+1}^{(j)} - \hat{v}_{t+1}^{(k)})$,
18: **end for**

---

that $\mathcal{C}(\tilde{x}_{t+1}^{(k)} - \hat{x}_t^{(k)})$ is communicated between workers so that it can significantly save communication costs. With the compression residual set $\{\hat{x}_{t+1}^{(j)} | j \in \mathcal{N}_k \cup \{k\}\}$ on the $k$-th worker, we compensate for the intermediate full-precision variable $\tilde{x}_{t+1}^{(k)}$ as follows:

$$
x_{t+1}^{(k)} = \underbrace{\tilde{x}_{t+1}^{(k)}}_{\text{full-precision update}}
$$
$$
+ \underbrace{\alpha_x \eta}_{\text{balance coefficient}} \sum_{j \in \mathcal{N}_k} w_{kj} \underbrace{(\hat{x}_{t+1}^{(j)} - \hat{x}_{t+1}^{(k)})}_{\text{compression residual}}, \quad (5)
$$

where $\alpha_x > 0$ is a hyperparameter and $\alpha_x \eta < 1$. It should be noted that **we use the product $\alpha_x \eta$ to balance the full-precision update $\tilde{x}_{t+1}^{(k)}$ and the compression residual $\hat{x}_{t+1}^{(j)} - \hat{x}_{t+1}^{(k)}$. The dependence on the learning rate $\eta$ in the coefficient is a key design of our algorithm**. It should be noted that this is the first time the balance coefficient of the variables has been shown to depend on the learning rate $\eta$. The existing work (Zhao et al., 2022; Yau & Wai, 2023; Yan et al., 2023) on minimization problems does not have

such a design. The update for the dual variable $y_t^{(k)}$ follows the same routine, which can be found in Algorithm 1.

**Error Feedback for Gradients.** In the gradient tracking communication strategy, a gradient tracker is computed and communicated to approximate the global gradient estimator. Therefore, we also use the error feedback mechanism to mitigate the large compression error regarding the gradient tracker, which also includes two steps as the update of variables. Specifically, in the $t$-th iteration, the first step computes the momentum-based gradient estimator, STORM (Cutkosky & Orabona, 2019), for the primal variable on the $k$-th worker as follows:

$$
g_t^{(k)} = (1 - \rho_x \eta^2)(g_{t-1}^{(k)} - \nabla_x f^{(k)}(x_{t-1}^{(k)}, y_{t-1}^{(k)}; \xi_t^{(k)}))
$$
$$
+ \nabla_x f^{(k)}(x_t^{(k)}, y_t^{(k)}; \xi_t^{(k)}) , \quad (6)
$$

where $\rho_x > 0$ is a hyperparameter, and $\rho_x \eta^2 < 1$. Then, the gradient tracking communication mechanism computes a local full-precision gradient tracker as follows:

$$
\tilde{u}_t^{(k)} = u_{t-1}^{(k)} + g_t^{(k)} - g_{t-1}^{(k)} . \quad (7)
$$

The second step then uses the error feedback mechanism to communicate the gradient tracker. Specifically, the $k$-th worker uses the compression operator $\mathcal{C}(\cdot)$ to do compression: $\mathcal{C}(\tilde{u}_{t+1}^{(k)} - \hat{u}_t^{(k)})$, which is to compress the difference between the full-precision intermediate gradient tracker $\tilde{u}_{t+1}^{(k)}$ and the compression residual $\hat{u}_t^{(k)}$. In particular, the residual set $\{\hat{u}_t^{(j)}|j \in \mathcal{N}_k \cup \{k\}\}$ on the $k$-th worker is updated as:

$$\hat{u}_{t+1}^{(j)} = \hat{u}_t^{(j)} + \mathcal{C}(\tilde{u}_{t+1}^{(j)} - \hat{u}_t^{(j)}), \quad j \in \mathcal{N}_k \cup \{k\}. \quad (8)$$

It should be noted that $\mathcal{C}(\tilde{u}_{t+1}^{(j)} - \hat{u}_t^{(j)})$ is communicated across workers so that it can save communication costs. Then, our algorithm compensates the full-precision intermediate gradient tracker $\tilde{u}_{t+1}^{(k)}$ with the residual set $\{\hat{u}_t^{(j)}|j \in \mathcal{N}_k \cup \{k\}\}$ as follows:

$$u_{t+1}^{(k)} = \underbrace{\tilde{u}_{t+1}^{(k)}}_{\text{full-precision update}}$$
$$+ \underbrace{\alpha_u}_{\text{balance coefficient}} \sum_{j \in \mathcal{N}_k} w_{kj} \underbrace{(\hat{u}_{t+1}^{(j)} - \hat{u}_{t+1}^{(k)})}_{\text{compression residual}}, \quad (9)$$

where $0 < \alpha_u < 1$ is a hyperparameter to balance the local full-precision gradient tracker $\tilde{u}_{t+1}^{(k)}$ and the compensation residual $\hat{u}_{t+1}^{(j)} - \hat{u}_{t+1}^{(k)}$. It is worth noting that **the balance coefficient for the gradient tracker does NOT depend on the learning rate, which is different from the update of the variable**. The update for the gradient tracker $v_t^{(k)}$ with respect to the dual variable follows the same routine, which can be found in Algorithm 1.

**Discussion.** To better illustrate the motivation behind our balance coefficients, we provide more intuitive explanations. Specifically, we can use the full-precision method to illustrate the underlying reason. In full-precision methods such as DM-HSGD, the update of variables is adjusted by the learning rate (e.g., Steps 14 and 15 in Algorithm 1 of DM-HSGD), whereas the update of the gradient tracker does not involve the learning rate (e.g., Steps 11 and 12 in Algorithm 1 of DM-HSGD). Following this mechanism, the update in Eq. (5) operates on the variable $x$; therefore, we should incorporate the learning rate into the balance coefficient, because the compression residual is a key component in updating $x$. Meanwhile, the update in Eq. (9) operates on the gradient tracker; therefore, we do not incorporate the learning rate into the balance coefficient, in line with the full-precision algorithm. In summary, in our proposed algorithm, we developed a novel strategy to balance the full precision update and the compression residual. Our algorithm shows that the balance coefficient for the variable depends on the learning rate $\eta$, while that for the gradient tracker is independent of $\eta$. This is the first time discover this strategy for minimax optimization, which is different

from existing decentralized minimization algorithms (Zhao et al., 2022; Yau & Wai, 2023; Yan et al., 2023).

## 4. Convergence Analysis

### 4.1. Assumption

To establish the convergence rate of Algorithm 1, throughout this paper, we make the following assumptions about the nature of the loss function. These assumptions are very common in existing works on the minimax optimization problem (Lin et al., 2020; Luo et al., 2020; Huang et al., 2022; Qiu et al., 2020).

**Assumption 4.1.** For $k \in \{1, \cdots, K\}$, the loss function $f^{(k)}(\cdot, \cdot)$ satisfies the mean-square $L$-Lipschitz smoothness: $\mathbb{E}[\|\nabla_x f^{(k)}(x_1, y_1; \xi) - \nabla_x f^{(k)}(x_2, y_2; \xi)\|^2] \leq L^2\|x_1 - x_2\|^2 + L^2\|y_1 - y_2\|^2$ and $\mathbb{E}[\|\nabla_y f^{(k)}(x_1, y_1; \xi) - \nabla_y f^{(k)}(x_2, y_2; \xi)\|^2] \leq L^2\|x_1 - x_2\|^2 + L^2\|y_1 - y_2\|^2$, where $L > 0$ is a constant, $(x_1, y_1) \in \mathbb{R}^{d_x} \times \mathbb{R}^{d_y}$ and $(x_2, y_2) \in \mathbb{R}^{d_x} \times \mathbb{R}^{d_y}$.

**Assumption 4.2.** For $k \in \{1, \cdots, K\}$, the variances of $\nabla_x f^{(k)}(\cdot, \cdot; \xi)$ and $\nabla_y f^{(k)}(\cdot, \cdot; \xi)$ are upper bounded by $\sigma^2$, where $\sigma > 0$ is a constant.

**Assumption 4.3.** For $k \in \{1, \cdots, K\}$, $f^{(k)}(x, y)$ is $\mu$ strongly concave with respect to $y$ for any fixed $x$.

Based on these assumptions, we denote the condition number by $\kappa = L/\mu$. Additionally, we introduce the following function:

$$F(x) = f(x, y^*(x)), \quad (10)$$

where $y^*(x) = \arg\max_y f(x, y)$. According to (Lin et al., 2020), $F(x)$ is $L_F$-smooth, where $L_F = 2\kappa L$.

### 4.2. Convergence Rate

We establish the convergence rate of our algorithm CE-SGDAR in Theorem 4.4, whose proof is in Appendix D.

**Theorem 4.4.** *Given Assumptions 3.1-4.3, by setting $\rho_x = O\left(\frac{1}{K}\right)$, $\rho_y = O\left(\frac{1}{K}\right)$, $\alpha_x = O(\delta^2(1-\lambda))$, $\alpha_y = O(\delta^2(1-\lambda))$, $\alpha_u = O(\delta(1-\lambda))$, $\alpha_v = O(\delta(1-\lambda))$, $\gamma_x = O\left(\frac{\delta^3(1-\lambda)^4}{\kappa^3}\right)$, $\gamma_y = O\left(\frac{\delta^3(1-\lambda)^4}{\kappa}\right)$, $\eta = O\left(\frac{K\epsilon}{\kappa}\right)$, $B = O\left(\frac{1}{\epsilon}\right)$ (the batch size in the zeroth iteration), and $T = O\left(\frac{\kappa^4}{K\delta^3(1-\lambda)^4\epsilon^3}\right)$ under the worst case, CE-SGDAR in Algorithm 1 can achieve the $\epsilon$-accuracy solution: $\frac{1}{T}\sum_{t=0}^{T-1}\mathbb{E}[\|\nabla F(\bar{x}_t)\|^2] \leq O(\epsilon^2)$, where $\epsilon > 0$.*

*Remark* 4.5. Theorem 4.4 indicates that the iteration complexity of CE-SGDAR is $T = O\left(\frac{\kappa^4}{K\delta^3(1-\lambda)^4\epsilon^3}\right)$. When using only one worker and the full-precision update, i.e., $K = 1$, $1 - \lambda = 1$, and $\delta = 1$, the iteration complexity becomes $T = O\left(\frac{\kappa^4}{\epsilon^3}\right)$, which is even better than the single-machine counterpart $T = O\left(\frac{\kappa^{4.5}}{\epsilon^3}\right)$ in (Qiu et al.,

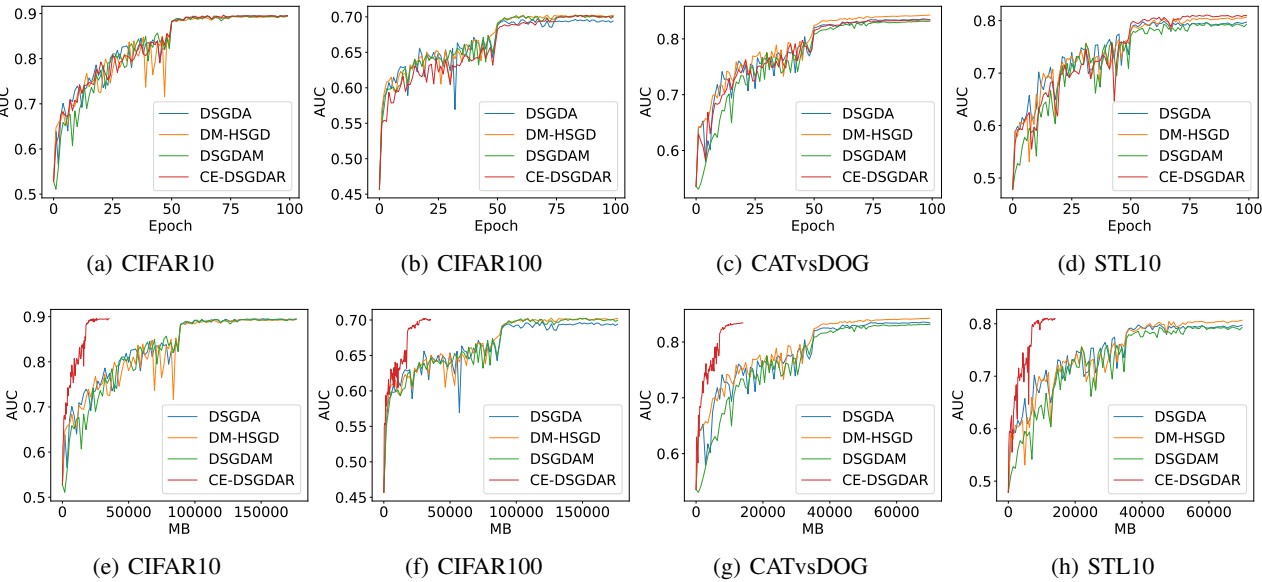

*Figure 1.* The test AUC score versus the number of epochs (1st row) and the communicated megabytes (2nd row).

2020) (See its Theorem 5.3) and (Huang et al., 2022).

*Remark* 4.6. Compared to the full-precision counterpart in the decentralized setting in (Xian et al., 2021), whose iteration complexity depends on the spectral gap in the order of $O\left(\frac{1}{(1-\lambda)^2}\right)$, our algorithms have a high-order dependence on the spectral gap due to the compression operation. The reason is that the balance coefficient, $\alpha_x$, $\alpha_y$, $\alpha_u$, and $\alpha_v$, depends on $1 - \lambda$ as shown in Eq. (109), which affects $\gamma_x$ and $\gamma_y$ as shown in Eq. (110) and therefore the final convergence rate. On the contrary, the full-precision counterpart does not have the balance coefficient, achieving a better dependence. All in all, the high-order dependence is due to the compression operation.

### 4.3. Proof Sketch

The proof sketch is deferred to Appendix C, where the key idea, the unique challenge, and the novel solution are discussed in detail. Generally speaking, the key challenge lies in the circular dependence between the consensus error and the compression error. As a result, when determining their coefficients in the potential function, there exists a circular dependence among these coefficients. This makes it more challenging to choose the coefficients of the potential function compared to full-precision decentralized minimax optimization, where no compression error is present.

## 5. Experiments

In our experiments, we verify the performance of our algorithm on two applications: AUC maximization and fair classification.

Regarding the AUC maximization problem, (Ying et al., 2016) shows that it can be formulated as a minimax optimization problem, which is defined in Eq. (11) as follows:

$$
\min_{x=[\theta,w_1,w_2]} \max_y f(x, y; a, b)
$$
$$
\triangleq (1-p)(g(\theta; a) - w_1)^2 \mathbb{I}_{[b=1]}
$$
$$
+ 2(1 + w_3)(pg(\theta; a)\mathbb{I}_{[b=-1]} - (1-p)g(w; a)\mathbb{I}_{[b=1]})
$$
$$
+ p(g(\theta; a) - w_2)^2 \mathbb{I}_{[b=-1]} - p(1-p)w_3^2 , \qquad (11)
$$

where $\theta$ is the classifier's weight, $w_1$, $w_2$, and $y$ are the parameters for computing the AUC loss, $(a, b)$ denotes the sample's feature and label respectively, $g(\theta; a)$ represents the prediction of the classifier $g(\cdot)$ for the sample $a$, and $p$ is the ratio of positive samples. When the classifier is a deep neural network, this loss function is a nonconvex-strongly-concave minimax problem.

### 5.1. Experimental Setup

**Datasets.** In the experiment, we use four image datasets: CATvsDOG [1], CIFAR10, CIFAR100 (Krizhevsky et al., 2009), and STL10 (Coates et al., 2011). Since AUC maximization is designed for the imbalanced data classification task, we constructed imbalanced datasets from these benchmark datasets. Specifically, we treat the first half of the classes in each dataset as the positive class and the second half as the negative class, thus transforming them into binary classification datasets. Subsequently, a controlled imbalance ratio was introduced by randomly selecting a subset from the positive class and combining it with the

---

[1] https://www.kaggle.com/c/dogs-vs-cats

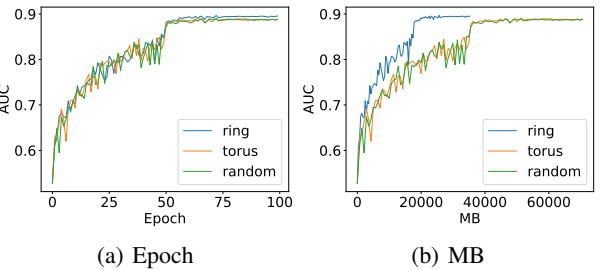

(a) Epoch      (b) MB

*Figure 2.* The test AUC score versus the number of epochs and communication cost for CIFAR10 dataset using different communication topologies.

samples from the negative class, resulting in an imbalanced dataset where the positive class samples constitute $10\%$ of the total samples. Following this, the training set of this dataset is randomly distributed among all workers, serving as their local training sets. The complete and identical testing set is distributed to each worker.

**Experimental settings.** To demonstrate the performance of our algorithms, we compare them with three full-precision baseline algorithms: decentralized stochastic gradient descent ascent algorithm (DSGDA) (Tsaknakis et al., 2020), decentralized minimax hybrid stochastic gradient descent (DM-HSGD) (Xian et al., 2021), and decentralized stochastic gradient descent ascent with momentum algorithm (DSGDAM). Note that we replace the STORM gradient estimator in DM-HSGD with the moving-average gradient estimator to get DSGDAM. In addition, we do not compare our algorithm with methods that require computing the full gradient (Chen et al., 2024; Zhang et al., 2024a), as they are impractical for large-scale datasets. The learning rate for all baseline methods is set to 0.1. The momentum coefficient of DM-HSGD is set to 0.9, while that of DSGDAM is set to 0.1, since these values have the best performance. For our algorithm, we set the learning rate $\eta$ to 0.1, $\gamma_x$ and $\gamma_y$ to 0.9. Furthermore, we set $\rho_x = \rho_y$ so that $\rho_x \eta^2 = 0.9$ for CE-DSGDAR, which is the same as DM-HSGD. For $\alpha_x$ and $\alpha_y$, we set $\alpha_x = \alpha_y$ and select its value from $\{0.1, 0.3, 0.5, 0.7, 0.9\}$. For $\alpha_u$ and $\alpha_v$, we set $\alpha_u = \alpha_v$ and select its value from $\{0.01, 0.03, 0.05, 0.07, 0.1\}$. In addition, we use eight workers to compose a LINE graph, and the batch size on each worker is set to 32. For the compression operator, we use Top-$K$ in our experiments, where $K = 10\%$.

### 5.2. Experimental Results

In Figure 1, we plot the AUC score of the test set versus the number of epochs and versus the communication cost (megabytes, MB) for all data sets. We have two observations: First, our algorithm, CE-DSGDAR, can achieve almost the same AUC score as the full-precision baselines,

which indicates that the compression operation in our algorithm does not hurt the convergence performance; Second, our algorithm enjoy much smaller communication costs than full-precision baselines. These two observations confirm the effectiveness of our algorithms in saving communication costs and preserving convergence performance.

**Ablation studies on communication topology.** In Figure 2, we plot the AUC score versus the number of epochs and communication cost for the CIFAR10 dataset using different communication topologies, including the ring, torus, and random graph. It can be seen that our algorithm is robust to various topologies.

**Ablation studies on compression ratios.** In Figure 3, we compare the performance of our algorithms when using Top-$10\%$ and Top-$5\%$ compression operators. It can be seen that our algorithm is robust to various compression ratios, as it can achieve almost the same AUC score when using different compression ratios.

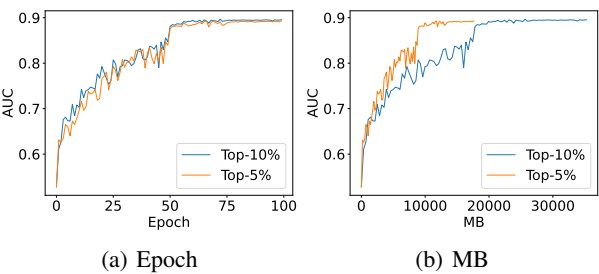

(a) Epoch      (b) MB

*Figure 3.* The test AUC score versus the number of epochs and communication cost for CIFAR10 dataset using different compression ratios.

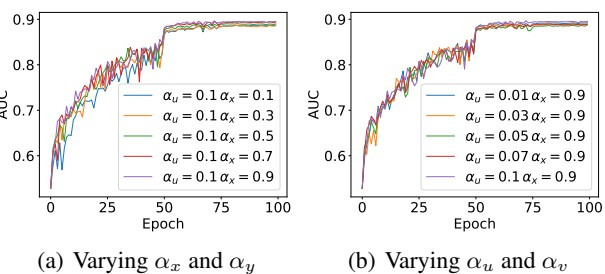

(a) Varying $\alpha_x$ and $\alpha_y$     (b) Varying $\alpha_u$ and $\alpha_v$

*Figure 4.* The test AUC score versus the number of epochs for CIFAR10 dataset. (a) Using different values for $\alpha_x = \alpha_y$ while fixing $\alpha_u = \alpha_v$. (b) Using different values for $\alpha_u = \alpha_v$ while fixing $\alpha_x = \alpha_y$.

**Ablation studies on hyperparameters.** In this experiment, we show the convergence performance of our algorithm when using different values for $\alpha_x$, $\alpha_y$, $\alpha_u$, and $\alpha_v$. Specifically, we fix all the other hyperparameters and tune the value of these four hyperparameters. More specifically, in Figure 4(a), we fix $\alpha_u = \alpha_v = 0.1$ and then tune

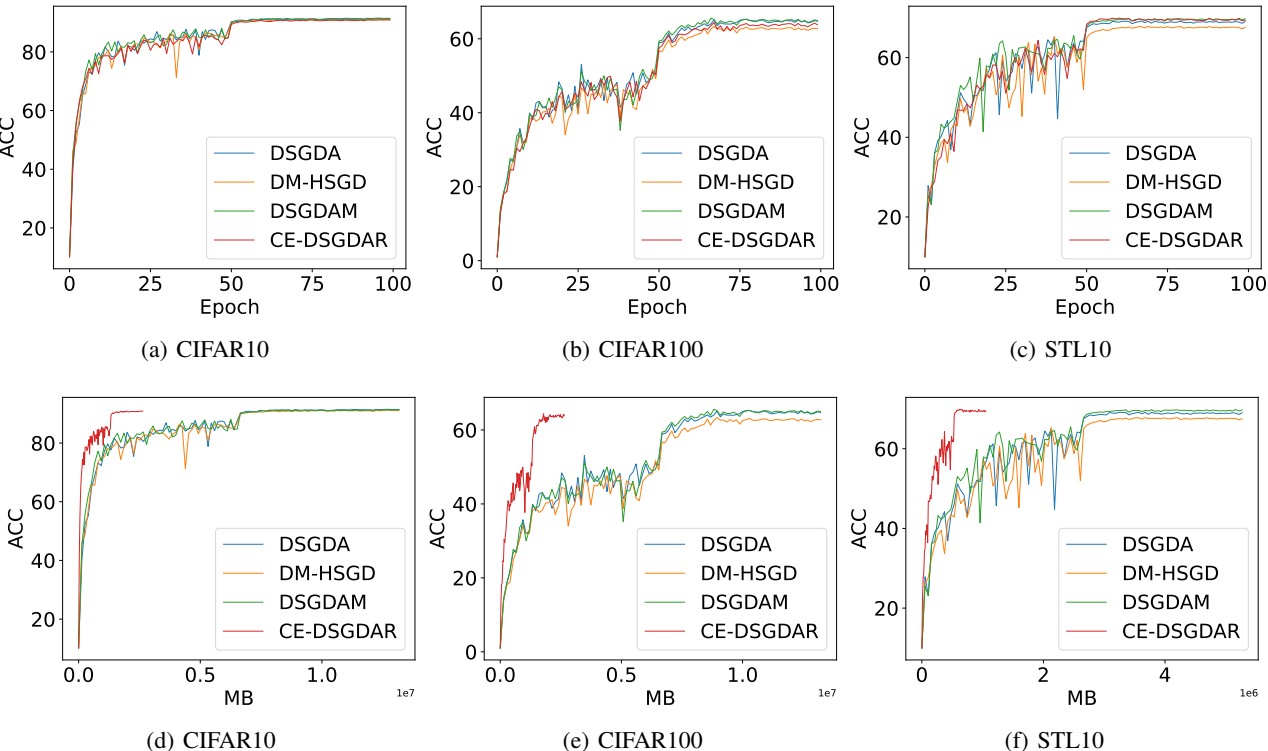

*Figure 5.* The test accuracy versus the number of epochs (1st row) and the communicated megabytes (2nd row).

$\alpha_x = \alpha_y$ from $\{0.1, 0.3, 0.5, 0.7, 0.9\}$ for our algorithm on CIFAR10 dataset. In Figure 4(b), we fix $\alpha_x = \alpha_y = 0.9$ and then tune $\alpha_u = \alpha_v$ from $\{0.01, 0.03, 0.05, 0.07, 0.1\}$. Note that we found that the algorithm does not converge when using the same scale for both $\alpha_x, \alpha_y$ and $\alpha_u, \alpha_v$. A possible reason is that a larger $\alpha_u$ or $\alpha_v$ leads to a larger gradient, which can result in the divergence issue. From these two figures, we can find that the convergence performance of our algorithm is not significantly affected by the value of $\alpha_x$, $\alpha_y$, $\alpha_u$, and $\alpha_v$, confirming the robustness of our algorithm to these hyperparameters.

### 5.3. More Experiments

In this experiment, we apply our algorithm to the fair classification problem (Sharma et al., 2022), which is defined as follows:

$$\min_{x \in \mathbb{R}^d} \max_{y \in \mathcal{Y}} \sum_{c=1}^{C} y_c \ell_c(x) - \frac{\lambda}{2} \|y\|^2 \,, \qquad (12)$$

where $\mathcal{Y} = \{y_c | y_c \geq 0, \sum_{c=1}^{C} y_c = 1\}$. Here, $x$ denotes the classifier's weight, $\ell_c(x)$ represents the loss function for the $c$-th classes and $C$ denotes the total number of classes, $y_c$ is the weight for the $c$-th loss function, and $\lambda > 0$ is a hyperparameter. This loss function is also a nonconvex-strongly-concave problem (Sharma et al., 2022).

In this experiment, we use three benchmark multi-class classification datasets: CIFAR10, CIFAR100 (Krizhevsky et al., 2009), and STL10 (Coates et al., 2011). Similar to the first experiment, the training set is randomly distributed among all workers, serving as their local training sets. The complete and identical testing set is distributed to each worker. In addition, the other experimental settings are the same as the first experiment. In Figure 5, we plot the accuracy of the test set versus the number of epochs and versus the communication cost (megabytes, MB) for all data sets. Similarly, we observe that our algorithm, CE-DSGDAR, achieves nearly the same accuracy as the full-precision baseline methods while significantly reducing communication costs, further confirming the effectiveness of the proposed algorithm.

## 6. Conclusions

Our paper developed a novel communication-efficient decentralized stochastic gradient descent ascent with momentum algorithm. Specifically, we demonstrate how to balance the full-precision local update and the compression residual via novel designs for coefficients. Meanwhile, we developed novel convergence analysis approaches to establish the convergence rate. Extensive experimental results confirm the effectiveness of our algorithm.

## Acknowledgements

We thank anonymous reviewers for constructive comments. Y. Zhang and H. Gao was partially supported by U.S. NSF CAREER 2339545, NSF IIS 2416607, NSF CNS 2107014.

## Impact Statement

This paper presents work whose goal is to advance the field of Machine Learning. There are many potential societal consequences of our work, none which we feel must be specifically highlighted here.

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

## A. Additional Experiments

In this experiment, we apply our algorithm to the robust logistic regression problem by following (Xian et al., 2021). Specifically, its loss function is defined as follows:

$$\min_{x\in\mathbb{R}^d} \max_{y\in\mathcal{Y}} \sum_{i=1}^{n} y_i \ell_i(x) - \frac{1}{2}\lambda_1 \|y - \frac{1}{n}\mathbf{1}\|^2 + \lambda_2 \sum_{j=1}^{d} \frac{\omega x_i^2}{1 + \omega x_i^2} \,, \tag{13}$$

where $\mathcal{Y} = \{y_i | y_i \geq 0, \sum_{i=1}^{n} y_i = 1\}$, $\ell_i(x) = \log(1 + \exp(-b_i a_i^T x))$ is the logistic loss function for the $i$-th sample's feature $a_i \in \mathbb{R}^d$ and label $b_i \in \{-1, 1\}$, $n$ is the number of samples, $d$ is the number of features, $\lambda_1 > 0$ and $\lambda_2 > 0$ are positive hyperparameters.

In this experiment, we use a9a dataset [2] and add an additional baseline, DGDA-VR (Zhang et al., 2024a), as this dataset is small and then we can compute the full gradient for DGDA-VR. For other experimental settings, such as learning rates, batch size, and communication topology, they are same as the first experiment. In Figure 6, we plot the loss function value with respect to both epochs and communication costs. Both plots confirm the effectiveness of our algorithm in reducing communication costs while preserving convergence performance.

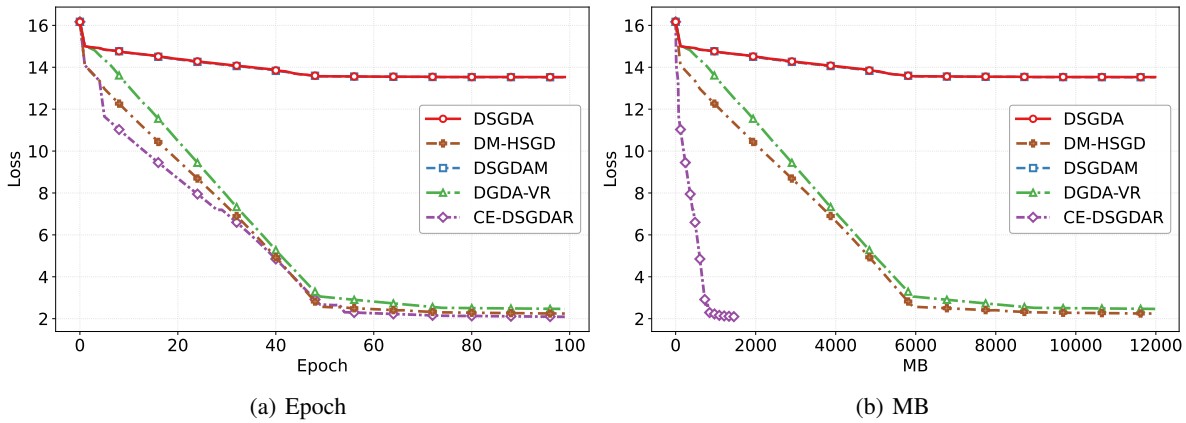

| (a) Epoch | (b) MB |

*Figure 6.* The loss function value versus the number of epochs and the communicated megabytes.

## B. Terminology

To establish the convergence rate, we introduce the following terminologies:

$$X_t = [x_t^{(1)}, x_t^{(2)}, \cdots, x_t^{(K)}] \,, \ Y_t = [y_t^{(1)}, y_t^{(2)}, \cdots, y_t^{(K)}] \,, \ \tilde{X}_t = [\tilde{x}_t^{(1)}, \tilde{x}_t^{(2)}, \cdots, \tilde{x}_t^{(K)}] \,, \ \tilde{Y}_t = [\tilde{y}_t^{(1)}, \tilde{y}_t^{(2)}, \cdots, \tilde{y}_t^{(K)}] \,,$$

$$\hat{X}_t = [\hat{x}_t^{(1)}, \hat{x}_t^{(2)}, \cdots, \hat{x}_t^{(K)}] \,, \hat{Y}_t = [\hat{y}_t^{(1)}, \hat{y}_t^{(2)}, \cdots, \hat{y}_t^{(K)}] \,, \ U_t = [u_t^{(1)}, u_t^{(2)}, \cdots, u_t^{(K)}] \,, \ V_t = [v_t^{(1)}, v_t^{(2)}, \cdots, v_t^{(K)}] \,,$$

$$\tilde{U}_t = [\tilde{u}_t^{(1)}, \tilde{u}_t^{(2)}, \cdots, \tilde{u}_t^{(K)}] \,, \ \tilde{V}_t = [\tilde{v}_t^{(1)}, \tilde{v}_t^{(2)}, \cdots, \tilde{v}_t^{(K)}] \,, \ \hat{U}_t = [\hat{u}_t^{(1)}, \hat{u}_t^{(2)}, \cdots, \hat{u}_t^{(K)}] \,, \ \hat{V}_t = [\hat{v}_t^{(1)}, \hat{v}_t^{(2)}, \cdots, \hat{v}_t^{(K)}] \,,$$

$$G_t = [g_t^{(1)}, g_t^{(2)}, \cdots, g_t^{(K)}] \,, \ H_t = [h_t^{(1)}, h_t^{(2)}, \cdots, h_t^{(K)}] \,,$$

$$\bar{X}_t = [\bar{x}_t, \bar{x}_t, \cdots, \bar{x}_t] \,, \ \bar{Y}_t = [\bar{y}_t, \bar{y}_t, \cdots, \bar{y}_t] \,, \ \bar{\tilde{X}}_t = [\bar{\tilde{x}}_t, \bar{\tilde{x}}_t, \cdots, \bar{\tilde{x}}_t] \,, \ \bar{\tilde{Y}}_t = [\bar{\tilde{x}}_t, \bar{\tilde{x}}_t, \cdots, \bar{\tilde{x}}_t] \,,$$

$$\bar{U}_t = [\bar{u}_t, \bar{u}_t, \cdots, \bar{u}_t] \,, \ \bar{V}_t = [\bar{v}_t, \bar{v}_t, \cdots, \bar{v}_t] \,, \ \bar{\tilde{U}}_t = [\bar{\tilde{u}}_t, \bar{\tilde{u}}_t, \cdots, \bar{\tilde{u}}_t] \,, \ \bar{\tilde{V}}_t = [\bar{\tilde{v}}_t, \bar{\tilde{v}}_t, \cdots, \bar{\tilde{v}}_t] \,. \tag{14}$$

Based on these terminologies, it is easy to obtain

$$\tilde{X}_{t+1} = X_t - \gamma_x \eta U_t \,, \quad \hat{X}_{t+1} = \hat{X}_t + \mathcal{C}(\tilde{X}_{t+1} - \hat{X}_t) \,, \quad X_{t+1} = \tilde{X}_{t+1} + \alpha_x \eta \hat{X}_{t+1}(W - I) \,,$$

$$\tilde{U}_{t+1} = U_t - G_t + G_{t+1} \,, \quad \hat{U}_{t+1} = \hat{U}_t + \mathcal{C}(\tilde{U}_{t+1} - \hat{U}_t) \,, \quad U_{t+1} = \tilde{U}_{t+1} + \alpha_u \hat{U}_{t+1}(W - I) \,,$$

$$\tilde{Y}_{t+1} = Y_t + \gamma_y \eta V_t \,, \quad \hat{Y}_{t+1} = \hat{Y}_t + \mathcal{C}(\tilde{Y}_{t+1} - \hat{Y}_t) \,, \quad Y_{t+1} = \tilde{Y}_{t+1} + \alpha_y \eta \hat{Y}_{t+1}(W - I) \,,$$

[2] https://www.csie.ntu.edu.tw/~cjlin/libsvmtools/datasets/

$$\tilde{V}_{t+1} = V_t - H_t + H_{t+1} \ , \quad \hat{V}_{t+1} = \hat{V}_t + \mathcal{C}(\tilde{V}_{t+1} - \hat{V}_t) \ , \quad V_{t+1} = \tilde{V}_{t+1} + \alpha_v \hat{V}_{t+1}(W - I) \ . \tag{15}$$

Additionally, since $W$ is a double stochastic matrix, it is easy to know $(W - I)\frac{1}{K}\mathbf{1}\mathbf{1}^T = 0$ where $I$ is an identity matrix. Then, based on $\bar{X}_t = X_t \frac{1}{K}\mathbf{1}\mathbf{1}^T$, we can obtain

$$\bar{X}_{t+1} = \bar{\tilde{X}}_{t+1} = \bar{X}_t - \gamma_x \eta \bar{U}_t \ , \quad \bar{U}_{t+1} = \bar{\tilde{U}}_{t+1} = \bar{U}_t + \bar{G}_{t+1} - \bar{G}_t \ ,$$
$$\bar{Y}_{t+1} = \bar{\tilde{Y}}_{t+1} = \bar{Y}_t + \gamma_y \eta \bar{V}_t \ , \quad \bar{V}_{t+1} = \bar{\tilde{V}}_{t+1} = \bar{V}_t + \bar{H}_{t+1} - \bar{H}_t \ . \tag{16}$$

Furthermore, it is easy to know $\bar{U}_t = \bar{G}_t$ and $\bar{V}_t = \bar{H}_t$.

## C. Proof Sketch

### C.1. Key Idea

The convergence of our two algorithms is affected by the *gradient estimation error*, *consensus error*, and *compression error* regarding both primal and dual variables. Therefore, we need to investigate all of them to establish the convergence rate of our algorithms. Then, *the key idea of our proof is to use a potential function to unify all these error bounds to investigate how they affect the convergence.*

To address these challenges when proving Theorem 4.4, we develop a potential function to unify the gradient estimation error , consensus error , and compression error as follows:

$$P_t = \mathbb{E}[F(\bar{x}_t)] + c_0 \mathbb{E}[\|\bar{y}_t - y^*(\bar{x}_t)\|^2]$$

$$+ c_1 \mathbb{E}\left[\left\|\frac{1}{K}\sum_{k=1}^{K} g_t^{(k)} - \frac{1}{K}\sum_{k=1}^{K}\nabla_x f^{(k)}(x_t^{(k)}, y_t^{(k)})\right\|^2\right]$$

$$+ c_2 \mathbb{E}\left[\left\|\frac{1}{K}\sum_{k=1}^{K} h_t^{(k)} - \frac{1}{K}\sum_{k=1}^{K}\nabla_y f^{(k)}(x_t^{(k)}, y_t^{(k)})\right\|^2\right]$$

$$+ c_{11}\frac{1}{K}\sum_{k=1}^{K}\mathbb{E}\left[\left\|g_t^{(k)} - \nabla_x f^{(k)}(x_t^{(k)}, y_t^{(k)})\right\|^2\right]$$

$$+ c_{12}\frac{1}{K}\sum_{k=1}^{K}\mathbb{E}\left[\left\|h_t^{(k)} - \nabla_y f^{(k)}(x_t^{(k)}, y_t^{(k)})\right\|^2\right]$$

$$+ c_3\frac{1}{K}\mathbb{E}[\|X_t - \bar{X}_t\|_F^2] + c_4\frac{1}{K}\mathbb{E}[\|Y_t - \bar{Y}_t\|_F^2]$$

$$+ c_5\frac{1}{K}\mathbb{E}[\|U_t - \bar{U}_t\|_F^2] + c_6\frac{1}{K}\mathbb{E}[\|V_t - \bar{V}_t\|_F^2]$$

$$+ c_7\frac{1}{K}\mathbb{E}[\|\hat{U}_t - U_t\|_F^2] + c_8\frac{1}{K}\mathbb{E}[\|\hat{V}_t - V_t\|_F^2]$$

$$+ c_9\frac{1}{K}\mathbb{E}[\|\tilde{X}_{t+1} - \hat{X}_t\|_F^2] + c_{10}\frac{1}{K}\mathbb{E}[\|\tilde{Y}_{t+1} - \hat{Y}_t\|_F^2] \ . \tag{17}$$

Then, we establish their error bounds in Lemmas D.10, D.2, D.3, D.4, D.5. Based on the potential function and the error bound of each term, we study how the potential function decreases across iterations for establishing the convergence rate, i.e., establishing the following upper bound

$$P_{t+1} - P_t \leq -\frac{\gamma_x \eta}{2}\mathbb{E}\left[\|\nabla F(\bar{x}_t)\|^2\right]$$
$$+ c_1\frac{2\rho_x^2\eta^4\sigma^2}{K} + c_2\frac{2\rho_y^2\eta^4\sigma^2}{K} + c_{11}2\rho_x^2\eta^4\sigma^2 + c_{12}2\rho_y^2\eta^4\sigma^2 + C_3 3\rho_x^2\eta^4\sigma^2 + C_4 3\rho_y^2\eta^4\sigma^2 \ ,$$

where $\{c_i\}_{i=0}^{12}$ will be carefully determined to guarantee convergence.

### C.2. Unique Challenge

A key challenge is to determine the coefficient $\{c_i\}_{i=0}^{12}$. Specifically, **the difficulty lies in the circle dependence among these coefficients due to the interaction between the primal and dual variables in the presence of the error feedback**

**mechanism.** More specifically, based on the aforementioned error bounds, we can obtain the following inequality (i.e., Eq. (45)):

$$
P_{t+1} - P_t
$$

$$
\leq -\frac{\gamma_x \eta}{2} \mathbb{E}\left[\|\nabla F(\bar{x}_t)\|^2\right] + c_1 \frac{2\rho_x^2 \eta^4 \sigma^2}{K} + c_2 \frac{2\rho_y^2 \eta^4 \sigma^2}{K} + c_{11} 2\rho_x^2 \eta^4 \sigma^2 + c_{12} 2\rho_y^2 \eta^4 \sigma^2 + C_3 3\rho_x^2 \eta^4 \sigma^2 + C_4 3\rho_y^2 \eta^4 \sigma^2
$$

$$
+ \underbrace{\left(-\frac{\gamma_x \eta}{4} + c_0 \frac{25\eta\gamma_x^2 L^2}{6\gamma_y \mu^3} + C_5 \frac{9\gamma_x^2}{2\alpha_x^2}\right)}_{\leq\, 0} \mathbb{E}\left[\|\bar{g}_t\|^2\right]
$$

$$
+ \underbrace{\left(-c_0 \frac{3\eta\gamma_y^2}{4} + C_6 \frac{9\gamma_y^2}{2\alpha_y^2}\right)}_{\leq\, 0} \mathbb{E}\left[\|\bar{h}_t\|^2\right]
$$

$$
+ \underbrace{\left(2\gamma_x \eta L^2 - c_0 \frac{\gamma_y \eta \mu}{4}\right)}_{\leq\, 0} \mathbb{E}\left[\|y^*(\bar{x}_t) - \bar{y}_t\|^2\right]
$$

$$
+ \underbrace{\left(2\gamma_x \eta L^2 + c_0 \frac{25\gamma_y \eta L^2}{3\mu} - c_3 \frac{\alpha_x \eta(1-\lambda)}{2} + C_5\right)}_{\leq\, 0} \frac{1}{K}\mathbb{E}\left[\|\bar{X}_t - X_t\|_F^2\right]
$$

$$
+ \underbrace{\left(2\gamma_x \eta L^2 + c_0 \frac{25\gamma_y \eta L^2}{3\mu} - c_4 \frac{\alpha_y \eta(1-\lambda)}{2} + C_6\right)}_{\leq\, 0} \frac{1}{K}\mathbb{E}\left[\|\bar{Y}_t - Y_t\|_F^2\right]
$$

$$
+ \underbrace{\left(2\gamma_x \eta - c_1 \rho_x \eta^2\right)}_{\leq\, 0} \mathbb{E}\left[\|\frac{1}{K}\sum_{k=1}^{K}\nabla_x f^{(k)}(x_t^{(k)}, y_t^{(k)}) - \frac{1}{K}\sum_{k=1}^{K} g_t^{(k)}\|^2\right]
$$

$$
+ \underbrace{\left(c_0 \frac{25\gamma_y \eta}{3\mu} - c_2 \rho_y \eta^2\right)}_{\leq\, 0} \mathbb{E}\left[\|\frac{1}{K}\sum_{k=1}^{K}\nabla_y f^{(k)}(x_t^{(k)}, y_t^{(k)}) - \frac{1}{K}\sum_{k=1}^{K} h_t^{(k)}\|^2\right]
$$

$$
+ \underbrace{\left(c_3 \frac{3\gamma_x^2 \eta}{\alpha_x(1-\lambda)} - c_5 \frac{\alpha_u(1-\lambda)}{2} + c_7 \frac{36\alpha_u^2}{\delta} + c_9 \frac{72\gamma_x^2 \alpha_u^2 \eta^2}{\delta} + C_5 \frac{9\gamma_x^2}{2\alpha_x^2}\right)}_{\leq\, 0} \frac{1}{K}\mathbb{E}\left[\|U_t - \bar{U}_t\|_F^2\right]
$$

$$
+ \underbrace{\left(c_4 \frac{3\gamma_y^2 \eta}{\alpha_y(1-\lambda)} - c_6 \frac{\alpha_v(1-\lambda)}{2} + c_8 \frac{36\alpha_v^2}{\delta} + c_{10} \frac{72\gamma_y^2 \alpha_v^2 \eta^2}{\delta} + C_6 \frac{9\gamma_y^2}{2\alpha_y^2}\right)}_{\leq\, 0} \frac{1}{K}\mathbb{E}\left[\|V_t - \bar{V}_t\|_F^2\right]
$$

$$
+ \underbrace{\left(c_3 \frac{4\alpha_x \eta(1-\delta)}{1-\lambda} - c_9 \frac{\delta}{2} + C_5\right)}_{\leq\, 0} \frac{1}{K}\mathbb{E}\left[\|\tilde{X}_{t+1} - \hat{X}_t\|_F^2\right]
$$

$$
+ \underbrace{\left(c_4 \frac{4\alpha_y \eta(1-\delta)}{1-\lambda} - c_{10} \frac{\delta}{2} + C_6\right)}_{\leq\, 0} \frac{1}{K}\mathbb{E}\left[\|\tilde{Y}_{t+1} - \hat{Y}_t\|_F^2\right]
$$

$$
+ \underbrace{\left(c_5 \frac{8\alpha_u(1-\delta)}{1-\lambda} - c_7 \frac{\delta}{8} + c_9 \frac{144\gamma_x^2 \alpha_u^2 \eta^2}{\delta}\right)}_{\leq\, 0} \frac{1}{K}\mathbb{E}\left[\|U_t - \hat{U}_t\|_F^2\right]
$$

$$
+ \underbrace{\left(c_6 \frac{8\alpha_v(1-\delta)}{1-\lambda} - c_8 \frac{\delta}{8} + c_{10} \frac{144\gamma_y^2 \alpha_v^2 \eta^2}{\delta}\right)}_{\leq\, 0} \frac{1}{K}\mathbb{E}\left[\|V_t - \hat{V}_t\|_F^2\right]
$$

$$+ \underbrace{(-c_{11}\rho_x\eta^2 + C_3 3\rho_x^2\eta^4)}_{\leq 0} \frac{1}{K}\sum_{k=1}^{K}\mathbb{E}\left[\left\|g_t^{(k)} - \nabla_x f^{(k)}(x_t^{(k)}, y_t^{(k)})\right\|^2\right]$$

$$+ \underbrace{(-c_{12}\rho_y\eta^2 + C_4 3\rho_y^2\eta^4)}_{\leq 0} \frac{1}{K}\sum_{k=1}^{K}\mathbb{E}\left[\left\|h_t^{(k)} - \nabla_y f^{(k)}(x_t^{(k)}, y_t^{(k)})\right\|^2\right] . \tag{18}$$

where $C_3$, $C_4$, $C_5$, and $C_6$ are defined in Eq. (46). Then, to establish the convergence rate, we need to eliminate all terms except the first line in this upper bound. Specifically, we need to select appropriate $\{c_i\}_{i=0}^{12}$ such that the coefficients of those terms are non-positive as shown in this inequality. However, due to the interaction between the primal and dual variables, there exists interdependence among these coefficients. As a result, it is challenging to determine the value of $\{c_i\}_{i=0}^{12}$.

For example, when eliminating $\mathbb{E}\left[\|U_t - \bar{U}_t\|_F^2\right]$, we should enforce

$$c_3 \frac{3\gamma_x^2\eta}{\alpha_x(1-\lambda)} + c_7 \frac{36\alpha_u^2}{\delta} + c_9 \frac{72\gamma_x^2\alpha_u^2\eta^2}{\delta} + \frac{9\gamma_x^2}{2\alpha_x^2}C_5 \leq c_5 \frac{\alpha_u(1-\lambda)}{2} , \tag{19}$$

which indicates $c_5$ depends on $\{c_1, c_2, c_3, c_6, c_7, c_8, c_9, c_{10}, c_{11}, c_{12}\}$.

When eliminating $\mathbb{E}\left[\|\tilde{X}_{t+1} - \hat{X}_t\|_F^2\right]$, we should enforce

$$c_3 \frac{4\alpha_x\eta(1-\delta)}{1-\lambda} + C_5 \leq c_9 \frac{\delta}{2} , \tag{20}$$

which indicates $c_9$ depends on $\{c_3, c_1, c_2, c_5, c_6, c_7, c_8, c_9, c_{10}, c_{11}, c_{12}\}$.

When eliminating $\mathbb{E}\left[\|U_t - \hat{U}_t\|_F^2\right]$, we should enforce

$$c_5 \frac{8\alpha_u(1-\delta)}{1-\lambda} + c_9 \frac{144\gamma_x^2\alpha_u^2\eta^2}{\delta} \leq c_7 \frac{\delta}{8} , \tag{21}$$

which indicates $c_7$ depends on $\{c_5, c_9\}$.

**In summary, $\{c_5, c_7, c_9\}$ depends on each other, which is shown in Figure 7. Other coefficients have similar interdependence. This kind of circle dependence makes it difficult to determine the values of these coefficients.**

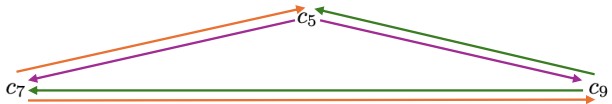

*Figure 7.* An illustration of the **circle dependence** among $\{c_5, c_7, c_9\}$: $c_5$ depends on $\{c_7, c_9\}$; $c_7$ depends on $\{c_5, c_9\}$; and $c_9$ depends on $\{c_5, c_7\}$.

**Comparison with the full-precision counterpart.** The full-precision decentralized minimax optimization algorithm does not have the terms: $\mathbb{E}\left[\|\tilde{X}_{t+1} - \hat{X}_t\|_F^2\right]$, $\mathbb{E}\left[\|\tilde{Y}_{t+1} - \hat{Y}_t\|_F^2\right]$, $\mathbb{E}\left[\|U_t - \hat{U}_t\|_F^2\right]$, and $\mathbb{E}\left[\|V_t - \hat{V}_t\|_F^2\right]$, which means $c_7 = 0$, $c_8 = 0$, $c_9 = 0$, and $c_{10} = 0$. Then, there does not exist interdependence among $\{c_5, c_7, c_9\}$. In other words, the error feedback mechanism makes the convergence analysis more challenging.

### C.3. Novel Solution

To break down the interdependence among different coefficients, we propose to use some coefficients to represent the others, and then determine the latter first. Just as shown in Eq. (50), we use $\{c_5, c_6, c_9, c_{10}\}$ to represent $\{c_7, c_8, c_{11}, c_{12}\}$. In this way, we can break down the dependence on $\{c_5, c_9\}$ for $c_7$. Then, we solve the inequalities to determine the value of $\{c_5, c_6, c_9, c_{10}\}$ first. After that, we can obtain the value of $\{c_7, c_8, c_{11}, c_{12}\}$.

However, there is still interdependence between $c_5$ and $c_9$. Just as shown in Eq. (64), when eliminating $\mathbb{E}\left[\|\tilde{X}_{t+1} - \hat{X}_t\|_F^2\right]$, $c_9$ **is affected by** $c_5$, **where** $c_5$ **is an unknown value at that time**. By examining Eq. (64), we found that there exist some terms that are independent of $\{c_i\}_{i=1}^{12}$. Then, we propose making $c_9$ equal to those terms as shown in Eq. (75), which means that the value of $c_9$ is determined. Then, since the value of $\{c_5, c_6, c_{10}\}$ is not available at that time, we will solve Eqs. (69-74) after all values of $\{c_5, c_6, c_9, c_{10}\}$ are determined. With this strategy, we are able to determine the value of all coefficients, completing the proof.

# D. Convergence Analysis of CD-SGDAR

## D.1. Useful Lemma

**Lemma D.1.** *(Zhao et al., 2022) Assume $W$ satisfies Assumption 3.2, then the spectral gap of $I + \alpha(W - I)$ is $\alpha(1 - \lambda)$ when $\alpha \in (0, 1)$.*

**Lemma D.2.** *Given Assumptions 3.1-4.3, $\alpha_x\eta \in (0, 1)$, and $\alpha_y\eta \in (0, 1)$, we have*

$$\|X_{t+1} - \bar{X}_{t+1}\|_F^2 \leq (1 - \frac{\alpha_x\eta(1-\lambda)}{2})\|X_t - \bar{X}_t\|_F^2 + \frac{3\gamma_x^2\eta}{\alpha_x(1-\lambda)}\|U_t - \bar{U}_t\|_F^2 + \frac{4\alpha_x\eta(1-\delta)}{1-\lambda}\|\tilde{X}_{t+1} - \hat{X}_t\|_F^2 \,,$$

$$\|Y_{t+1} - \bar{Y}_{t+1}\|_F^2 \leq (1 - \frac{\alpha_y\eta(1-\lambda)}{2})\|Y_t - \bar{Y}_t\|_F^2 + \frac{3\gamma_y^2\eta}{\alpha_y(1-\lambda)}\|V_t - \bar{V}_t\|_F^2 + \frac{4\alpha_y\eta(1-\delta)}{1-\lambda}\|\tilde{Y}_{t+1} - \hat{Y}_t\|_F^2 \,. \qquad (22)$$

*Proof.* Given $\alpha_x\eta \in (0, 1)$, we have

$$\|X_{t+1} - \bar{X}_{t+1}\|_F^2$$
$$= \|\tilde{X}_{t+1} + \alpha_x\eta\hat{X}_{t+1}(W - I) - \bar{\tilde{X}}_{t+1}\|_F^2$$
$$= \|\tilde{X}_{t+1} + \alpha_x\eta(\hat{X}_t + \mathcal{C}(\tilde{X}_{t+1} - \hat{X}_t))(W - I) - \bar{\tilde{X}}_{t+1}\|_F^2$$
$$= \|\tilde{X}_{t+1} + \alpha_x\eta\tilde{X}_{t+1}(W - I) + \alpha_x\eta(\hat{X}_t - \tilde{X}_{t+1} + \mathcal{C}(\tilde{X}_{t+1} - \hat{X}_t))(W - I) - \bar{\tilde{X}}_{t+1}\|_F^2$$
$$= \|\tilde{X}_{t+1}(I + \alpha_x\eta(W - I)) - \bar{\tilde{X}}_{t+1} + \alpha_x\eta(\hat{X}_t - \tilde{X}_{t+1} + \mathcal{C}(\tilde{X}_{t+1} - \hat{X}_t))(W - I)\|_F^2$$
$$\leq (1 + c_1)\|\tilde{X}_{t+1}(I + \alpha_x\eta(W - I)) - \bar{\tilde{X}}_{t+1}\|_F^2 + (1 + c_1^{-1})\alpha_x^2\eta^2\|(\hat{X}_t - \tilde{X}_{t+1} + \mathcal{C}(\tilde{X}_{t+1} - \hat{X}_t))(W - I)\|_F^2$$
$$\leq (1 + c_1)(1 - \alpha_x\eta(1-\lambda))^2\|\tilde{X}_{t+1} - \bar{\tilde{X}}_{t+1}\|_F^2 + 4(1 + c_1^{-1})\alpha_x^2\eta^2(1-\delta)\|\tilde{X}_{t+1} - \hat{X}_t\|_F^2$$
$$\leq (1 - \alpha_x\eta(1-\lambda))\|\tilde{X}_{t+1} - \bar{\tilde{X}}_{t+1}\|_F^2 + \frac{4\alpha_x\eta(1-\delta)}{1-\lambda}\|\tilde{X}_{t+1} - \hat{X}_t\|_F^2$$
$$\leq (1 - \alpha_x\eta(1-\lambda))(1 + c_2)\|X_t - \bar{X}_t\|_F^2 + (1 - \alpha_x\eta(1-\lambda))(1 + c_2^{-1})\gamma_x^2\eta^2\|U_t - \bar{U}_t\|_F^2$$
$$\quad + \frac{4\alpha_x\eta(1-\delta)}{1-\lambda}\|\tilde{X}_{t+1} - \hat{X}_t\|_F^2$$
$$\leq (1 - \frac{\alpha_x\eta(1-\lambda)}{2})\|X_t - \bar{X}_t\|_F^2 + \frac{3\gamma_x^2\eta}{\alpha_x(1-\lambda)}\|U_t - \bar{U}_t\|_F^2 + \frac{4\alpha_x\eta(1-\delta)}{1-\lambda}\|\tilde{X}_{t+1} - \hat{X}_t\|_F^2 \,, \qquad (23)$$

where $c_1 = \frac{\alpha_x\eta(1-\lambda)}{1-\alpha_x\eta(1-\lambda)}$ and $c_2 = \frac{\alpha_x\eta(1-\lambda)}{2}$, the sixth step holds due to $\|W - I\|_2 \leq 2$, Lemma D.1, and Assumption 3.1. Similarly, we can prove the second inequality.

$\square$

**Lemma D.3.** *Given Assumptions 3.1-4.3, $\alpha_u \in (0, 1)$, and $\alpha_v \in (0, 1)$, we have*

$$\|U_{t+1} - \bar{U}_{t+1}\|_F^2 \leq (1 - \frac{\alpha_u(1-\lambda)}{2})\|U_t - \bar{U}_t\|_F^2 + \frac{11}{\alpha_u(1-\lambda)}\|G_{t+1} - G_t\|_F^2 + \frac{8\alpha_u(1-\delta)}{1-\lambda}\|U_t - \hat{U}_t\|_F^2 \,,$$

$$\|V_{t+1} - \bar{V}_{t+1}\|_F^2 \leq (1 - \frac{\alpha_v(1-\lambda)}{2})\|V_t - \bar{V}_t\|_F^2 + \frac{11}{\alpha_v(1-\lambda)}\|H_{t+1} - H_t\|_F^2 + \frac{8\alpha_v(1-\delta)}{1-\lambda}\|V_t - \hat{V}_t\|_F^2 \,. \qquad (24)$$

*Proof.* Given $\alpha_u \in (0, 1)$, we have

$$\|U_{t+1} - \bar{U}_{t+1}\|_F^2$$

$$= \|\tilde{U}_{t+1} + \alpha_u \hat{U}_{t+1}(W - I) - \bar{\tilde{U}}_{t+1}\|_F^2$$

$$= \|\tilde{U}_{t+1} + \alpha_u(\hat{U}_t + \mathcal{C}(\tilde{U}_{t+1} - \hat{U}_t))(W - I) - \bar{\tilde{U}}_{t+1}\|_F^2$$

$$= \|\tilde{U}_{t+1} + \alpha_u \tilde{U}_{t+1}(W - I) + \alpha_u(\hat{U}_t - \tilde{U}_{t+1} + \mathcal{C}(\tilde{U}_{t+1} - \hat{U}_t))(W - I) - \bar{\tilde{U}}_{t+1}\|_F^2$$

$$= \|\tilde{U}_{t+1}(I + \alpha_u(W - I)) - \bar{\tilde{U}}_{t+1} + \alpha_u(\hat{U}_t - \tilde{U}_{t+1} + \mathcal{C}(\tilde{U}_{t+1} - \hat{U}_t))(W - I)\|_F^2$$

$$\leq (1 + c_1)\|\tilde{U}_{t+1}(I + \alpha_u(W - I)) - \bar{\tilde{U}}_{t+1}\|_F^2 + (1 + c_1^{-1})\alpha_u^2\|(\hat{U}_t - \tilde{U}_{t+1} + \mathcal{C}(\tilde{U}_{t+1} - \hat{U}_t))(W - I)\|_F^2$$

$$\leq (1 + c_1)(1 - \alpha_u(1 - \lambda))^2\|\tilde{U}_{t+1} - \bar{\tilde{U}}_{t+1}\|_F^2 + 4(1 + c_1^{-1})\alpha_u^2(1 - \delta)\|\tilde{U}_{t+1} - \hat{U}_t\|_F^2$$

$$\leq (1 - \alpha_u(1 - \lambda))\|\tilde{U}_{t+1} - \bar{\tilde{U}}_{t+1}\|_F^2 + \frac{4\alpha_u(1 - \delta)}{1 - \lambda}\|\tilde{U}_{t+1} - \hat{U}_t\|_F^2$$

$$\leq (1 - \alpha_u(1 - \lambda))(1 + c_2)\|U_t - \bar{U}_t\|_F^2$$
$$+ (1 - \alpha_u(1 - \lambda))(1 + c_2^{-1})\|G_{t+1} - G_t - \bar{G}_{t+1} + \bar{G}_t\|_F^2 + \frac{4\alpha_u(1 - \delta)}{1 - \lambda}\|\tilde{U}_{t+1} - \hat{U}_t\|_F^2$$

$$\leq (1 - \frac{\alpha_u(1 - \lambda)}{2})\|U_t - \bar{U}_t\|_F^2 + \frac{3}{\alpha_u(1 - \lambda)}\|G_{t+1} - G_t - \bar{G}_{t+1} + \bar{G}_t\|_F^2 + \frac{4\alpha_u(1 - \delta)}{1 - \lambda}\|U_t - G_t + G_{t+1} - \hat{U}_t\|_F^2$$

$$\leq (1 - \frac{\alpha_u(1 - \lambda)}{2})\|U_t - \bar{U}_t\|_F^2 + \frac{11}{\alpha_u(1 - \lambda)}\|G_{t+1} - G_t\|_F^2 + \frac{8\alpha_u(1 - \delta)}{1 - \lambda}\|U_t - \hat{U}_t\|_F^2, \tag{25}$$

where $c_1 = \frac{\alpha_u(1-\lambda)}{1-\alpha_u(1-\lambda)}$ and $c_2 = \frac{\alpha_u(1-\lambda)}{2}$, the sixth step holds due to $\|W - I\|_2 \leq 2$, Lemma D.1, and Assumption 3.1, the last step follows from $\alpha_u < 1$, $1 - \delta < 1$, and $\|G_{t+1} - G_t - \bar{G}_{t+1} + \bar{G}_t\|_F^2 \leq \|G_{t+1} - G_t\|_F^2$. Similarly, we can prove the second inequality. $\qquad\square$

**Lemma D.4.** *Given Assumptions 3.1-4.3, we have*

$$\|\tilde{X}_{t+2} - \hat{X}_{t+1}\|_F^2 \leq (1 - \frac{\delta}{2})\|\tilde{X}_{t+1} - \hat{X}_t\|_F^2 + \frac{6}{\delta}\|X_{t+1} - X_t\|_F^2 + \frac{6\gamma_x^2\eta^2}{\delta}\|U_{t+1} - U_t\|_F^2,$$

$$\|\tilde{Y}_{t+2} - \hat{Y}_{t+1}\|_F^2 \leq (1 - \frac{\delta}{2})\|\tilde{Y}_{t+1} - \hat{Y}_t\|_F^2 + \frac{6}{\delta}\|Y_{t+1} - Y_t\|_F^2 + \frac{6\gamma_y^2\eta^2}{\delta}\|V_{t+1} - V_t\|_F^2. \tag{26}$$

*Proof.*

$$\|\tilde{X}_{t+2} - \hat{X}_{t+1}\|_F^2$$

$$= \|\tilde{X}_{t+2} - \hat{X}_t - \mathcal{C}(\tilde{X}_{t+1} - \hat{X}_t)\|_F^2$$

$$= \|\tilde{X}_{t+2} - \tilde{X}_{t+1} + \tilde{X}_{t+1} - \hat{X}_t - \mathcal{C}(\tilde{X}_{t+1} - \hat{X}_t)\|_F^2$$

$$\leq (1 + c)\|\tilde{X}_{t+1} - \hat{X}_t - \mathcal{C}(\tilde{X}_{t+1} - \hat{X}_t)\|_F^2 + (1 + c^{-1})\|\tilde{X}_{t+2} - \tilde{X}_{t+1}\|_F^2$$

$$\leq (1 + c)(1 - \delta)\|\tilde{X}_{t+1} - \hat{X}_t\|_F^2 + (1 + c^{-1})\|X_{t+1} - \gamma_x\eta U_{t+1} - X_t + \gamma_x\eta U_t\|_F^2$$

$$\leq (1 - \frac{\delta}{2})\|\tilde{X}_{t+1} - \hat{X}_t\|_F^2 + \frac{6}{\delta}\|X_{t+1} - X_t\|_F^2 + \frac{6\gamma_x^2\eta^2}{\delta}\|U_{t+1} - U_t\|_F^2, \tag{27}$$

where $c = \frac{\delta}{2}$, the fourth step hold due to Assumption 3.1. Similarly, we can prove the second inequality.

$\qquad\square$

**Lemma D.5.** *Given Assumptions 3.1-4.3, $\alpha_u \leq \frac{\delta}{16}$, and $\alpha_v \leq \frac{\delta}{16}$, we have*

$$\|\hat{U}_{t+1} - U_{t+1}\|_F^2 \leq (1 - \frac{\delta}{8})\|U_t - \hat{U}_t\|_F^2 + \frac{46}{\delta}\|G_{t+1} - G_t\|_F^2 + \frac{36\alpha_u^2}{\delta}\|U_t - \bar{U}_t\|_F^2,$$

$$\|\hat{V}_{t+1} - V_{t+1}\|_F^2 \leq (1 - \frac{\delta}{8})\|V_t - \hat{V}_t\|_F^2 + \frac{46}{\delta}\|H_{t+1} - H_t\|_F^2 + \frac{36\alpha_v^2}{\delta}\|V_t - \bar{V}_t\|_F^2. \tag{28}$$

*Proof.*

$$\|\hat{U}_{t+1} - U_{t+1}\|_F^2$$

$$= \|\hat{U}_t + \mathcal{C}(\tilde{U}_{t+1} - \hat{U}_t) - \tilde{U}_{t+1} - \alpha_u \hat{U}_{t+1}(W - I)\|_F^2$$

$$= \|\hat{U}_t - \tilde{U}_{t+1} + \mathcal{C}(\tilde{U}_{t+1} - \hat{U}_t) - \alpha_u(\hat{U}_t + \mathcal{C}(\tilde{U}_{t+1} - \hat{U}_t))(W - I)\|_F^2$$

$$= \|\hat{U}_t - \tilde{U}_{t+1} + \mathcal{C}(\tilde{U}_{t+1} - \hat{U}_t) - \alpha_u(\hat{U}_t - \tilde{U}_{t+1} + \mathcal{C}(\tilde{U}_{t+1} - \hat{U}_t))(W - I) - \alpha_u \tilde{U}_{t+1}(W - I)\|_F^2$$

$$= \|(\hat{U}_t - \tilde{U}_{t+1} + \mathcal{C}(\tilde{U}_{t+1} - \hat{U}_t))(I - \alpha_u(W - I)) - \alpha_u \tilde{U}_{t+1}(W - I)\|_F^2$$

$$= \|(\hat{U}_t - \tilde{U}_{t+1} + \mathcal{C}(\tilde{U}_{t+1} - \hat{U}_t))(I - \alpha_u(W - I)) - \alpha_u(\tilde{U}_{t+1} - \bar{\bar{U}}_t)(W - I)\|_F^2$$

$$\leq (1 + c_1)\|(\hat{U}_t - \tilde{U}_{t+1} + \mathcal{C}(\tilde{U}_{t+1} - \hat{U}_t))(I - \alpha_u(W - I))\|_F^2 + (1 + c_1^{-1})\alpha_u^2\|(\tilde{U}_{t+1} - \bar{\bar{U}}_t)(W - I)\|_F^2$$

$$\leq (1 + c_1)(1 + 2\alpha_u)^2\|\hat{U}_t - \tilde{U}_{t+1} + \mathcal{C}(\tilde{U}_{t+1} - \hat{U}_t)\|_F^2 + (1 + c_1^{-1})4\alpha_u^2\|\tilde{U}_{t+1} - \bar{\bar{U}}_t\|_F^2$$

$$\leq (1 + c_1)(1 + 2\alpha_u)^2(1 - \delta)\|\tilde{U}_{t+1} - \hat{U}_t\|_F^2 + (1 + c_1^{-1})4\alpha_u^2\|\tilde{U}_{t+1} - \bar{\bar{U}}_t\|_F^2$$

$$\leq (1 - \frac{\delta}{4})\|\tilde{U}_{t+1} - \hat{U}_t\|_F^2 + \frac{12\alpha_u^2}{\delta}\|\tilde{U}_{t+1} - \bar{\bar{U}}_t\|_F^2$$

$$= (1 - \frac{\delta}{4})\|U_t - G_t + G_{t+1} - \hat{U}_t\|_F^2 + \frac{12\alpha_u^2}{\delta}\|U_t - G_t + G_{t+1} - \bar{U}_t\|_F^2$$

$$\leq (1 - \frac{\delta}{4})(1 + c_2)\|U_t - \hat{U}_t\|_F^2 + (1 - \frac{\delta}{4})(1 + c_2^{-1})\|G_{t+1} - G_t\|_F^2 + \frac{36\alpha_u^2}{\delta}\|G_{t+1} - G_t\|_F^2 + \frac{36\alpha_u^2}{\delta}\|U_t - \bar{U}_t\|_F^2$$

$$\leq (1 - \frac{\delta}{8})\|U_t - \hat{U}_t\|_F^2 + \frac{9}{\delta}\|G_{t+1} - G_t\|_F^2 + \frac{36\alpha_u^2}{\delta}\|G_{t+1} - G_t\|_F^2 + \frac{36\alpha_u^2}{\delta}\|U_t - \bar{U}_t\|_F^2$$

$$\leq (1 - \frac{\delta}{8})\|U_t - \hat{U}_t\|_F^2 + \frac{46}{\delta}\|G_{t+1} - G_t\|_F^2 + \frac{36\alpha_u^2}{\delta}\|U_t - \bar{U}_t\|_F^2 \,, \tag{29}$$

where $c_1 = \frac{\delta}{2}$ and $c_2 = \frac{\delta}{8}$, the last equality holds due to $\bar{\bar{U}}_t W = \bar{\bar{U}}_t$, the second inequality holds due to $\|(\alpha_u(W - I) - I)\|_2 \leq \|\alpha_u(W - I)\|_2 + \|I\|_2 \leq 2\alpha_u + 1$, the third inequality holds due to Assumption 3.1, and the fourth inequality holds due to $\alpha_u \leq \frac{\delta}{16}$ and the following inequality:

$$(1 + c_1)(1 + 2\alpha_u)^2(1 - \delta)$$

$$\leq (1 + \frac{\delta}{2})(1 + \frac{\delta}{8})^2(1 - \delta)$$

$$\leq (1 - \frac{\delta}{2})(1 + \frac{\delta^2}{64} + \frac{\delta}{4})$$

$$= 1 + \frac{\delta^2}{64} + \frac{\delta}{4} - \frac{\delta}{2} - \frac{\delta}{2}\frac{\delta^2}{64} - \frac{\delta}{2}\frac{\delta}{4}$$

$$\leq 1 - \frac{\delta}{4} - \frac{\delta^3}{128} - \frac{3\delta^2}{64}$$

$$\leq 1 - \frac{\delta}{4} \,. \tag{30}$$

Similarly, we can prove the second inequality.

$\square$

**Lemma D.6.** *Given Assumptions 3.1-4.3, $\alpha_u \in (0, 1)$, and $\alpha_v \in (0, 1)$, we have*

$$\|U_{t+1} - U_t\|_F^2 \leq 51\|G_{t+1} - G_t\|_F^2 + 12\alpha_u^2\|U_t - \bar{U}_t\|_F^2 + 24\alpha_u^2\|U_t - \hat{U}_t\|_F^2 \,,$$

$$\|V_{t+1} - V_t\|_F^2 \leq 51\|H_{t+1} - H_t\|_F^2 + 12\alpha_v^2\|V_t - \bar{V}_t\|_F^2 + 24\alpha_v^2\|V_t - \hat{V}_t\|_F^2 \,. \tag{31}$$

*Proof.*

$$\|U_{t+1} - U_t\|_F^2$$

$$= \|\tilde{U}_{t+1} + \alpha_u \hat{U}_{t+1}(W - I) - U_t\|_F^2$$

$$= \|\tilde{U}_{t+1} + \alpha_u(\hat{U}_t + \mathcal{C}(\tilde{U}_{t+1} - \hat{U}_t))(W - I) - U_t\|_F^2$$

$$= \|\tilde{U}_{t+1} + \alpha_u \tilde{U}_{t+1}(W - I) + \alpha_u(\hat{U}_t - \tilde{U}_{t+1} + \mathcal{C}(\tilde{U}_{t+1} - \hat{U}_t))(W - I) - U_t\|_F^2$$

$$= \|\tilde{U}_{t+1}(I + \alpha_u(W - I)) + \alpha_u(\hat{U}_t - \tilde{U}_{t+1} + \mathcal{C}(\tilde{U}_{t+1} - \hat{U}_t))(W - I) - U_t\|_F^2$$

$$= \|(G_{t+1} - G_t)(I + \alpha_u(W - I)) + \alpha_u U_t(W - I) + \alpha_u(\hat{U}_t - \tilde{U}_{t+1} + \mathcal{C}(\tilde{U}_{t+1} - \hat{U}_t))(W - I)\|_F^2$$

$$= \|(G_{t+1} - G_t)(I + \alpha_u(W - I)) + \alpha_u(U_t - \bar{U}_t)(W - I) + \alpha_u(\hat{U}_t - \tilde{U}_{t+1} + \mathcal{C}(\tilde{U}_{t+1} - \hat{U}_t))(W - I)\|_F^2$$

$$\leq 3\|(G_{t+1} - G_t)(I + \alpha_u(W - I))\|_F^2 + 3\alpha_u^2\|(U_t - \bar{U}_t)(W - I)\|_F^2 + 3\alpha_u^2\|(\hat{U}_t - \tilde{U}_{t+1} + \mathcal{C}(\tilde{U}_{t+1} - \hat{U}_t))(W - I)\|_F^2$$

$$\leq 27\|G_{t+1} - G_t\|_F^2 + 12\alpha_u^2\|U_t - \bar{U}_t\|_F^2 + 12\alpha_u^2(1 - \delta)\|\tilde{U}_{t+1} - \hat{U}_t\|_F^2$$

$$\leq 27\|G_{t+1} - G_t\|_F^2 + 12\alpha_u^2\|U_t - \bar{U}_t\|_F^2 + 24\alpha_u^2\|U_t - \hat{U}_t\|_F^2 + 24\alpha_u^2\|G_{t+1} - G_t\|_F^2$$

$$\leq 51\|G_{t+1} - G_t\|_F^2 + 12\alpha_u^2\|U_t - \bar{U}_t\|_F^2 + 24\alpha_u^2\|U_t - \hat{U}_t\|_F^2 , \tag{32}$$

where the last equality holds due to $\bar{U}_t W = \bar{U}_t$, the second inequality holds due to $\|W - I\|_2 \leq 2$, $\|(\alpha_u(W - I) + I)\|_2 \leq \|\alpha_u(W - I)\|_2 + \|I\|_2 \leq 2\alpha_u + 1 \leq 3$, and Assumption 3.1, the third inequality holds due to $\delta < 1$, and the last inequality holds due to $\alpha_u < 1$. Similarly, we can prove the second inequality. $\square$

**Lemma D.7.** *Given Assumptions 3.1-4.3, $\alpha_x\eta \in (0, 1)$, and $\alpha_y\eta \in (0, 1)$, we have*

$$\|X_{t+1} - X_t\|_F^2 \leq 54\gamma_x^2\eta^2\|U_t - \bar{U}_t\|_F^2 + 54\gamma_x^2\eta^2\|\bar{U}_t\|_F^2 + 12\alpha_x^2\eta^2\|X_t - \bar{X}_t\|_F^2 + 12\alpha_x^2\eta^2\|\tilde{X}_{t+1} - \hat{X}_t\|_F^2 ,$$

$$\|Y_{t+1} - Y_t\|_F^2 \leq 54\gamma_y^2\eta^2\|V_t - \bar{V}_t\|_F^2 + 54\gamma_y^2\eta^2\|\bar{V}_t\|_F^2 + 12\alpha_y^2\eta^2\|Y_t - \bar{Y}_t\|_F^2 + 12\alpha_y^2\eta^2\|\tilde{Y}_{t+1} - \hat{Y}_t\|_F^2 . \tag{33}$$

*Proof.*

$$\|X_{t+1} - X_t\|_F^2$$

$$= \|\tilde{X}_{t+1} + \alpha_x\eta\hat{X}_{t+1}(W - I) - X_t\|_F^2$$

$$= \|\tilde{X}_{t+1} + \alpha_x\eta(\hat{X}_t + \mathcal{C}(\tilde{X}_{t+1} - \hat{X}_t))(W - I) - X_t\|_F^2$$

$$= \|\tilde{X}_{t+1} + \alpha_x\eta\tilde{X}_{t+1}(W - I) + \alpha_x\eta(\hat{X}_t - \tilde{X}_{t+1} + \mathcal{C}(\tilde{X}_{t+1} - \hat{X}_t))(W - I) - X_t\|_F^2$$

$$= \|\tilde{X}_{t+1}(I + \alpha_x\eta(W - I)) + \alpha_x\eta(\hat{X}_t - \tilde{X}_{t+1} + \mathcal{C}(\tilde{X}_{t+1} - \hat{X}_t))(W - I) - X_t\|_F^2$$

$$= \| - \gamma_x\eta U_t(I + \alpha_x\eta(W - I)) + \alpha_x\eta X_t(W - I) + \alpha_x\eta(\hat{X}_t - \tilde{X}_{t+1} + \mathcal{C}(\tilde{X}_{t+1} - \hat{X}_t))(W - I)\|_F^2$$

$$= \| - \gamma_x\eta U_t(I + \alpha_x\eta(W - I)) + \alpha_x\eta(X_t - \bar{X}_t)(W - I) + \alpha_x\eta(\hat{X}_t - \tilde{X}_{t+1} + \mathcal{C}(\tilde{X}_{t+1} - \hat{X}_t))(W - I)\|_F^2$$

$$\leq 3\gamma_x^2\eta^2\|U_t(I + \alpha_x\eta(W - I))\|_F^2 + 3\alpha_x^2\eta^2\|(X_t - \bar{X}_t)(W - I)\|_F^2$$
$$+ 3\alpha_x^2\eta^2\|(\hat{X}_t - \tilde{X}_{t+1} + \mathcal{C}(\tilde{X}_{t+1} - \hat{X}_t))(W - I)\|_F^2$$

$$\leq 27\gamma_x^2\eta^2\|U_t\|_F^2 + 12\alpha_x^2\eta^2\|X_t - \bar{X}_t\|_F^2 + 12\alpha_x^2\eta^2\|\tilde{X}_{t+1} - \hat{X}_t\|_F^2$$

$$\leq 54\gamma_x^2\eta^2\|U_t - \bar{U}_t\|_F^2 + 54\gamma_x^2\eta^2\|\bar{U}_t\|_F^2 + 12\alpha_x^2\eta^2\|X_t - \bar{X}_t\|_F^2 + 12\alpha_x^2\eta^2\|\tilde{X}_{t+1} - \hat{X}_t\|_F^2 , \tag{34}$$

where the last equality holds due to $\bar{X}_t W = \bar{X}_t$, the second inequality holds due to $\|W - I\|_2 \leq 2$, $\|\alpha_x\eta(W - I) + I\|_2 \leq \|\alpha_x\eta(W - I)\|_2 + \|I\|_2 \leq 2\alpha_x\eta + 1 \leq 3$, and Assumption 3.1. Similarly, we can prove the second inequality.

$\square$

**Lemma D.8.** *Given Assumptions 3.1-4.3, by setting $\gamma_y < \frac{1}{6L}$ and $\eta < 1$, we have*

$$\|\bar{y}_{t+1} - y^*(\bar{x}_{t+1})\|^2 \leq (1 - \frac{\gamma_y\eta\mu}{4})\|\bar{y}_t - y^*(\bar{x}_t)\|^2 - \frac{3\eta\gamma_y^2}{4}\|\bar{h}_t\|^2 + \frac{25\eta\gamma_x^2 L^2}{6\gamma_y\mu^3}\|\bar{g}_t\|^2$$

$$+ \frac{25\gamma_y\eta}{3\mu}\|\frac{1}{K}\sum_{k=1}^K \nabla_y f^{(k)}(x_t^{(k)}, y_t^{(k)}) - \frac{1}{K}\sum_{k=1}^K h_t^{(k)}\|^2$$

$$+ \frac{25\gamma_y\eta L^2}{3\mu}\frac{1}{K}\|\bar{X}_t - X_t\|_F^2 + \frac{25\gamma_y\eta L^2}{3\mu}\frac{1}{K}\|\bar{Y}_t - Y_t\|_F^2 . \tag{35}$$

*Proof.* By setting $\gamma_y < \frac{1}{6L}$ and $\eta < 1$, we can get

$$\|\bar{y}_{t+1} - y^*(\bar{x}_{t+1})\|^2$$

$$\leq (1 - \frac{\gamma_y \eta \mu}{4})\|\bar{y}_t - y^*(\bar{x}_t)\|^2 - \frac{3\eta\gamma_y^2}{4}\|\bar{h}_t\|^2 + \frac{25\gamma_y\eta}{6\mu}\|\nabla_y f(\bar{x}_t, \bar{y}_t) - \bar{h}_t\|^2 + \frac{25\eta\gamma_x^2 L^2}{6\gamma_y\mu^3}\|\bar{g}_t\|^2$$

$$\leq (1 - \frac{\gamma_y \eta \mu}{4})\|\bar{y}_t - y^*(\bar{x}_t)\|^2 - \frac{3\eta\gamma_y^2}{4}\|\bar{h}_t\|^2 + \frac{25\eta\gamma_x^2 L^2}{6\gamma_y\mu^3}\|\bar{g}_t\|^2$$

$$+ \frac{25\gamma_y\eta}{3\mu}\|\frac{1}{K}\sum_{k=1}^{K}\nabla_y f^{(k)}(\bar{x}_t, \bar{y}_t) - \frac{1}{K}\sum_{k=1}^{K}\nabla_y f^{(k)}(x_t^{(k)}, y_t^{(k)})\|^2 + \frac{25\gamma_y\eta}{3\mu}\|\frac{1}{K}\sum_{k=1}^{K}\nabla_y f^{(k)}(x_t^{(k)}, y_t^{(k)}) - \frac{1}{K}\sum_{k=1}^{K}h_t^{(k)}\|^2$$

$$\leq (1 - \frac{\gamma_y \eta \mu}{4})\|\bar{y}_t - y^*(\bar{x}_t)\|^2 - \frac{3\eta\gamma_y^2}{4}\|\bar{h}_t\|^2 + \frac{25\eta\gamma_x^2 L^2}{6\gamma_y\mu^3}\|\bar{g}_t\|^2 + \frac{25\gamma_y\eta}{3\mu}\|\frac{1}{K}\sum_{k=1}^{K}\nabla_y f^{(k)}(x_t^{(k)}, y_t^{(k)}) - \frac{1}{K}\sum_{k=1}^{K}h_t^{(k)}\|^2$$

$$+ \frac{25\gamma_y\eta L^2}{3\mu}\frac{1}{K}\|\bar{X}_t - X_t\|_F^2 + \frac{25\gamma_y\eta L^2}{3\mu}\frac{1}{K}\|\bar{Y}_t - Y_t\|_F^2 , \tag{36}$$

where the first step holds due to Lemma 28 in (Huang et al., 2022), and the last step holds due to Assumption 4.1.

$\square$

**Lemma D.9.** *Given Assumptions 3.1-4.3, we have*

$$\mathbb{E}[\|G_{t+1} - G_t\|_F^2] \leq 3\rho_x^2\eta^4 \sum_{k=1}^{K}\mathbb{E}[\|\nabla_x f^{(k)}(x_t^{(k)}, y_t^{(k)}) - g_t^{(k)}\|^2] + 3L^2\mathbb{E}[\|X_{t+1} - X_t\|_F^2]$$

$$+ 3L^2\mathbb{E}[\|Y_{t+1} - Y_t\|_F^2] + 3\rho_x^2\eta^4\sigma^2 K ,$$

$$\mathbb{E}[\|H_{t+1} - H_t\|_F^2] \leq 3\rho_y^2\eta^4 \sum_{k=1}^{K}\mathbb{E}[\|\nabla_y f^{(k)}(x_t^{(k)}, y_t^{(k)}) - h_t^{(k)}\|^2] + 3L^2\mathbb{E}[\|X_{t+1} - X_t\|_F^2]$$

$$+ 3L^2\mathbb{E}[\|Y_{t+1} - Y_t\|_F^2] + 3\rho_y^2\eta^4\sigma^2 K . \tag{37}$$

*Proof.*

$$\mathbb{E}[\|G_{t+1} - G_t\|_F^2] = \sum_{k=1}^{K}\mathbb{E}[\|g_{t+1}^{(k)} - g_t^{(k)}\|^2]$$

$$= \sum_{k=1}^{K}\mathbb{E}[\|(1 - \rho_x\eta^2)(g_t^{(k)} - \nabla_x f^{(k)}(x_t^{(k)}, y_t^{(k)}; \xi_{t+1}^{(k)})) + \nabla_x f^{(k)}(x_{t+1}^{(k)}, y_{t+1}^{(k)}; \xi_{t+1}^{(k)}) - g_t^{(k)}\|^2]$$

$$\leq 3\sum_{k=1}^{K}\mathbb{E}[\| - \rho_x\eta^2 g_t^{(k)} + \rho_x\eta^2\nabla_x f^{(k)}(x_t^{(k)}, y_t^{(k)})\|^2]$$

$$+ 3\sum_{k=1}^{K}\mathbb{E}[\| - \rho_x\eta^2\nabla_x f^{(k)}(x_t^{(k)}, y_t^{(k)}) + \rho_x\eta^2\nabla_x f^{(k)}(x_t^{(k)}, y_t^{(k)}; \xi_{t+1}^{(k)})\|^2]$$

$$+ 3\sum_{k=1}^{K}\mathbb{E}[\| - \nabla_x f^{(k)}(x_t^{(k)}, y_t^{(k)}; \xi_{t+1}^{(k)}) + \nabla_x f^{(k)}(x_{t+1}^{(k)}, y_{t+1}^{(k)}; \xi_{t+1}^{(k)})\|^2]$$

$$\leq 3\rho_x^2\eta^4 \sum_{k=1}^{K}\mathbb{E}[\|\nabla_x f^{(k)}(x_t^{(k)}, y_t^{(k)}) - g_t^{(k)}\|^2] + 3\rho_x^2\eta^4\sigma^2 K + 3L^2\mathbb{E}[\|X_{t+1} - X_t\|_F^2] + 3L^2\mathbb{E}[\|Y_{t+1} - Y_t\|_F^2] , \tag{38}$$

where the last step holds due to Assumptions 4.1, 4.2.
$\square$

**Lemma D.10.** *Given Assumptions 3.1-4.3, we have*

$$\mathbb{E}\left[\left\|\frac{1}{K}\sum_{k=1}^{K}g_{t+1}^{(k)} - \frac{1}{K}\sum_{k=1}^{K}\nabla_x f^{(k)}(x_{t+1}^{(k)}, y_{t+1}^{(k)})\right\|^2\right] \leq (1 - \rho_x\eta^2)\mathbb{E}\left[\left\|\frac{1}{K}\sum_{k=1}^{K}g_t^{(k)} - \frac{1}{K}\sum_{k=1}^{K}\nabla_x f^{(k)}(x_t^{(k)}, y_t^{(k)})\right\|^2\right]$$

$$+ \frac{2L^2}{K^2}\mathbb{E}\Big[\big\|X_{t+1} - X_t\big\|_F^2\Big] + \frac{2L^2}{K^2}\mathbb{E}\Big[\big\|Y_{t+1} - Y_t\big\|_F^2\Big] + \frac{2\rho_x^2\eta^4\sigma^2}{K} \ ,$$

$$\mathbb{E}\Big[\Big\|\frac{1}{K}\sum_{k=1}^{K}h_{t+1}^{(k)} - \frac{1}{K}\sum_{k=1}^{K}\nabla_y f^{(k)}(x_{t+1}^{(k)}, y_{t+1}^{(k)})\Big\|^2\Big] \leq (1-\rho_y\eta^2)\mathbb{E}\Big[\Big\|\frac{1}{K}\sum_{k=1}^{K}h_t^{(k)} - \frac{1}{K}\sum_{k=1}^{K}\nabla_y f^{(k)}(x_t^{(k)}, y_t^{(k)})\Big\|^2\Big]$$

$$+ \frac{2L^2}{K^2}\mathbb{E}\Big[\big\|X_{t+1} - X_t\big\|_F^2\Big] + \frac{2L^2}{K^2}\mathbb{E}\Big[\big\|Y_{t+1} - Y_t\big\|_F^2\Big] + \frac{2\rho_y^2\eta^4\sigma^2}{K} \ . \tag{39}$$

$$\frac{1}{K}\sum_{k=1}^{K}\mathbb{E}\Big[\big\|g_{t+1}^{(k)} - \nabla_x f^{(k)}(x_{t+1}^{(k)}, y_{t+1}^{(k)})\big\|^2\Big] \leq (1-\rho_x\eta^2)\frac{1}{K}\sum_{k=1}^{K}\mathbb{E}\Big[\big\|g_t^{(k)} - \nabla_x f^{(k)}(x_t^{(k)}, y_t^{(k)})\big\|^2\Big]$$

$$+ \frac{2L^2}{K}\mathbb{E}\Big[\big\|X_{t+1} - X_t\big\|_F^2\Big] + \frac{2L^2}{K}\mathbb{E}\Big[\big\|Y_{t+1} - Y_t\big\|_F^2\Big] + 2\rho_x^2\eta^4\sigma^2 \ ,$$

$$\frac{1}{K}\sum_{k=1}^{K}\mathbb{E}\Big[\big\|h_{t+1}^{(k)} - \nabla_y f^{(k)}(x_{t+1}^{(k)}, y_{t+1}^{(k)})\big\|^2\Big] \leq (1-\rho_y\eta^2)\frac{1}{K}\sum_{k=1}^{K}\mathbb{E}\Big[\big\|h_t^{(k)} - \nabla_y f^{(k)}(x_t^{(k)}, y_t^{(k)})\big\|^2\Big]$$

$$+ \frac{2L^2}{K}\mathbb{E}\Big[\big\|X_{t+1} - X_t\big\|_F^2\Big] + \frac{2L^2}{K}\mathbb{E}\Big[\big\|Y_{t+1} - Y_t\big\|_F^2\Big] + 2\rho_y^2\eta^4\sigma^2 \ . \tag{40}$$

*Proof.* For the first inequality, we can prove it as follows:

$$\mathbb{E}\Big[\Big\|\frac{1}{K}\sum_{k=1}^{K}g_{t+1}^{(k)} - \frac{1}{K}\sum_{k=1}^{K}\nabla_x f^{(k)}(x_{t+1}^{(k)}, y_{t+1}^{(k)})\Big\|^2\Big]$$

$$= \mathbb{E}\Big[\Big\|\frac{1}{K}\sum_{k=1}^{K}\big((1-\rho_x\eta^2)(g_t^{(k)} - \nabla_x f^{(k)}(x_t^{(k)}, y_t^{(k)};\xi_{t+1}^{(k)})) + \nabla_x f^{(k)}(x_{t+1}^{(k)}, y_{t+1}^{(k)};\xi_{t+1}^{(k)})\big) - \frac{1}{K}\sum_{k=1}^{K}\nabla_x f^{(k)}(x_{t+1}^{(k)}, y_{t+1}^{(k)})\Big\|^2\Big]$$

$$= (1-\rho_x\eta^2)^2\mathbb{E}\Big[\Big\|\frac{1}{K}\sum_{k=1}^{K}(g_t^{(k)} - \nabla_x f^{(k)}(x_t^{(k)}, y_t^{(k)}))\Big\|^2\Big]$$

$$+ \mathbb{E}\Big[\Big\|(1-\rho_x\eta^2)\frac{1}{K}\sum_{k=1}^{K}\Big(\nabla_x f^{(k)}(x_t^{(k)}, y_t^{(k)}) - \nabla_x f^{(k)}(x_t^{(k)}, y_t^{(k)};\xi_{t+1}^{(k)}) + \nabla_x f^{(k)}(x_{t+1}^{(k)}, y_{t+1}^{(k)};\xi_{t+1}^{(k)})$$

$$- \nabla_x f^{(k)}(x_{t+1}^{(k)}, y_{t+1}^{(k)})\Big) + \rho_x\eta^2\frac{1}{K}\sum_{k=1}^{K}(\nabla_x f^{(k)}(x_{t+1}^{(k)}, y_{t+1}^{(k)};\xi_{t+1}^{(k)}) - \nabla_x f^{(k)}(x_{t+1}^{(k)}, y_{t+1}^{(k)}))\Big\|^2\Big]$$

$$\leq (1-\rho_x\eta^2)^2\mathbb{E}\Big[\Big\|\frac{1}{K}\sum_{k=1}^{K}((g_t^{(k)} - \nabla_x f^{(k)}(x_t^{(k)}, y_t^{(k)})))\Big\|^2\Big]$$

$$+ 2(1-\rho_x\eta^2)^2\mathbb{E}\Big[\Big\|\frac{1}{K}\sum_{k=1}^{K}(\nabla_x f^{(k)}(x_t^{(k)}, y_t^{(k)}) - \nabla_x f^{(k)}(x_t^{(k)}, y_t^{(k)};\xi_{t+1}^{(k)})$$

$$+ \nabla_x f^{(k)}(x_{t+1}^{(k)}, y_{t+1}^{(k)};\xi_{t+1}^{(k)}) - \nabla_x f^{(k)}(x_{t+1}^{(k)}, y_{t+1}^{(k)}))\Big\|^2\Big]$$

$$+ 2\rho_x^2\eta^4\mathbb{E}\Big\|\frac{1}{K}\sum_{k=1}^{K}(\nabla_x f^{(k)}(x_{t+1}^{(k)}, y_{t+1}^{(k)};\xi_{t+1}^{(k)}) - \nabla_x f^{(k)}(x_{t+1}^{(k)}, y_{t+1}^{(k)}))\Big\|^2\Big]$$

$$\leq (1-\rho_x\eta^2)\mathbb{E}\Big[\Big\|\frac{1}{K}\sum_{k=1}^{K}(g_t^{(k)} - \nabla_x f^{(k)}(x_t^{(k)}, y_t^{(k)}))\Big\|^2\Big]$$

$$+ 2\frac{1}{K^2}\sum_{k=1}^{K}\mathbb{E}\Big[\big\|\nabla_x f^{(k)}(x_{t+1}^{(k)}, y_{t+1}^{(k)};\xi_{t+1}^{(k)}) - \nabla_x f^{(k)}(x_t^{(k)}, y_t^{(k)};\xi_{t+1}^{(k)})\big\|^2\Big]$$

$$+ 2\frac{\rho_x^2\eta^4}{K^2}\sum_{k=1}^{K}\mathbb{E}\big\|\nabla_x f^{(k)}(x_{t+1}^{(k)}, y_{t+1}^{(k)};\xi_{t+1}^{(k)}) - \nabla_x f^{(k)}(x_{t+1}^{(k)}, y_{t+1}^{(k)})\big\|^2\Big]$$

$$\leq (1 - \rho_x \eta^2) \mathbb{E}\Big[\Big\|\frac{1}{K}\sum_{k=1}^{K}(g_t^{(k)} - \nabla_x f^{(k)}(x_t^{(k)}, y_t^{(k)}))\Big\|^2\Big] + \frac{2\rho_x^2 \eta^4 \sigma^2}{K}$$

$$+ 2\frac{1}{K^2}L^2 \sum_{k=1}^{K}\mathbb{E}\Big[\Big\|x_{t+1}^{(k)} - x_t^{(k)}\Big\|^2 + \Big\|y_{t+1}^{(k)} - y_t^{(k)}\Big\|^2\Big]$$

$$= (1 - \rho_x \eta^2)\mathbb{E}\Big[\Big\|\frac{1}{K}\sum_{k=1}^{K}(g_t^{(k)} - \nabla_x f^{(k)}(x_t^{(k)}, y_t^{(k)}))\Big\|^2\Big]$$

$$+ \frac{2L^2}{K^2}\mathbb{E}\Big[\Big\|X_{t+1} - X_t\Big\|_F^2\Big] + \frac{2L^2}{K^2}\mathbb{E}\Big[\Big\|Y_{t+1} - Y_t\Big\|_F^2\Big] + \frac{2\rho_x^2 \eta^4 \sigma^2}{K}\,, \tag{41}$$

where the second step holds due to $\mathbb{E}[(1 - \rho_x\eta^2)\frac{1}{K}\sum_{k=1}^{K}(\nabla_x f^{(k)}(x_t^{(k)}, y_t^{(k)}) - \nabla_x f^{(k)}(x_t^{(k)}, y_t^{(k)}; \xi_{t+1}^{(k)}) + \nabla_x f^{(k)}(x_{t+1}^{(k)}, y_{t+1}^{(k)}; \xi_{t+1}^{(k)}) - \nabla_x f^{(k)}(x_{t+1}^{(k)}, y_{t+1}^{(k)})) + \rho_x\eta^2 \frac{1}{K}\sum_{k=1}^{K}(\nabla_x f^{(k)}(x_{t+1}^{(k)}, y_{t+1}^{(k)}; \xi_{t+1}^{(k)}) - \nabla_x f^{(k)}(x_{t+1}^{(k)}, y_{t+1}^{(k)}))] = 0$, and the second to last step holds due to Assumptions 4.1, 4.2.

The second inequality can be proved in the exactly same approach so that we do not include its detailed proof.

For the third inequality, we can prove it as follows:

$$\frac{1}{K}\sum_{k=1}^{K}\mathbb{E}\Big[\Big\|g_{t+1}^{(k)} - \nabla_x f^{(k)}(x_{t+1}^{(k)}, y_{t+1}^{(k)})\Big\|^2\Big]$$

$$= \frac{1}{K}\sum_{k=1}^{K}\mathbb{E}\Big[\Big\|(1 - \rho_x\eta^2)(g_t^{(k)} - \nabla_x f^{(k)}(x_t^{(k)}, y_t^{(k)}; \xi_{t+1}^{(k)})) + \nabla_x f^{(k)}(x_{t+1}^{(k)}, y_{t+1}^{(k)}; \xi_{t+1}^{(k)}) - \nabla_x f^{(k)}(x_{t+1}^{(k)}, y_{t+1}^{(k)})\Big\|^2\Big]$$

$$= \frac{1}{K}\sum_{k=1}^{K}\mathbb{E}\Big[\Big\|(1 - \rho_x\eta^2)(g_t^{(k)} - \nabla_x f^{(k)}(x_t^{(k)}, y_t^{(k)}))\Big\|^2\Big]$$

$$+ \frac{1}{K}\sum_{k=1}^{K}\mathbb{E}\Big[\Big\|(1 - \rho_x\eta^2)\Big(\nabla_x f^{(k)}(x_{t+1}^{(k)}, y_{t+1}^{(k)}; \xi_{t+1}^{(k)}) - \nabla_x f^{(k)}(x_{t+1}^{(k)}, y_{t+1}^{(k)})$$

$$- \nabla_x f^{(k)}(x_t^{(k)}, y_t^{(k)}; \xi_{t+1}^{(k)}) + \nabla_x f^{(k)}(x_t^{(k)}, y_t^{(k)})\Big) + \rho_x\eta^2(\nabla_x f^{(k)}(x_{t+1}^{(k)}, y_{t+1}^{(k)}; \xi_{t+1}^{(k)}) - \nabla_x f^{(k)}(x_{t+1}^{(k)}, y_{t+1}^{(k)}))\Big\|^2\Big]$$

$$\leq (1 - \rho_x\eta^2)^2 \frac{1}{K}\sum_{k=1}^{K}\mathbb{E}\Big[\Big\|g_t^{(k)} - \nabla_x f^{(k)}(x_t^{(k)}, y_t^{(k)})\Big\|^2\Big]$$

$$+ \frac{2}{K}\sum_{k=1}^{K}\mathbb{E}\Big[\Big\|\nabla_x f^{(k)}(x_{t+1}^{(k)}, y_{t+1}^{(k)}; \xi_{t+1}^{(k)}) - \nabla_x f^{(k)}(x_t^{(k)}, y_t^{(k)}; \xi_{t+1}^{(k)})\Big\|^2\Big]$$

$$+ \frac{2\rho_x^2\eta^4}{K}\sum_{k=1}^{K}\mathbb{E}\Big[\Big\|\nabla_x f^{(k)}(x_{t+1}^{(k)}, y_{t+1}^{(k)}; \xi_{t+1}^{(k)}) - \nabla_x f^{(k)}(x_{t+1}^{(k)}, y_{t+1}^{(k)})\Big\|^2\Big]$$

$$\leq (1 - \rho_x\eta^2)\frac{1}{K}\sum_{k=1}^{K}\mathbb{E}\Big[\Big\|g_t^{(k)} - \nabla_x f^{(k)}(x_t^{(k)}, y_t^{(k)})\Big\|^2\Big] + \frac{2L^2}{K}\sum_{k=1}^{K}\mathbb{E}\Big[\Big\|x_{t+1}^{(k)} - x_t^{(k)}\Big\|^2 + \Big\|y_{t+1}^{(k)} - y_t^{(k)}\Big\|^2\Big] + 2\rho_x^2\eta^4\sigma^2$$

$$= (1 - \rho_x\eta^2)\frac{1}{K}\sum_{k=1}^{K}\mathbb{E}\Big[\Big\|g_t^{(k)} - \nabla_x f^{(k)}(x_t^{(k)}, y_t^{(k)})\Big\|^2\Big] + \frac{2L^2}{K}\mathbb{E}\Big[\Big\|X_{t+1} - X_t\Big\|_F^2\Big] + \frac{2L^2}{K}\mathbb{E}\Big[\Big\|Y_{t+1} - Y_t\Big\|_F^2\Big] + 2\rho_x^2\eta^4\sigma^2\,, \tag{42}$$

where the second step holds due to $\mathbb{E}[(1 - \rho_x\eta^2)(\nabla_x f^{(k)}(x_{t+1}^{(k)}, y_{t+1}^{(k)}; \xi_{t+1}^{(k)}) - \nabla_x f^{(k)}(x_{t+1}^{(k)}, y_{t+1}^{(k)}) - \nabla_x f^{(k)}(x_t^{(k)}, y_t^{(k)}; \xi_{t+1}^{(k)}) + \nabla_x f^{(k)}(x_t^{(k)}, y_t^{(k)})) + \rho_x\eta^2(\nabla_x f^{(k)}(x_{t+1}^{(k)}, y_{t+1}^{(k)}; \xi_{t+1}^{(k)}) - \nabla_x f^{(k)}(x_{t+1}^{(k)}, y_{t+1}^{(k)}))] = 0$, and the second to last step holds due to Assumptions 4.1, 4.2. $\qquad\square$

### D.2. Proof of Theorem 4.4

*Proof.* At first, we have

$$F(\bar{x}_{t+1}) \le F(\bar{x}_t) + \langle \nabla F(\bar{x}_t), \bar{x}_{t+1} - \bar{x}_t \rangle + \frac{L_F}{2} \|\bar{x}_{t+1} - \bar{x}_t\|^2$$

$$= F(\bar{x}_t) - \gamma_x \eta \langle \nabla F(\bar{x}_t), \bar{g}_t \rangle + \frac{\gamma_x^2 \eta^2 L_F}{2} \|\bar{g}_t\|^2$$

$$= F(\bar{x}_t) - \frac{\gamma_x \eta}{2} \|\nabla F(\bar{x}_t)\|^2 + \Big(\frac{\gamma_x^2 \eta^2 L_F}{2} - \frac{\gamma_x \eta}{2}\Big) \|\bar{g}_t\|^2 + \frac{\gamma_x \eta}{2} \|\nabla F(\bar{x}_t) - \bar{g}_t\|^2$$

$$\le F(\bar{x}_t) - \frac{\gamma_x \eta}{2} \|\nabla F(\bar{x}_t)\|^2 + \Big(\frac{\gamma_x^2 \eta^2 L_F}{2} - \frac{\gamma_x \eta}{2}\Big) \|\bar{g}_t\|^2 + 2\gamma_x \eta \|\nabla F(\bar{x}_t) - \frac{1}{K}\sum_{k=1}^{K} \nabla_x f^{(k)}(\bar{x}_t, \bar{y}_t)\|^2$$

$$+ 2\gamma_x \eta \| \frac{1}{K}\sum_{k=1}^{K} \nabla_x f^{(k)}(\bar{x}_t, \bar{y}_t) - \frac{1}{K}\sum_{k=1}^{K} \nabla_x f^{(k)}(x_t^{(k)}, y_t^{(k)})\|^2 + 2\gamma_x \eta \| \frac{1}{K}\sum_{k=1}^{K} \nabla_x f^{(k)}(x_t^{(k)}, y_t^{(k)}) - \frac{1}{K}\sum_{k=1}^{K} g_t^{(k)}\|^2$$

$$\le F(\bar{x}_t) - \frac{\gamma_x \eta}{2} \|\nabla F(\bar{x}_t)\|^2 - \frac{\gamma_x \eta}{4} \|\bar{g}_t\|^2 + 2\gamma_x \eta L^2 \|y^*(\bar{x}_t) - \bar{y}_t\|^2$$

$$+ 2\gamma_x \eta L^2 \frac{1}{K} \|\bar{X}_t - X_t\|_F^2 + 2\gamma_x \eta L^2 \frac{1}{K} \|\bar{Y}_t - Y_t\|_F^2 + 2\gamma_x \eta \| \frac{1}{K}\sum_{k=1}^{K} \nabla_x f^{(k)}(x_t^{(k)}, y_t^{(k)}) - \frac{1}{K}\sum_{k=1}^{K} g_t^{(k)}\|^2 , \tag{43}$$

where the last step holds due to $\eta \le \frac{1}{2\gamma_x L_F}$ and Assumption 4.1.

Given the potential function in Eq. (17), based on Lemmas D.8, D.10, D.4, D.5, D.2, D.3, we can obtain

$$P_{t+1} - P_t$$

$$\le -\frac{\gamma_x \eta}{2} \mathbb{E}\Big[\|\nabla F(\bar{x}_t)\|^2\Big] + (2\gamma_x \eta L^2 - c_0 \frac{\gamma_y \eta \mu}{4}) \mathbb{E}\Big[\|y^*(\bar{x}_t) - \bar{y}_t\|^2\Big]$$

$$+ (-\frac{\gamma_x \eta}{4} + c_0 \frac{25\eta \gamma_x^2 L^2}{6\gamma_y \mu^3}) \mathbb{E}\Big[\|\bar{g}_t\|^2\Big] - c_0 \frac{3\eta \gamma_y^2}{4} \mathbb{E}\Big[\|\bar{h}_t\|^2\Big]$$

$$+ (2\gamma_x \eta L^2 + c_0 \frac{25\gamma_y \eta L^2}{3\mu} - c_3 \frac{\alpha_x \eta(1-\lambda)}{2}) \frac{1}{K} \mathbb{E}\Big[\|\bar{X}_t - X_t\|_F^2\Big]$$

$$+ (2\gamma_x \eta L^2 + c_0 \frac{25\gamma_y \eta L^2}{3\mu} - c_4 \frac{\alpha_y \eta(1-\lambda)}{2}) \frac{1}{K} \mathbb{E}\Big[\|\bar{Y}_t - Y_t\|_F^2\Big]$$

$$+ (c_3 \frac{3\gamma_x^2 \eta}{\alpha_x(1-\lambda)} - c_5 \frac{\alpha_u(1-\lambda)}{2} + c_7 \frac{36\alpha_u^2}{\delta}) \frac{1}{K} \mathbb{E}\Big[\|U_t - \bar{U}_t\|_F^2\Big]$$

$$+ (c_4 \frac{3\gamma_y^2 \eta}{\alpha_y(1-\lambda)} - c_6 \frac{\alpha_v(1-\lambda)}{2} + c_8 \frac{36\alpha_v^2}{\delta}) \frac{1}{K} \mathbb{E}\Big[\|V_t - \bar{V}_t\|_F^2\Big]$$

$$+ (c_3 \frac{4\alpha_x \eta(1-\delta)}{1-\lambda} - c_9 \frac{\delta}{2}) \frac{1}{K} \mathbb{E}\Big[\|\tilde{X}_{t+1} - \hat{X}_t\|_F^2\Big] + (c_4 \frac{4\alpha_y \eta(1-\delta)}{1-\lambda} - c_{10} \frac{\delta}{2}) \frac{1}{K} \mathbb{E}\Big[\|\tilde{Y}_{t+1} - \hat{Y}_t\|_F^2\Big]$$

$$+ (c_5 \frac{8\alpha_u(1-\delta)}{1-\lambda} - c_7 \frac{\delta}{8}) \frac{1}{K} \mathbb{E}\Big[\|U_t - \hat{U}_t\|_F^2\Big] + (c_6 \frac{8\alpha_v(1-\delta)}{1-\lambda} - c_8 \frac{\delta}{8}) \frac{1}{K} \mathbb{E}\Big[\|V_t - \hat{V}_t\|_F^2\Big]$$

$$+ (2\gamma_x \eta - c_1 \rho_x \eta^2) \mathbb{E}\Big[\|\frac{1}{K}\sum_{k=1}^{K} \nabla_x f^{(k)}(x_t^{(k)}, y_t^{(k)}) - \frac{1}{K}\sum_{k=1}^{K} g_t^{(k)}\|^2\Big]$$

$$+ (c_0 \frac{25\gamma_y \eta}{3\mu} - c_2 \rho_y \eta^2) \mathbb{E}\Big[\|\frac{1}{K}\sum_{k=1}^{K} \nabla_y f^{(k)}(x_t^{(k)}, y_t^{(k)}) - \frac{1}{K}\sum_{k=1}^{K} h_t^{(k)}\|^2\Big]$$

$$+ (-c_{11} \rho_x \eta^2) \frac{1}{K} \sum_{k=1}^{K} \mathbb{E}\Big[\Big\|g_t^{(k)} - \nabla_x f^{(k)}(x_t^{(k)}, y_t^{(k)})\Big\|^2\Big] + (-c_{12} \rho_y \eta^2) \frac{1}{K} \sum_{k=1}^{K} \mathbb{E}\Big[\Big\|h_t^{(k)} - \nabla_y f^{(k)}(x_t^{(k)}, y_t^{(k)})\Big\|^2\Big]$$

$$+ (c_1 \frac{2L^2}{K} + c_2 \frac{2L^2}{K} + c_9 \frac{6}{\delta} + c_{11} 2L^2 + c_{12} 2L^2) \frac{1}{K} \mathbb{E}[\|X_{t+1} - X_t\|_F^2]$$

$$+ (c_1 \frac{2L^2}{K} + c_2 \frac{2L^2}{K} + c_{10} \frac{6}{\delta} + c_{11} 2L^2 + c_{12} 2L^2) \frac{1}{K} \mathbb{E}[\|Y_{t+1} - Y_t\|_F^2]$$

$$+ (c_5 \frac{11}{\alpha_u(1-\lambda)} + c_7 \frac{46}{\delta}) \frac{1}{K} \mathbb{E}\Big[\|G_{t+1} - G_t\|_F^2\Big] + (c_6 \frac{11}{\alpha_v(1-\lambda)} + c_8 \frac{46}{\delta}) \frac{1}{K} \mathbb{E}\Big[\|H_{t+1} - H_t\|_F^2\Big]$$

$$+ c_9 \frac{6\gamma_x^2 \eta^2}{\delta} \frac{1}{K} \mathbb{E}\Big[\|U_{t+1} - U_t\|_F^2\Big] + c_{10} \frac{6\gamma_y^2 \eta^2}{\delta} \frac{1}{K} \mathbb{E}\Big[\|V_{t+1} - V_t\|_F^2\Big]$$

$$+ c_1 \frac{2\rho_x^2 \eta^4 \sigma^2}{K} + c_2 \frac{2\rho_y^2 \eta^4 \sigma^2}{K} + c_{11} 2\rho_x^2 \eta^4 \sigma^2 + c_{12} 2\rho_y^2 \eta^4 \sigma^2 . \tag{44}$$

By expanding $\mathbb{E}[\|U_{t+1} - U_t\|_F^2]$ and $\mathbb{E}[\|V_{t+1} - V_t\|_F^2]$ based on Lemma D.6, $\mathbb{E}[\|G_{t+1} - G_t\|_F^2]$ and $\mathbb{E}[\|H_{t+1} - H_t\|_F^2]$ based on Lemma D.9, and $\mathbb{E}[\|X_{t+1} - X_t\|_F^2]$ and $\mathbb{E}[\|Y_{t+1} - Y_t\|_F^2]$ based on Lemma D.7, we can obtain

$$
\begin{aligned}
&P_{t+1} - P_t \\
&\leq -\frac{\gamma_x \eta}{2} \mathbb{E}\Big[\|\nabla F(\bar{x}_t)\|^2\Big] + (2\gamma_x \eta L^2 - c_0 \frac{\gamma_y \eta \mu}{4}) \mathbb{E}\Big[\|y^*(\bar{x}_t) - \bar{y}_t\|^2\Big] \\
&\quad + (-\frac{\gamma_x \eta}{4} + c_0 \frac{25\eta \gamma_x^2 L^2}{6\gamma_y \mu^3} + C_5 \frac{9\gamma_x^2}{2\alpha_x^2}) \mathbb{E}\Big[\|\bar{g}_t\|^2\Big] + (-c_0 \frac{3\eta \gamma_y^2}{4} + C_6 \frac{9\gamma_y^2}{2\alpha_y^2}) \mathbb{E}\Big[\|\bar{h}_t\|^2\Big] \\
&\quad + (2\gamma_x \eta L^2 + c_0 \frac{25\gamma_y \eta L^2}{3\mu} - c_3 \frac{\alpha_x \eta(1-\lambda)}{2} + C_5) \frac{1}{K} \mathbb{E}\Big[\|\bar{X}_t - X_t\|_F^2\Big] \\
&\quad + (2\gamma_x \eta L^2 + c_0 \frac{25\gamma_y \eta L^2}{3\mu} - c_4 \frac{\alpha_y \eta(1-\lambda)}{2} + C_6) \frac{1}{K} \mathbb{E}\Big[\|\bar{Y}_t - Y_t\|_F^2\Big] \\
&\quad + (c_3 \frac{3\gamma_x^2 \eta}{\alpha_x(1-\lambda)} - c_5 \frac{\alpha_u(1-\lambda)}{2} + c_7 \frac{36\alpha_u^2}{\delta} + c_9 \frac{72\gamma_x^2 \alpha_u^2 \eta^2}{\delta} + C_5 \frac{9\gamma_x^2}{2\alpha_x^2}) \frac{1}{K} \mathbb{E}\Big[\|U_t - \bar{U}_t\|_F^2\Big] \\
&\quad + (c_4 \frac{3\gamma_y^2 \eta}{\alpha_y(1-\lambda)} - c_6 \frac{\alpha_v(1-\lambda)}{2} + c_8 \frac{36\alpha_v^2}{\delta} + c_{10} \frac{72\gamma_y^2 \alpha_v^2 \eta^2}{\delta} + C_6 \frac{9\gamma_y^2}{2\alpha_y^2}) \frac{1}{K} \mathbb{E}\Big[\|V_t - \bar{V}_t\|_F^2\Big] \\
&\quad + (c_3 \frac{4\alpha_x \eta(1-\delta)}{1-\lambda} - c_9 \frac{\delta}{2} + C_5) \frac{1}{K} \mathbb{E}\Big[\|\tilde{X}_{t+1} - \hat{X}_t\|_F^2\Big] \\
&\quad + (c_4 \frac{4\alpha_y \eta(1-\delta)}{1-\lambda} - c_{10} \frac{\delta}{2} + C_6) \frac{1}{K} \mathbb{E}\Big[\|\tilde{Y}_{t+1} - \hat{Y}_t\|_F^2\Big] \\
&\quad + (c_5 \frac{8\alpha_u(1-\delta)}{1-\lambda} - c_7 \frac{\delta}{8} + c_9 \frac{144\gamma_x^2 \alpha_u^2 \eta^2}{\delta}) \frac{1}{K} \mathbb{E}\Big[\|U_t - \hat{U}_t\|_F^2\Big] \\
&\quad + (c_6 \frac{8\alpha_v(1-\delta)}{1-\lambda} - c_8 \frac{\delta}{8} + c_{10} \frac{144\gamma_y^2 \alpha_v^2 \eta^2}{\delta}) \frac{1}{K} \mathbb{E}\Big[\|V_t - \hat{V}_t\|_F^2\Big] \\
&\quad + (2\gamma_x \eta - c_1 \rho_x \eta^2) \mathbb{E}\Big[\|\frac{1}{K} \sum_{k=1}^{K} \nabla_x f^{(k)}(x_t^{(k)}, y_t^{(k)}) - \frac{1}{K} \sum_{k=1}^{K} g_t^{(k)}\|^2\Big] \\
&\quad + (c_0 \frac{25\gamma_y \eta}{3\mu} - c_2 \rho_y \eta^2) \mathbb{E}\Big[\|\frac{1}{K} \sum_{k=1}^{K} \nabla_y f^{(k)}(x_t^{(k)}, y_t^{(k)}) - \frac{1}{K} \sum_{k=1}^{K} h_t^{(k)}\|^2\Big] \\
&\quad + (-c_{11} \rho_x \eta^2 + C_3 3\rho_x^2 \eta^4) \frac{1}{K} \sum_{k=1}^{K} \mathbb{E}\Big[\Big\|g_t^{(k)} - \nabla_x f^{(k)}(x_t^{(k)}, y_t^{(k)})\Big\|^2\Big] \\
&\quad + (-c_{12} \rho_y \eta^2 + C_4 3\rho_y^2 \eta^4) \frac{1}{K} \sum_{k=1}^{K} \mathbb{E}\Big[\Big\|h_t^{(k)} - \nabla_y f^{(k)}(x_t^{(k)}, y_t^{(k)})\Big\|^2\Big] \\
&\quad + c_1 \frac{2\rho_x^2 \eta^4 \sigma^2}{K} + c_2 \frac{2\rho_y^2 \eta^4 \sigma^2}{K} + c_{11} 2\rho_x^2 \eta^4 \sigma^2 + c_{12} 2\rho_y^2 \eta^4 \sigma^2 + C_3 3\rho_x^2 \eta^4 \sigma^2 + C_4 3\rho_y^2 \eta^4 \sigma^2 ,
\end{aligned}
\tag{45}
$$

where

$$C_1 = c_1 \frac{2L^2}{K} + c_2 \frac{2L^2}{K} + c_9 \frac{6}{\delta} + c_{11} 2L^2 + c_{12} 2L^2 ,$$

$$C_2 = c_1 \frac{2L^2}{K} + c_2 \frac{2L^2}{K} + c_{10} \frac{6}{\delta} + c_{11} 2L^2 + c_{12} 2L^2 ,$$

$$C_3 = (c_5 \frac{11}{\alpha_u(1-\lambda)} + c_7 \frac{46}{\delta} + c_9 \frac{306\gamma_x^2\eta^2}{\delta}) ,$$

$$C_4 = (c_6 \frac{11}{\alpha_v(1-\lambda)} + c_8 \frac{46}{\delta} + c_{10} \frac{306\gamma_y^2\eta^2}{\delta}) ,$$

$$C_5 = C_1 12\alpha_x^2\eta^2 + C_3 36\alpha_x^2\eta^2 L^2 + C_4 36\alpha_x^2\eta^2 L^2 ,$$

$$C_6 = C_2 12\alpha_y^2\eta^2 + C_3 36\alpha_y^2\eta^2 L^2 + C_4 36\alpha_y^2\eta^2 L^2 . \tag{46}$$

To eliminate $\mathbb{E}[\|y^*(\bar{x}_t) - \bar{y}_t\|^2]$ , $\mathbb{E}\Big[\|\frac{1}{K}\sum_{k=1}^{K}\nabla_x f^{(k)}(x_t^{(k)}, y_t^{(k)}) - \frac{1}{K}\sum_{k=1}^{K} g_t^{(k)}\|^2\Big]$, and $\mathbb{E}\Big[\|\frac{1}{K}\sum_{k=1}^{K}\nabla_y f^{(k)}(x_t^{(k)}, y_t^{(k)}) - \frac{1}{K}\sum_{k=1}^{K} h_t^{(k)}\|^2\Big]$, we enforce

$$2\gamma_x\eta L^2 - c_0 \frac{\gamma_y\eta\mu}{4} \le 0 ,$$

$$2\gamma_x\eta - c_1\rho_x\eta^2 \le 0 ,$$

$$c_0 \frac{25\gamma_y\eta}{3\mu} - c_2\rho_y\eta^2 \le 0 . \tag{47}$$

Then, we can set

$$c_0 = \frac{8\gamma_x L^2}{\gamma_y\mu} , \quad c_1 = \frac{2\gamma_x}{\rho_x\eta} , \quad c_2 = \frac{200\gamma_x L^2}{3\rho_y\eta\mu^2} . \tag{48}$$

To eliminate $\frac{1}{K}\mathbb{E}\Big[\|U_t - \hat{U}_t\|_F^2\Big]$ and $\frac{1}{K}\mathbb{E}\Big[\|V_t - \hat{V}_t\|_F^2\Big]$, we enforce

$$c_5 \frac{8\alpha_u(1-\delta)}{1-\lambda} - c_7 \frac{\delta}{8} + c_9 \frac{144\gamma_x^2\alpha_u^2\eta^2}{\delta} \le 0 ,$$

$$c_6 \frac{8\alpha_v(1-\delta)}{1-\lambda} - c_8 \frac{\delta}{8} + c_{10} \frac{144\gamma_y^2\alpha_v^2\eta^2}{\delta} \le 0 . \tag{49}$$

Then, we can set

$$c_7 = c_5 \frac{64\alpha_u}{\delta(1-\lambda)} + c_9 \frac{1152\gamma_x^2\alpha_u^2\eta^2}{\delta^2} ,$$

$$c_8 = c_6 \frac{64\alpha_v}{\delta(1-\lambda)} + c_{10} \frac{1152\gamma_y^2\alpha_v^2\eta^2}{\delta^2} . \tag{50}$$

To eliminate $\frac{1}{K}\sum_{k=1}^{K}\mathbb{E}\Big[\|g_t^{(k)} - \nabla_x f^{(k)}(x_t^{(k)}, y_t^{(k)})\|^2\Big]$, we enforce

$$- c_{11}\rho_x\eta^2 + C_3 3\rho_x^2\eta^4$$

$$= -c_{11} + (c_5 \frac{11}{\alpha_u(1-\lambda)} + c_7 \frac{46}{\delta} + c_9 \frac{306\gamma_x^2\eta^2}{\delta})3\rho_x\eta^2$$

$$= -c_{11} + (c_5 \frac{11}{\alpha_u(1-\lambda)} + c_5 \frac{2944\alpha_u}{\delta^2(1-\lambda)} + c_9 \frac{52992\gamma_x^2\alpha_u^2\eta^2}{\delta^3} + c_9 \frac{306\gamma_x^2\eta^2}{\delta})3\rho_x\eta^2$$

$$\le 0 . \tag{51}$$

Then, we can set

$$c_{11} = c_5 \frac{33\eta^2 \rho_x}{\alpha_u (1-\lambda)} + c_5 \frac{8832\alpha_u \eta^2 \rho_x}{\delta^2 (1-\lambda)} + c_9 \frac{158976\gamma_x^2 \alpha_u^2 \eta^4 \rho_x}{\delta^3} + c_9 \frac{918\gamma_x^2 \eta^4 \rho_x}{\delta} \ . \tag{52}$$

To eliminate $\frac{1}{K} \sum_{k=1}^{K} \mathbb{E}\left[\left\| h_t^{(k)} - \nabla_y f^{(k)}(x_t^{(k)}, y_t^{(k)}) \right\|^2\right]$, we enforce

$$
\begin{aligned}
&- c_{12} \rho_y \eta^2 + C_4 3\rho_y^2 \eta^4 \\
&= -c_{12} + (c_6 \frac{11}{\alpha_v (1-\lambda)} + c_8 \frac{46}{\delta} + c_{10} \frac{306\gamma_y^2 \eta^2}{\delta}) 3\rho_y \eta^2 \\
&= -c_{12} + (c_6 \frac{11}{\alpha_v (1-\lambda)} + c_6 \frac{2944\alpha_v}{\delta^2 (1-\lambda)} + c_{10} \frac{52992\gamma_y^2 \alpha_v^2 \eta^2}{\delta^3} + c_{10} \frac{306\gamma_y^2 \eta^2}{\delta}) 3\rho_y \eta^2 \\
&\leq 0 \ .
\end{aligned}
\tag{53}
$$

Then, we set

$$c_{12} = c_6 \frac{33\eta^2 \rho_y}{\alpha_v (1-\lambda)} + c_6 \frac{8832\alpha_v \eta^2 \rho_y}{\delta^2 (1-\lambda)} + c_{10} \frac{158976\gamma_y^2 \alpha_v^2 \eta^4 \rho_y}{\delta^3} + c_{10} \frac{918\gamma_y^2 \eta^4 \rho_y}{\delta} \ . \tag{54}$$

In summary, we use $c_5$, $c_6$, $c_9$, and $c_{10}$ to represent $c_7$, $c_8$, $c_{11}$, and $c_{12}$ as follows, which is a critical step for our proof.

$$
\begin{aligned}
c_0 &= \frac{8\gamma_x L^2}{\gamma_y \mu} \ , \quad c_1 = \frac{2\gamma_x}{\rho_x \eta} \ , \quad c_2 = \frac{200\gamma_x L^2}{3\rho_y \eta \mu^2} \ , \\
c_7 &= c_5 \frac{64\alpha_u}{\delta (1-\lambda)} + c_9 \frac{1152\gamma_x^2 \alpha_u^2 \eta^2}{\delta^2} \ , \\
c_8 &= c_6 \frac{64\alpha_v}{\delta (1-\lambda)} + c_{10} \frac{1152\gamma_y^2 \alpha_v^2 \eta^2}{\delta^2} \ , \\
c_{11} &= c_5 \frac{33\eta^2 \rho_x}{\alpha_u (1-\lambda)} + c_5 \frac{8832\alpha_u \eta^2 \rho_x}{\delta^2 (1-\lambda)} + c_9 \frac{158976\gamma_x^2 \alpha_u^2 \eta^4 \rho_x}{\delta^3} + c_9 \frac{918\gamma_x^2 \eta^4 \rho_x}{\delta} \ , \\
c_{12} &= c_6 \frac{33\eta^2 \rho_y}{\alpha_v (1-\lambda)} + c_6 \frac{8832\alpha_v \eta^2 \rho_y}{\delta^2 (1-\lambda)} + c_{10} \frac{158976\gamma_y^2 \alpha_v^2 \eta^4 \rho_y}{\delta^3} + c_{10} \frac{918\gamma_y^2 \eta^4 \rho_y}{\delta} \ .
\end{aligned}
\tag{55}
$$

Based on these values, we can obtain

$$
\begin{aligned}
P_{t+1} &- P_t \\
\leq &- \frac{\gamma_x \eta}{2} \mathbb{E}\left[\|\nabla F(\bar{x}_t)\|^2\right] \\
&+ (-\frac{\gamma_x \eta}{4} + c_0 \frac{25\eta\gamma_x^2 L^2}{6\gamma_y \mu^3} + C_5 \frac{9\gamma_x^2}{2\alpha_x^2}) \mathbb{E}\left[\|\bar{g}_t\|^2\right] \\
&+ (-c_0 \frac{3\eta\gamma_y^2}{4} + C_6 \frac{9\gamma_y^2}{2\alpha_y^2}) \mathbb{E}\left[\|\bar{h}_t\|^2\right] \\
&+ (2\gamma_x \eta L^2 + c_0 \frac{25\gamma_y \eta L^2}{3\mu} - c_3 \frac{\alpha_x \eta (1-\lambda)}{2} + C_5) \frac{1}{K} \mathbb{E}\left[\|\bar{X}_t - X_t\|_F^2\right] \\
&+ (2\gamma_x \eta L^2 + c_0 \frac{25\gamma_y \eta L^2}{3\mu} - c_4 \frac{\alpha_y \eta (1-\lambda)}{2} + C_6) \frac{1}{K} \mathbb{E}\left[\|\bar{Y}_t - Y_t\|_F^2\right] \\
&+ (c_3 \frac{3\gamma_x^2 \eta}{\alpha_x (1-\lambda)} - c_5 \frac{\alpha_u (1-\lambda)}{2} + c_7 \frac{36\alpha_u^2}{\delta} + c_9 \frac{72\gamma_x^2 \alpha_u^2 \eta^2}{\delta} + C_5 \frac{9\gamma_x^2}{2\alpha_x^2}) \frac{1}{K} \mathbb{E}\left[\|U_t - \bar{U}_t\|_F^2\right] \\
&+ (c_4 \frac{3\gamma_y^2 \eta}{\alpha_y (1-\lambda)} - c_6 \frac{\alpha_v (1-\lambda)}{2} + c_8 \frac{36\alpha_v^2}{\delta} + c_{10} \frac{72\gamma_y^2 \alpha_v^2 \eta^2}{\delta} + C_6 \frac{9\gamma_y^2}{2\alpha_y^2}) \frac{1}{K} \mathbb{E}\left[\|V_t - \bar{V}_t\|_F^2\right] \\
&+ (c_3 \frac{4\alpha_x \eta (1-\delta)}{1-\lambda} - c_9 \frac{\delta}{2} + C_5) \frac{1}{K} \mathbb{E}\left[\|\tilde{X}_{t+1} - \hat{X}_t\|_F^2\right]
\end{aligned}
$$

$$+ \left(c_4 \frac{4\alpha_y \eta(1-\delta)}{1-\lambda} - c_{10}\frac{\delta}{2} + C_6\right)\frac{1}{K}\mathbb{E}\Big[\|\tilde{Y}_{t+1} - \hat{Y}_t\|_F^2\Big]$$

$$+ c_1 \frac{2\rho_x^2\eta^4\sigma^2}{K} + c_2 \frac{2\rho_y^2\eta^4\sigma^2}{K} + c_{11}2\rho_x^2\eta^4\sigma^2 + c_{12}2\rho_y^2\eta^4\sigma^2 + C_3 3\rho_x^2\eta^4\sigma^2 + C_4 3\rho_y^2\eta^4\sigma^2 \ . \tag{56}$$

In what follows, we use the values in Eq. (55) to simplify $C_1$, $C_2$, $C_3$, and $C_4$. It is easy to obtain

$$C_3 = c_5\frac{11}{\alpha_u(1-\lambda)} + c_5\frac{2944\alpha_u}{\delta^2(1-\lambda)} + c_9\frac{52992\gamma_x^2\alpha_u^2\eta^2}{\delta^3} + c_9\frac{306\gamma_x^2\eta^2}{\delta} \ ,$$

$$C_4 = c_6\frac{11}{\alpha_v(1-\lambda)} + c_6\frac{2944\alpha_v}{\delta^2(1-\lambda)} + c_{10}\frac{52992\gamma_y^2\alpha_v^2\eta^2}{\delta^3} + c_{10}\frac{306\gamma_y^2\eta^2}{\delta} \ ,$$

$$C_1 = c_1\frac{2L^2}{K} + c_2\frac{2L^2}{K} + c_9\frac{6}{\delta} + c_{11}2L^2 + c_{12}2L^2$$

$$= \frac{2\gamma_x}{\rho_x\eta}\frac{2L^2}{K} + \frac{200\gamma_x L^2}{3\rho_y\eta\mu^2}\frac{2L^2}{K} + c_9\frac{6}{\delta}$$

$$+ \left(c_5\frac{33\eta^2\rho_x}{\alpha_u(1-\lambda)} + c_5\frac{8832\alpha_u\eta^2\rho_x}{\delta^2(1-\lambda)} + c_9\frac{158976\gamma_x^2\alpha_u^2\eta^4\rho_x}{\delta^3} + c_9\frac{918\gamma_x^2\eta^4\rho_x}{\delta}\right)2L^2$$

$$+ \left(c_6\frac{33\eta^2\rho_y}{\alpha_v(1-\lambda)} + c_6\frac{8832\alpha_v\eta^2\rho_y}{\delta^2(1-\lambda)} + c_{10}\frac{158976\gamma_y^2\alpha_v^2\eta^4\rho_y}{\delta^3} + c_{10}\frac{918\gamma_y^2\eta^4\rho_y}{\delta}\right)2L^2 \ ,$$

$$C_2 = c_1\frac{2L^2}{K} + c_2\frac{2L^2}{K} + c_{10}\frac{6}{\delta} + c_{11}2L^2 + c_{12}2L^2$$

$$= \frac{2\gamma_x}{\rho_x\eta}\frac{2L^2}{K} + \frac{200\gamma_x L^2}{3\rho_y\eta\mu^2}\frac{2L^2}{K} + c_{10}\frac{6}{\delta}$$

$$+ \left(c_5\frac{33\eta^2\rho_x}{\alpha_u(1-\lambda)} + c_5\frac{8832\alpha_u\eta^2\rho_x}{\delta^2(1-\lambda)} + c_9\frac{158976\gamma_x^2\alpha_u^2\eta^4\rho_x}{\delta^3} + c_9\frac{918\gamma_x^2\eta^4\rho_x}{\delta}\right)2L^2$$

$$+ \left(c_6\frac{33\eta^2\rho_y}{\alpha_v(1-\lambda)} + c_6\frac{8832\alpha_v\eta^2\rho_y}{\delta^2(1-\lambda)} + c_{10}\frac{158976\gamma_y^2\alpha_v^2\eta^4\rho_y}{\delta^3} + c_{10}\frac{918\gamma_y^2\eta^4\rho_y}{\delta}\right)2L^2 \ . \tag{57}$$

After simplifying $C_1$, $C_2$, $C_3$, and $C_4$, we further simplify $C_5$ and $C_6$ as follows:

$$C_5$$

$$= C_1 12\alpha_x^2\eta^2 + C_3 36\alpha_x^2\eta^2 L^2 + C_4 36\alpha_x^2\eta^2 L^2$$

$$= \left(c_1\frac{2L^2}{K} + c_2\frac{2L^2}{K} + c_9\frac{6}{\delta} + (C_3 3\rho_x\eta^2)2L^2 + (C_4 3\rho_y\eta^2)2L^2\right)12\alpha_x^2\eta^2$$

$$+ C_3 36\alpha_x^2\eta^2 L^2 + C_4 36\alpha_x^2\eta^2 L^2$$

$$\leq c_1\frac{2L^2}{K}12\alpha_x^2\eta^2 + c_2\frac{2L^2}{K}12\alpha_x^2\eta^2 + c_9\frac{6}{\delta}12\alpha_x^2\eta^2 + C_3 108\alpha_x^2\eta^2 L^2 + C_4 108\alpha_x^2\eta^2 L^2$$

$$= \frac{48\gamma_x\alpha_x^2\eta L^2}{\rho_x K} + \frac{4800\gamma_x\alpha_x^2\eta L^4}{3\rho_y\mu^2 K} + c_9\frac{72\alpha_x^2\eta^2}{\delta}$$

$$+ c_5\frac{1188\alpha_x^2\eta^2 L^2}{\alpha_u(1-\lambda)} + c_5\frac{317952\alpha_x^2\alpha_u\eta^2 L^2}{\delta^2(1-\lambda)}$$

$$+ c_9\frac{5723136\alpha_x^2\gamma_x^2\alpha_u^2\eta^4 L^2}{\delta^3} + c_9\frac{33048\alpha_x^2\gamma_x^2\eta^4 L^2}{\delta}$$

$$+ c_6\frac{1188\alpha_x^2\eta^2 L^2}{\alpha_v(1-\lambda)} + c_6\frac{317952\alpha_x^2\alpha_v\eta^2 L^2}{\delta^2(1-\lambda)}$$

$$+ c_{10}\frac{5723136\alpha_x^2\gamma_y^2\alpha_v^2\eta^4 L^2}{\delta^3} + c_{10}\frac{33048\alpha_x^2\gamma_y^2\eta^4 L^2}{\delta} \ , \tag{58}$$

where the second to last step holds due to $\rho_x\eta^2 < 1$ and $\rho_y\eta^2 < 1$.

$$C_6$$

$$
\begin{aligned}
&= C_2 12\alpha_y^2\eta^2 + C_3 36\alpha_y^2\eta^2 L^2 + C_4 36\alpha_y^2\eta^2 L^2 \\
&= c_1\frac{2L^2}{K}12\alpha_y^2\eta^2 + c_2\frac{2L^2}{K}12\alpha_y^2\eta^2 + c_{10}\frac{6}{\delta}12\alpha_y^2\eta^2 + (C_3 3\rho_x\eta^2)2L^2 12\alpha_y^2\eta^2 \\
&\quad + (C_4 3\rho_y\eta^2)2L^2 12\alpha_y^2\eta^2 + C_3 36\alpha_y^2\eta^2 L^2 + C_4 36\alpha_y^2\eta^2 L^2 \\
&\leq c_1\frac{2L^2}{K}12\alpha_y^2\eta^2 + c_2\frac{2L^2}{K}12\alpha_y^2\eta^2 + c_{10}\frac{6}{\delta}12\alpha_y^2\eta^2 + C_3 108\alpha_y^2\eta^2 L^2 + C_4 108\alpha_y^2\eta^2 L^2 \\
&= \frac{48\gamma_x\alpha_y^2\eta L^2}{\rho_x K} + \frac{4800\gamma_x\alpha_y^2\eta L^4}{3\rho_y\mu^2 K} + c_{10}\frac{72\alpha_y^2\eta^2}{\delta} \\
&\quad + c_5\frac{1188\alpha_y^2\eta^2 L^2}{\alpha_u(1-\lambda)} + c_5\frac{317952\alpha_y^2\alpha_u\eta^2 L^2}{\delta^2(1-\lambda)} \\
&\quad + c_9\frac{5723136\alpha_y^2\gamma_x^2\alpha_u^2\eta^4 L^2}{\delta^3} + c_9\frac{33048\alpha_y^2\gamma_x^2\eta^4 L^2}{\delta} \\
&\quad + c_6\frac{1188\alpha_y^2\eta^2 L^2}{\alpha_v(1-\lambda)} + c_6\frac{317952\alpha_y^2\alpha_v\eta^2 L^2}{\delta^2(1-\lambda)} \\
&\quad + c_{10}\frac{5723136\alpha_y^2\gamma_y^2\alpha_v^2\eta^4 L^2}{\delta^3} + c_{10}\frac{33048\alpha_y^2\gamma_y^2\eta^4 L^2}{\delta}\ ,
\end{aligned}
\tag{59}
$$

where the second to last step holds due to $\rho_x\eta^2 < 1$ and $\rho_y\eta^2 < 1$.

It can be observed that $\{C_i\}_{i=1}^6$ and $\{c_7, c_8, c_{11}, c_{12}\}$ are represented by $c_5$, $c_6$, $c_9$, and $c_{10}$. In what follows, we further represent $\{c_3, c_4\}$ by $c_5$, $c_6$, $c_9$, and $c_{10}$. Then, after determining the value of $\{c_5, c_6, c_9, c_{10}\}$, we can infer the value of $\{c_3, c_4, c_7, c_8, c_{11}, c_{12}\}$ and $\{C_i\}_{i=1}^6$.

Specifically, to eliminate $\frac{1}{K}\mathbb{E}\left[\|\bar{X}_t - X_t\|_F^2\right]$, we enforce

$$
2\gamma_x\eta L^2 + c_0\frac{25\gamma_y\eta L^2}{3\mu} - c_3\frac{\alpha_x\eta(1-\lambda)}{2} + C_5 \leq 0\ .
\tag{60}
$$

Then, we can set

$$
c_3 = \frac{4\gamma_x L^2}{\alpha_x(1-\lambda)} + \frac{400\gamma_x L^4}{3\alpha_x\mu^2(1-\lambda)} + C_5\frac{2}{\alpha_x\eta(1-\lambda)}\ .
\tag{61}
$$

Similarly, to eliminate $\frac{1}{K}\mathbb{E}\left[\|\bar{Y}_t - Y_t\|_F^2\right]$, we enforce

$$
2\gamma_x\eta L^2 + c_0\frac{25\gamma_y\eta L^2}{3\mu} - c_4\frac{\alpha_y\eta(1-\lambda)}{2} + C_6 \leq 0\ .
\tag{62}
$$

Then, we can set

$$
c_4 = \frac{4\gamma_x L^2}{\alpha_y(1-\lambda)} + \frac{400\gamma_x L^4}{3\alpha_y\mu^2(1-\lambda)} + C_6\frac{2}{\alpha_y\eta(1-\lambda)}\ .
\tag{63}
$$

By now, $c_3$ and $c_4$ are also represented by $\{c_5, c_6, c_9, c_{10}\}$. In what follows, we determine the value of $\{c_5, c_6, c_9, c_{10}\}$.

Specifically, to eliminate $\frac{1}{K}\mathbb{E}\left[\|\tilde{X}_{t+1} - \hat{X}_t\|_F^2\right]$, we enforce

$$
\begin{aligned}
&c_3\frac{4\alpha_x\eta(1-\delta)}{1-\lambda} - c_9\frac{\delta}{2} + C_5 \\
&= \left(\frac{4\gamma_x L^2}{\alpha_x(1-\lambda)} + \frac{400\gamma_x L^4}{3\alpha_x\mu^2(1-\lambda)} + C_5\frac{2}{\alpha_x\eta(1-\lambda)}\right)\frac{4\alpha_x\eta(1-\delta)}{1-\lambda} - c_9\frac{\delta}{2} + C_5 \\
&\leq \frac{16\gamma_x\eta(1-\delta)L^2}{(1-\lambda)^2} + \frac{1600\gamma_x\eta(1-\delta)L^4}{3\mu^2(1-\lambda)^2} - c_9\frac{\delta}{2} + C_5\frac{9}{(1-\lambda)^2}
\end{aligned}
$$

$$\leq \frac{16\gamma_x\eta L^2}{(1-\lambda)^2} + \frac{534\gamma_x\eta L^4}{\mu^2(1-\lambda)^2} - c_9\frac{\delta}{2}$$

$$+ \frac{48\gamma_x\alpha_x^2\eta L^2}{\rho_x K}\frac{9}{(1-\lambda)^2} + \frac{4800\gamma_x\alpha_x^2\eta L^4}{3\rho_y\mu^2 K}\frac{9}{(1-\lambda)^2} + c_9\frac{72\alpha_x^2\eta^2}{\delta}\frac{9}{(1-\lambda)^2}$$

$$+ c_5\frac{1188\alpha_x^2\eta^2 L^2}{\alpha_u(1-\lambda)}\frac{9}{(1-\lambda)^2} + c_5\frac{317952\alpha_x^2\alpha_u\eta^2 L^2}{\delta^2(1-\lambda)}\frac{9}{(1-\lambda)^2}$$

$$+ c_9\frac{5723136\alpha_x^2\gamma_x^2\alpha_u^2\eta^4 L^2}{\delta^3}\frac{9}{(1-\lambda)^2} + c_9\frac{33048\alpha_x^2\gamma_x^2\eta^4 L^2}{\delta}\frac{9}{(1-\lambda)^2}$$

$$+ c_6\frac{1188\alpha_x^2\eta^2 L^2}{\alpha_v(1-\lambda)}\frac{9}{(1-\lambda)^2} + c_6\frac{317952\alpha_x^2\alpha_v\eta^2 L^2}{\delta^2(1-\lambda)}\frac{9}{(1-\lambda)^2}$$

$$+ c_{10}\frac{5723136\alpha_x^2\gamma_y^2\alpha_v^2\eta^4 L^2}{\delta^3}\frac{9}{(1-\lambda)^2} + c_{10}\frac{33048\alpha_x^2\gamma_y^2\eta^4 L^2}{\delta}\frac{9}{(1-\lambda)^2}$$

$$= \frac{16\gamma_x\eta L^2}{(1-\lambda)^2} + \frac{534\gamma_x\eta L^4}{\mu^2(1-\lambda)^2} - c_9\frac{\delta}{2}$$

$$+ \frac{432\gamma_x\alpha_x^2\eta L^2}{\rho_x(1-\lambda)^2 K} + \frac{14400\gamma_x\alpha_x^2\eta L^4}{\rho_y\mu^2(1-\lambda)^2 K} + c_9\frac{648\alpha_x^2\eta^2}{\delta(1-\lambda)^2}$$

$$+ c_5\frac{10692\alpha_x^2\eta^2 L^2}{\alpha_u(1-\lambda)^3} + c_5\frac{2861568\alpha_x^2\alpha_u\eta^2 L^2}{\delta^2(1-\lambda)^3}$$

$$+ c_9\frac{51508224\alpha_x^2\gamma_x^2\alpha_u^2\eta^4 L^2}{\delta^3(1-\lambda)^2} + c_9\frac{297432\alpha_x^2\gamma_x^2\eta^4 L^2}{\delta(1-\lambda)^2}$$

$$+ c_6\frac{10692\alpha_x^2\eta^2 L^2}{\alpha_v(1-\lambda)^3} + c_6\frac{2861568\alpha_x^2\alpha_v\eta^2 L^2}{\delta^2(1-\lambda)^3}$$

$$+ c_{10}\frac{51508224\alpha_x^2\gamma_y^2\alpha_v^2\eta^4 L^2}{\delta^3(1-\lambda)^2} + c_{10}\frac{297432\alpha_x^2\gamma_y^2\eta^4 L^2}{\delta(1-\lambda)^2}$$

$$\leq 0 \,. \tag{64}$$

To solve this inequality, we can set

$$\frac{16\gamma_x\eta L^2}{(1-\lambda)^2} + \frac{534\gamma_x\eta L^4}{\mu^2(1-\lambda)^2} + \frac{432\gamma_x\alpha_x^2\eta L^2}{\rho_x(1-\lambda)^2 K} + \frac{14400\gamma_x\alpha_x^2\eta L^4}{\rho_y\mu^2(1-\lambda)^2 K} \leq c_9\frac{\delta}{20} \,, \tag{65}$$

$$c_9\frac{648\alpha_x^2\eta^2}{\delta(1-\lambda)^2} \leq c_9\frac{\delta}{20} \,, \tag{66}$$

$$c_9\frac{51508224\alpha_x^2\gamma_x^2\alpha_u^2\eta^4 L^2}{\delta^3(1-\lambda)^2} \leq c_9\frac{\delta}{20} \,, \tag{67}$$

$$c_9\frac{297432\alpha_x^2\gamma_x^2\eta^4 L^2}{\delta(1-\lambda)^2} \leq c_9\frac{\delta}{20} \,, \tag{68}$$

$$c_5\frac{10692\alpha_x^2\eta^2 L^2}{\alpha_u(1-\lambda)^3} \leq c_9\frac{\delta}{20} \,, \tag{69}$$

$$c_5\frac{2861568\alpha_x^2\alpha_u\eta^2 L^2}{\delta^2(1-\lambda)^3} \leq c_9\frac{\delta}{20} \,, \tag{70}$$

$$c_6\frac{10692\alpha_x^2\eta^2 L^2}{\alpha_v(1-\lambda)^3} \leq c_9\frac{\delta}{20} \,, \tag{71}$$

$$c_6\frac{2861568\alpha_x^2\alpha_v\eta^2 L^2}{\delta^2(1-\lambda)^3} \leq c_9\frac{\delta}{20} \,, \tag{72}$$

$$c_{10}\frac{51508224\alpha_x^2\gamma_y^2\alpha_v^2\eta^4 L^2}{\delta^3(1-\lambda)^2} \leq c_9\frac{\delta}{20} \,, \tag{73}$$

$$c_{10}\frac{297432\alpha_x^2\gamma_y^2\eta^4 L^2}{\delta(1-\lambda)^2} \leq c_9\frac{\delta}{20} \ . \tag{74}$$

To solve these inequalities, we will first solve Eq. (65) to obtain the value of $c_9$ and then solve Eqs. (66-68). However, since the value of $\{c_5, c_6, c_{10}\}$ is not available at this moment, we will solve Eqs. (69-74) after all values of $\{c_5, c_6, c_9, c_{10}\}$ are determined.

Specifically, to solve Eq. (65), we can set

$$c_9 = \frac{11000\gamma_x\eta L^4}{\mu^2\delta(1-\lambda)^2} + \frac{8640\gamma_x\alpha_x^2\eta L^2}{\rho_x\delta(1-\lambda)^2 K} + \frac{288000\gamma_x\alpha_x^2\eta L^4}{\rho_y\mu^2\delta(1-\lambda)^2 K} \ . \tag{75}$$

To solve Eq. (66) with $\eta < 1$, we can set

$$\alpha_x \leq \frac{\delta(1-\lambda)}{114} \ . \tag{76}$$

To solve Eq. (67) with $\alpha_u < 1$, $\alpha_x\eta < 1$ and $\eta < 1$, we can set

$$\gamma_x \leq \frac{\delta^2(1-\lambda)}{32097 L} \ . \tag{77}$$

To solve Eq. (68) with $\alpha_x\eta < 1$ and $\eta < 1$, we can set

$$\gamma_x \leq \frac{\delta(1-\lambda)}{2439 L} \ . \tag{78}$$

Similarly, to eliminate $\frac{1}{K}\mathbb{E}\left[\|\tilde{Y}_{t+1} - \hat{Y}_t\|_F^2\right]$, we enforce

$$
\begin{aligned}
& c_4\frac{4\alpha_y\eta(1-\delta)}{1-\lambda} - c_{10}\frac{\delta}{2} + C_6 \\
\leq\ & \frac{16\eta\gamma_x(1-\delta)L^2}{(1-\lambda)^2} + \frac{534\eta\gamma_x(1-\delta)L^4}{\mu^2(1-\lambda)^2} + C_6\frac{8(1-\delta)}{(1-\lambda)^2} + C_6 - c_{10}\frac{\delta}{2} \\
\leq\ & \frac{16\eta\gamma_x L^2}{(1-\lambda)^2} + \frac{534\eta\gamma_x L^4}{\mu^2(1-\lambda)^2} + C_6\frac{9}{(1-\lambda)^2} - c_{10}\frac{\delta}{2} \\
\leq\ & \frac{16\eta\gamma_x L^2}{(1-\lambda)^2} + \frac{534\eta\gamma_x L^4}{\mu^2(1-\lambda)^2} - c_{10}\frac{\delta}{2} \\
& + \frac{48\gamma_x\alpha_y^2\eta L^2}{\rho_x K}\frac{9}{(1-\lambda)^2} + \frac{4800\gamma_x\alpha_y^2\eta L^4}{3\rho_y\mu^2 K}\frac{9}{(1-\lambda)^2} + c_{10}\frac{72\alpha_y^2\eta^2}{\delta}\frac{9}{(1-\lambda)^2} \\
& + c_5\frac{1188\alpha_y^2\eta^2 L^2}{\alpha_u(1-\lambda)}\frac{9}{(1-\lambda)^2} + c_5\frac{317952\alpha_y^2\alpha_u\eta^2 L^2}{\delta^2(1-\lambda)}\frac{9}{(1-\lambda)^2} \\
& + c_9\frac{5723136\alpha_y^2\gamma_x^2\alpha_u^2\eta^4 L^2}{\delta^3}\frac{9}{(1-\lambda)^2} + c_9\frac{33048\alpha_y^2\gamma_x^2\eta^4 L^2}{\delta}\frac{9}{(1-\lambda)^2} \\
& + c_6\frac{1188\alpha_y^2\eta^2 L^2}{\alpha_v(1-\lambda)}\frac{9}{(1-\lambda)^2} + c_6\frac{317952\alpha_y^2\alpha_v\eta^2 L^2}{\delta^2(1-\lambda)}\frac{9}{(1-\lambda)^2} \\
& + c_{10}\frac{5723136\alpha_y^2\gamma_y^2\alpha_v^2\eta^4 L^2}{\delta^3}\frac{9}{(1-\lambda)^2} + c_{10}\frac{33048\alpha_y^2\gamma_y^2\eta^4 L^2}{\delta}\frac{9}{(1-\lambda)^2} \\
=\ & \frac{16\eta\gamma_x L^2}{(1-\lambda)^2} + \frac{534\eta\gamma_x L^4}{\mu^2(1-\lambda)^2} - c_{10}\frac{\delta}{2} \\
& + \frac{432\gamma_x\alpha_y^2\eta L^2}{\rho_x(1-\lambda)^2 K} + \frac{14400\gamma_x\alpha_y^2\eta L^4}{\rho_y\mu^2(1-\lambda)^2 K} + c_{10}\frac{648\alpha_y^2\eta^2}{\delta(1-\lambda)^2} \\
& + c_5\frac{10692\alpha_y^2\eta^2 L^2}{\alpha_u(1-\lambda)^3} + c_5\frac{2861568\alpha_y^2\alpha_u\eta^2 L^2}{\delta^2(1-\lambda)^3}
\end{aligned}
$$

$$+ c_9 \frac{51508224\alpha_y^2\gamma_x^2\alpha_u^2\eta^4 L^2}{\delta^3(1-\lambda)^2} + c_9 \frac{297432\alpha_y^2\gamma_x^2\eta^4 L^2}{\delta(1-\lambda)^2}$$

$$+ c_6 \frac{10692\alpha_y^2\eta^2 L^2}{\alpha_v(1-\lambda)^3} + c_6 \frac{2861568\alpha_y^2\alpha_v\eta^2 L^2}{\delta^2(1-\lambda)^3}$$

$$+ c_{10} \frac{51508224\alpha_y^2\gamma_y^2\alpha_v^2\eta^4 L^2}{\delta^3(1-\lambda)^2} + c_{10} \frac{297432\alpha_y^2\gamma_y^2\eta^4 L^2}{\delta(1-\lambda)^2} \le 0 . \tag{79}$$

To solve this inequality, we can set

$$\frac{16\eta\gamma_x L^2}{(1-\lambda)^2} + \frac{534\eta\gamma_x L^4}{\mu^2(1-\lambda)^2} + \frac{432\gamma_x\alpha_y^2\eta L^2}{\rho_x(1-\lambda)^2 K} + \frac{14400\gamma_x\alpha_y^2\eta L^4}{\rho_y\mu^2(1-\lambda)^2 K} \le c_{10}\frac{\delta}{20} , \tag{80}$$

$$c_{10} \frac{648\alpha_y^2\eta^2}{\delta(1-\lambda)^2} \le c_{10}\frac{\delta}{20} , \tag{81}$$

$$c_{10} \frac{51508224\alpha_y^2\gamma_y^2\alpha_v^2\eta^4 L^2}{\delta^3(1-\lambda)^2} \le c_{10}\frac{\delta}{20} , \tag{82}$$

$$c_{10} \frac{297432\alpha_y^2\gamma_y^2\eta^4 L^2}{\delta(1-\lambda)^2} \le c_{10}\frac{\delta}{20} , \tag{83}$$

$$c_5 \frac{10692\alpha_y^2\eta^2 L^2}{\alpha_u(1-\lambda)^3} \le c_{10}\frac{\delta}{20} , \tag{84}$$

$$c_5 \frac{2861568\alpha_y^2\alpha_u\eta^2 L^2}{\delta^2(1-\lambda)^3} \le c_{10}\frac{\delta}{20} , \tag{85}$$

$$c_9 \frac{51508224\alpha_y^2\gamma_x^2\alpha_u^2\eta^4 L^2}{\delta^3(1-\lambda)^2} \le c_{10}\frac{\delta}{20} , \tag{86}$$

$$c_9 \frac{297432\alpha_y^2\gamma_x^2\eta^4 L^2}{\delta(1-\lambda)^2} \le c_{10}\frac{\delta}{20} , \tag{87}$$

$$c_6 \frac{10692\alpha_y^2\eta^2 L^2}{\alpha_v(1-\lambda)^3} \le c_{10}\frac{\delta}{20} , \tag{88}$$

$$c_6 \frac{2861568\alpha_y^2\alpha_v\eta^2 L^2}{\delta^2(1-\lambda)^3} \le c_{10}\frac{\delta}{20} . \tag{89}$$

Similarly, we first solve Eqs. (80-83). After all values of $\{c_5, c_6, c_9, c_{10}\}$ are determined, we solve Eqs. (84-89).

Specifically, to solve Eq. (80), we can set

$$c_{10} = \frac{11000\gamma_x\eta L^4}{\mu^2\delta(1-\lambda)^2} + \frac{8640\gamma_x\alpha_y^2\eta L^2}{\rho_x\delta(1-\lambda)^2 K} + \frac{288000\gamma_x\alpha_y^2\eta L^4}{\rho_y\mu^2\delta(1-\lambda)^2 K} . \tag{90}$$

To solve Eq. (81) with $\eta < 1$, we can set

$$\alpha_y \le \frac{\delta(1-\lambda)}{114} . \tag{91}$$

To solve Eq. (82) with $\alpha_v < 1$, $\alpha_y\eta < 1$ and $\eta < 1$, we can set

$$\gamma_y \le \frac{\delta^2(1-\lambda)}{32097L} . \tag{92}$$

To solve Eq. (83) with $\alpha_y\eta < 1$ and $\eta < 1$, we can set

$$\gamma_y \le \frac{\delta(1-\lambda)}{2439L} . \tag{93}$$

Moreover, to eliminate $\frac{1}{K}\mathbb{E}\left[\|U_t - \bar{U}_t\|_F^2\right]$, we enforce

$$
c_3 \frac{3\gamma_x^2\eta}{\alpha_x(1-\lambda)} - c_5 \frac{\alpha_u(1-\lambda)}{2} + c_7 \frac{36\alpha_u^2}{\delta} + c_9 \frac{72\gamma_x^2\alpha_u^2\eta^2}{\delta} + C_5 \frac{9\gamma_x^2}{2\alpha_x^2}
$$

$$
= \frac{4\gamma_x L^2}{\alpha_x(1-\lambda)} \frac{3\gamma_x^2\eta}{\alpha_x(1-\lambda)} + \frac{400\gamma_x L^4}{3\alpha_x\mu^2(1-\lambda)} \frac{3\gamma_x^2\eta}{\alpha_x(1-\lambda)} + C_5 \frac{2}{\alpha_x\eta(1-\lambda)} \frac{3\gamma_x^2\eta}{\alpha_x(1-\lambda)}
$$

$$
+ c_5 \frac{64\alpha_u}{\delta(1-\lambda)} \frac{36\alpha_u^2}{\delta} + c_9 \frac{1152\gamma_x^2\alpha_u^2\eta^2}{\delta^2} \frac{36\alpha_u^2}{\delta}
$$

$$
- c_5 \frac{\alpha_u(1-\lambda)}{2} + c_9 \frac{72\gamma_x^2\alpha_u^2\eta^2}{\delta} + \frac{9\gamma_x^2}{2\alpha_x^2} C_5
$$

$$
\leq \frac{12\eta\gamma_x^3 L^2}{\alpha_x^2(1-\lambda)^2} + \frac{400\eta\gamma_x^3 L^4}{\alpha_x^2\mu^2(1-\lambda)^2} + C_5 \frac{11\gamma_x^2}{\alpha_x^2(1-\lambda)^2} + c_5 \frac{2304\alpha_u^3}{\delta^2(1-\lambda)} + c_9 \frac{41472\gamma_x^2\alpha_u^4\eta^2}{\delta^3}
$$

$$
- c_5 \frac{\alpha_u(1-\lambda)}{2} + c_9 \frac{72\gamma_x^2\alpha_u^2\eta^2}{\delta}
$$

$$
\leq \frac{12\eta\gamma_x^3 L^2}{\alpha_x^2(1-\lambda)^2} + \frac{400\eta\gamma_x^3 L^4}{\alpha_x^2\mu^2(1-\lambda)^2}
$$

$$
+ c_5 \frac{2304\alpha_u^3}{\delta^2(1-\lambda)} - c_5 \frac{\alpha_u(1-\lambda)}{2}
$$

$$
+ c_9 \frac{41472\gamma_x^2\alpha_u^4\eta^2}{\delta^3} + c_9 \frac{72\gamma_x^2\alpha_u^2\eta^2}{\delta}
$$

$$
+ \frac{48\gamma_x\alpha_x^2\eta L^2}{\rho_x K} \frac{11\gamma_x^2}{\alpha_x^2(1-\lambda)^2} + \frac{4800\gamma_x\alpha_x^2\eta L^4}{3\rho_y\mu^2 K} \frac{11\gamma_x^2}{\alpha_x^2(1-\lambda)^2} + c_9 \frac{72\alpha_x^2\eta^2}{\delta} \frac{11\gamma_x^2}{\alpha_x^2(1-\lambda)^2}
$$

$$
+ c_5 \frac{1188\alpha_x^2\eta^2 L^2}{\alpha_u(1-\lambda)} \frac{11\gamma_x^2}{\alpha_x^2(1-\lambda)^2} + c_5 \frac{317952\alpha_x^2\alpha_u\eta^2 L^2}{\delta^2(1-\lambda)} \frac{11\gamma_x^2}{\alpha_x^2(1-\lambda)^2}
$$

$$
+ c_9 \frac{5723136\alpha_x^2\gamma_x^2\alpha_u^2\eta^4 L^2}{\delta^3} \frac{11\gamma_x^2}{\alpha_x^2(1-\lambda)^2} + c_9 \frac{33048\alpha_x^2\gamma_x^2\eta^4 L^2}{\delta} \frac{11\gamma_x^2}{\alpha_x^2(1-\lambda)^2}
$$

$$
+ c_6 \frac{1188\alpha_x^2\eta^2 L^2}{\alpha_v(1-\lambda)} \frac{11\gamma_x^2}{\alpha_x^2(1-\lambda)^2} + c_6 \frac{317952\alpha_x^2\alpha_v\eta^2 L^2}{\delta^2(1-\lambda)} \frac{11\gamma_x^2}{\alpha_x^2(1-\lambda)^2}
$$

$$
+ c_{10} \frac{5723136\alpha_x^2\gamma_y^2\alpha_v^2\eta^4 L^2}{\delta^3} \frac{11\gamma_x^2}{\alpha_x^2(1-\lambda)^2} + c_{10} \frac{33048\alpha_x^2\gamma_y^2\eta^4 L^2}{\delta} \frac{11\gamma_x^2}{\alpha_x^2(1-\lambda)^2}
$$

$$
= \frac{12\eta\gamma_x^3 L^2}{\alpha_x^2(1-\lambda)^2} + \frac{400\eta\gamma_x^3 L^4}{\alpha_x^2\mu^2(1-\lambda)^2}
$$

$$
+ c_5 \frac{2304\alpha_u^3}{\delta^2(1-\lambda)} - c_5 \frac{\alpha_u(1-\lambda)}{2}
$$

$$
+ c_9 \frac{41472\gamma_x^2\alpha_u^4\eta^2}{\delta^3} + c_9 \frac{72\gamma_x^2\alpha_u^2\eta^2}{\delta}
$$

$$
+ \frac{528\gamma_x^3\eta L^2}{\rho_x(1-\lambda)^2 K} + \frac{17600\gamma_x^3\eta L^4}{\rho_y\mu^2(1-\lambda)^2 K} + c_9 \frac{792\eta^2\gamma_x^2}{\delta(1-\lambda)^2}
$$

$$
+ c_5 \frac{13068\eta^2\gamma_x^2 L^2}{\alpha_u(1-\lambda)^3} + c_5 \frac{3497472\alpha_u\eta^2\gamma_x^2 L^2}{\delta^2(1-\lambda)^3}
$$

$$
+ c_9 \frac{62954496\alpha_u^2\eta^4\gamma_x^4 L^2}{\delta^3(1-\lambda)^2} + c_9 \frac{363528\eta^4\gamma_x^4 L^2}{\delta(1-\lambda)^2}
$$

$$
+ c_6 \frac{13068\eta^2\gamma_x^2 L^2}{\alpha_v(1-\lambda)^3} + c_6 \frac{3497472\alpha_v\eta^2\gamma_x^2 L^2}{\delta^2(1-\lambda)^3}
$$

$$
+ c_{10} \frac{62954496\alpha_v^2\eta^4\gamma_x^2\gamma_y^2 L^2}{\delta^3(1-\lambda)^2} + c_{10} \frac{363528\eta^4\gamma_x^2\gamma_y^2 L^2}{\delta(1-\lambda)^2} \ . \tag{94}
$$

To solve this inequality, we set

$$\frac{12\eta\gamma_x^3 L^2}{\alpha_x^2(1-\lambda)^2} + \frac{400\eta\gamma_x^3 L^4}{\alpha_x^2\mu^2(1-\lambda)^2} + \frac{528\gamma_x^3\eta L^2}{\rho_x(1-\lambda)^2 K} + \frac{17600\gamma_x^3\eta L^4}{\rho_y\mu^2(1-\lambda)^2 K} \le c_5\frac{\alpha_u(1-\lambda)}{30} \ , \tag{95}$$

$$c_5\frac{2304\alpha_u^3}{\delta^2(1-\lambda)} \le c_5\frac{\alpha_u(1-\lambda)}{30} \ , \tag{96}$$

$$c_5\frac{13068\eta^2\gamma_x^2 L^2}{\alpha_u(1-\lambda)^3} \le c_5\frac{\alpha_u(1-\lambda)}{30} \ , \tag{97}$$

$$c_5\frac{3497472\alpha_u\eta^2\gamma_x^2 L^2}{\delta^2(1-\lambda)^3} \le c_5\frac{\alpha_u(1-\lambda)}{30} \ , \tag{98}$$

$$c_9\frac{41472\gamma_x^2\alpha_u^4\eta^2}{\delta^3} \le c_5\frac{\alpha_u(1-\lambda)}{30} \ , \tag{99}$$

$$c_9\frac{72\gamma_x^2\alpha_u^2\eta^2}{\delta} \le c_5\frac{\alpha_u(1-\lambda)}{30} \ , \tag{100}$$

$$c_9\frac{792\eta^2\gamma_x^2}{\delta(1-\lambda)^2} \le c_5\frac{\alpha_u(1-\lambda)}{30} \ , \tag{101}$$

$$c_9\frac{62954496\alpha_u^2\eta^4\gamma_x^4 L^2}{\delta^3(1-\lambda)^2} \le c_5\frac{\alpha_u(1-\lambda)}{30} \ , \tag{102}$$

$$c_9\frac{363528\eta^4\gamma_x^4 L^2}{\delta(1-\lambda)^2} \le c_5\frac{\alpha_u(1-\lambda)}{30} \ , \tag{103}$$

$$c_6\frac{13068\eta^2\gamma_x^2 L^2}{\alpha_v(1-\lambda)^3} \le c_5\frac{\alpha_u(1-\lambda)}{30} \ , \tag{104}$$

$$c_6\frac{3497472\alpha_v\eta^2\gamma_x^2 L^2}{\delta^2(1-\lambda)^3} \le c_5\frac{\alpha_u(1-\lambda)}{30} \ , \tag{105}$$

$$c_{10}\frac{62954496\alpha_v^2\eta^4\gamma_x^2\gamma_y^2 L^2}{\delta^3(1-\lambda)^2} \le c_5\frac{\alpha_u(1-\lambda)}{30} \ , \tag{106}$$

$$c_{10}\frac{363528\eta^4\gamma_x^2\gamma_y^2 L^2}{\delta(1-\lambda)^2} \le c_5\frac{\alpha_u(1-\lambda)}{30} \ . \tag{107}$$

Then, we solve Eqs. (95-98) first and then solve Eqs. (99-107) after all values of $\{c_5, c_6, c_9, c_{10}\}$ are determined.

To solve Eq. (95), we set

$$c_5 = \frac{12360\eta\gamma_x^3 L^4}{\alpha_u\alpha_x^2\mu^2(1-\lambda)^3} + \frac{15840\gamma_x^3\eta L^2}{\alpha_u\rho_x(1-\lambda)^3 K} + \frac{528000\gamma_x^3\eta L^4}{\alpha_u\rho_y\mu^2(1-\lambda)^3 K} \ . \tag{108}$$

To solve Eq. (96), we set

$$\alpha_u \le \frac{\delta(1-\lambda)}{263} \ . \tag{109}$$

To solve Eq. (97) with $\eta < 1$, we set

$$\gamma_x \le \frac{\alpha_u(1-\lambda)^2}{627L} \ . \tag{110}$$

To solve Eq. (98) with $\eta < 1$, we set

$$\gamma_x \le \frac{\delta(1-\lambda)^2}{10244L} \ . \tag{111}$$

Furthermore, to eliminate $\frac{1}{K}\mathbb{E}\left[\|V_t - \bar{V}_t\|_F^2\right]$, we enforce

$$c_4\frac{3\gamma_y^2\eta}{\alpha_y(1-\lambda)} - c_6\frac{\alpha_v(1-\lambda)}{2} + c_8\frac{36\alpha_v^2}{\delta} + c_{10}\frac{72\gamma_y^2\alpha_v^2\eta^2}{\delta} + C_6\frac{9\gamma_y^2}{2\alpha_y^2}$$

$$\begin{aligned}
&= \frac{4\gamma_x L^2}{\alpha_y(1-\lambda)} \frac{3\gamma_y^2\eta}{\alpha_y(1-\lambda)} + \frac{400\gamma_x L^4}{3\alpha_y\mu^2(1-\lambda)} \frac{3\gamma_y^2\eta}{\alpha_y(1-\lambda)} + C_6 \frac{2}{\alpha_y\eta(1-\lambda)} \frac{3\gamma_y^2\eta}{\alpha_y(1-\lambda)} + \frac{9\gamma_y^2}{2\alpha_y^2} C_6 \\
&\quad - c_6 \frac{\alpha_v(1-\lambda)}{2} + c_8 \frac{36\alpha_v^2}{\delta} + c_{10} \frac{72\gamma_y^2\alpha_v^2\eta^2}{\delta} \\
&= \frac{12\eta\gamma_x\gamma_y^2 L^2}{\alpha_y^2(1-\lambda)^2} + \frac{400\eta\gamma_x\gamma_y^2 L^4}{\alpha_y^2\mu^2(1-\lambda)^2} + C_6 \frac{6\gamma_y^2}{\alpha_y^2(1-\lambda)^2} + \frac{9\gamma_y^2}{2\alpha_y^2} C_6 \\
&\quad - c_6 \frac{\alpha_v(1-\lambda)}{2} + c_8 \frac{36\alpha_v^2}{\delta} + c_{10} \frac{72\gamma_y^2\alpha_v^2\eta^2}{\delta} \\
&\leq \frac{12\eta\gamma_x\gamma_y^2 L^2}{\alpha_y^2(1-\lambda)^2} + \frac{400\eta\gamma_x\gamma_y^2 L^4}{\alpha_y^2\mu^2(1-\lambda)^2} + C_6 \frac{11\gamma_y^2}{\alpha_y^2(1-\lambda)^2} \\
&\quad - c_6 \frac{\alpha_v(1-\lambda)}{2} + c_6 \frac{64\alpha_v}{\delta(1-\lambda)} \frac{36\alpha_v^2}{\delta} + c_{10} \frac{1152\gamma_y^2\alpha_v^2\eta^2}{\delta^2} \frac{36\alpha_v^2}{\delta} + c_{10} \frac{72\gamma_y^2\alpha_v^2\eta^2}{\delta} \\
&\leq \frac{12\eta\gamma_x\gamma_y^2 L^2}{\alpha_y^2(1-\lambda)^2} + \frac{400\eta\gamma_x\gamma_y^2 L^4}{\alpha_y^2\mu^2(1-\lambda)^2} - c_6 \frac{\alpha_v(1-\lambda)}{2} + c_6 \frac{2304\alpha_v^3(1-\delta)}{\delta^2(1-\lambda)} \\
&\quad + c_{10} \frac{41472\gamma_y^2\alpha_v^4\eta^2}{\delta^3} + c_{10} \frac{72\gamma_y^2\alpha_v^2\eta^2}{\delta} \\
&\quad + \frac{528\gamma_x\gamma_y^2\eta L^2}{\rho_x(1-\lambda)^2 K} + \frac{17600\gamma_x\gamma_y^2\eta L^4}{\rho_y\mu^2(1-\lambda)^2 K} + c_{10} \frac{792\eta^2\gamma_y^2}{\delta(1-\lambda)^2} \\
&\quad + c_5 \frac{13068\eta^2\gamma_y^2 L^2}{\alpha_u(1-\lambda)^3} + c_5 \frac{3497472\alpha_u\eta^2\gamma_y^2 L^2}{\delta^2(1-\lambda)^3} \\
&\quad + c_9 \frac{62954496\alpha_u^2\eta^4\gamma_x^2\gamma_y^2 L^2}{\delta^3(1-\lambda)^2} + c_9 \frac{363528\eta^4\gamma_x^2\gamma_y^2 L^2}{\delta(1-\lambda)^2} \\
&\quad + c_6 \frac{13068\eta^2\gamma_y^2 L^2}{\alpha_v(1-\lambda)^3} + c_6 \frac{3497472\alpha_v\eta^2\gamma_y^2 L^2}{\delta^2(1-\lambda)^3} \\
&\quad + c_{10} \frac{62954496\alpha_v^2\eta^4\gamma_y^4 L^2}{\delta^3(1-\lambda)^2} + c_{10} \frac{363528\eta^4\gamma_y^4 L^2}{\delta(1-\lambda)^2} \\
&\leq 0 .
\end{aligned} \tag{112}$$

To solve this inequality, we can set

$$\frac{12\eta\gamma_x\gamma_y^2 L^2}{\alpha_y^2(1-\lambda)^2} + \frac{400\eta\gamma_x\gamma_y^2 L^4}{\alpha_y^2\mu^2(1-\lambda)^2} + \frac{528\gamma_x\gamma_y^2\eta L^2}{\rho_x(1-\lambda)^2 K} + \frac{17600\gamma_x\gamma_y^2\eta L^4}{\rho_y\mu^2(1-\lambda)^2 K} \leq c_6 \frac{\alpha_v(1-\lambda)}{30} , \tag{113}$$

$$c_6 \frac{2304\alpha_v^3}{\delta^2(1-\lambda)} \leq c_6 \frac{\alpha_v(1-\lambda)}{30} , \tag{114}$$

$$c_6 \frac{13068\eta^2\gamma_y^2 L^2}{\alpha_v(1-\lambda)^3} \leq c_6 \frac{\alpha_v(1-\lambda)}{30} , \tag{115}$$

$$c_6 \frac{3497472\alpha_v\eta^2\gamma_y^2 L^2}{\delta^2(1-\lambda)^3} \leq c_6 \frac{\alpha_v(1-\lambda)}{30} , \tag{116}$$

$$c_{10} \frac{41472\gamma_y^2\alpha_v^4\eta^2}{\delta^3} \leq c_6 \frac{\alpha_v(1-\lambda)}{30} , \tag{117}$$

$$c_{10} \frac{72\gamma_y^2\alpha_v^2\eta^2}{\delta} \leq c_6 \frac{\alpha_v(1-\lambda)}{30} , \tag{118}$$

$$c_{10} \frac{792\eta^2\gamma_y^2}{\delta(1-\lambda)^2} \leq c_6 \frac{\alpha_v(1-\lambda)}{30} , \tag{119}$$

$$c_{10} \frac{62954496\alpha_v^2\eta^4\gamma_y^4 L^2}{\delta^3(1-\lambda)^2} \leq c_6 \frac{\alpha_v(1-\lambda)}{30} , \tag{120}$$

$$c_{10} \frac{363528\eta^4\gamma_y^4 L^2}{\delta(1-\lambda)^2} \leq c_6 \frac{\alpha_v(1-\lambda)}{30} \ , \tag{121}$$

$$c_5 \frac{13068\eta^2\gamma_y^2 L^2}{\alpha_u(1-\lambda)^3} \leq c_6 \frac{\alpha_v(1-\lambda)}{30} \ , \tag{122}$$

$$c_5 \frac{3497472\alpha_u\eta^2\gamma_y^2 L^2}{\delta^2(1-\lambda)^3} \leq c_6 \frac{\alpha_v(1-\lambda)}{30} \ , \tag{123}$$

$$c_9 \frac{62954496\alpha_u^2\eta^4\gamma_x^2\gamma_y^2 L^2}{\delta^3(1-\lambda)^2} \leq c_6 \frac{\alpha_v(1-\lambda)}{30} \ , \tag{124}$$

$$c_9 \frac{363528\eta^4\gamma_x^2\gamma_y^2 L^2}{\delta(1-\lambda)^2} \leq c_6 \frac{\alpha_v(1-\lambda)}{30} \ . \tag{125}$$

Similarly, we solve Eqs. (113-116) first and then solve Eqs. (117-125) after all values of $\{c_5, c_6, c_9, c_{10}\}$ are determined. Specifically, to solve Eq. (113), we can set

$$c_6 = \frac{12360\eta\gamma_x\gamma_y^2 L^4}{\alpha_v\alpha_y^2\mu^2(1-\lambda)^3} + \frac{15840\gamma_x\gamma_y^2\eta L^2}{\alpha_v\rho_x(1-\lambda)^3 K} + \frac{528000\gamma_x\gamma_y^2\eta L^4}{\alpha_v\rho_y\mu^2(1-\lambda)^3 K} \ . \tag{126}$$

To solve Eq. (114), we can set

$$\alpha_v \leq \frac{\delta(1-\lambda)}{263} \ . \tag{127}$$

To solve Eq. (115) with $\eta < 1$, we can set

$$\gamma_y \leq \frac{\alpha_v(1-\lambda)^2}{627L} \ . \tag{128}$$

To solve Eq. (116) with $\eta < 1$, we can set

$$\gamma_y \leq \frac{\delta(1-\lambda)^2}{10244L} \ . \tag{129}$$

By now, we have determined all values of $\{c_5, c_6, c_9, c_{10}\}$, which are shown as follows:

$$\begin{aligned}
c_5 &= \frac{12360\eta\gamma_x^3 L^4}{\alpha_u\alpha_x^2\mu^2(1-\lambda)^3} + \frac{15840\eta\gamma_x^3}{\alpha_u(1-\lambda)^3}\Big(\frac{L^2}{\rho_x K} + \frac{100L^4}{3\rho_y\mu^2 K}\Big) \ , \\
c_6 &= \frac{12360\eta\gamma_x\gamma_y^2 L^4}{\alpha_v\alpha_y^2\mu^2(1-\lambda)^3} + \frac{15840\eta\gamma_x\gamma_y^2}{\alpha_v(1-\lambda)^3}\Big(\frac{L^2}{\rho_x K} + \frac{100L^4}{3\rho_y\mu^2 K}\Big) \ , \\
c_9 &= \frac{11000\eta\gamma_x L^4}{\mu^2\delta(1-\lambda)^2} + \frac{8640\alpha_x^2\eta\gamma_x}{\delta(1-\lambda)^2}\Big(\frac{L^2}{\rho_x K} + \frac{100L^4}{3\rho_y\mu^2 K}\Big) \ , \\
c_{10} &= \frac{11000\eta\gamma_x L^4}{\mu^2\delta(1-\lambda)^2} + \frac{8640\alpha_y^2\eta\gamma_x}{\delta(1-\lambda)^2}\Big(\frac{L^2}{\rho_x K} + \frac{100L^4}{3\rho_y\mu^2 K}\Big) \ .
\end{aligned} \tag{130}$$

In what follows, we solve the left Eqs. (69-74), Eqs. (84-89), Eqs. (99-107), and Eqs. (117-125) in terms of the values of $\{c_5, c_6, c_9, c_{10}\}$.

For Eq. (69), we have

$$\begin{aligned}
&c_5 \frac{10692\alpha_x^2\eta^2 L^2}{\alpha_u(1-\lambda)^3}\frac{20}{\delta} \\
&= \frac{2643062400\eta^3\gamma_x^3 L^6}{\alpha_u^2\mu^2\delta(1-\lambda)^6} + \frac{3387225600\alpha_x^2\eta^3\gamma_x^3 L^2}{\alpha_u^2\delta(1-\lambda)^6}\Big(\frac{L^2}{\rho_x K} + \frac{100L^4}{3\rho_y\mu^2 K}\Big) \\
&\leq c_9
\end{aligned}$$

$$= \frac{11000\gamma_x\eta L^4}{\mu^2\delta(1-\lambda)^2} + \frac{8640\gamma_x\alpha_x^2\eta}{\delta(1-\lambda)^2}\left(\frac{L^2}{\rho_x K} + \frac{100L^4}{3\rho_y\mu^2 K}\right). \tag{131}$$

To solve this inequality, we set

$$\frac{2643062400\eta^3\gamma_x^3 L^6}{\alpha_u^2\mu^2\delta(1-\lambda)^6} \leq \frac{11000\gamma_x\eta L^4}{\mu^2\delta(1-\lambda)^2},$$

$$\frac{3387225600\alpha_x^2\eta^3\gamma_x^3 L^2}{\alpha_u^2\delta(1-\lambda)^6} \leq \frac{8640\gamma_x\alpha_x^2\eta}{\delta(1-\lambda)^2}. \tag{132}$$

Then, due to $\eta < 1$, we can obtain

$$\gamma_x \leq \min\left\{\frac{\alpha_u(1-\lambda)^2}{491L}, \frac{\alpha_u(1-\lambda)^2}{627L}\right\}. \tag{133}$$

For Eq. (70), we have

$$c_5 \frac{2861568\alpha_x^2\alpha_u\eta^2 L^2}{\delta^2(1-\lambda)^3}\frac{20}{\delta}$$

$$= \frac{707379609600\eta^3\gamma_x^3 L^6}{\mu^2\delta^3(1-\lambda)^6} + \frac{906544742400\alpha_x^2\eta^3\gamma_x^3 L^2}{\delta^3(1-\lambda)^6}\left(\frac{L^2}{\rho_x K} + \frac{100L^4}{3\rho_y\mu^2 K}\right)$$

$$\leq c_9$$

$$= \frac{11000\gamma_x\eta L^4}{\mu^2\delta(1-\lambda)^2} + \frac{8640\gamma_x\alpha_x^2\eta}{\delta(1-\lambda)^2}\left(\frac{L^2}{\rho_x K} + \frac{100L^4}{3\rho_y\mu^2 K}\right). \tag{134}$$

To solve this inequality, we set

$$\frac{707379609600\eta^3\gamma_x^3 L^6}{\mu^2\delta^3(1-\lambda)^6} \leq \frac{11000\gamma_x\eta L^4}{\mu^2\delta(1-\lambda)^2},$$

$$\frac{906544742400\alpha_x^2\eta^3\gamma_x^3 L^2}{\delta^3(1-\lambda)^6} \leq \frac{8640\gamma_x\alpha_x^2\eta}{\delta(1-\lambda)^2}. \tag{135}$$

Then, due to $\eta < 1$, we can obtain

$$\gamma_x \leq \frac{\delta(1-\lambda)^2}{10244L}. \tag{136}$$

For Eq. (71), we have

$$c_6 \frac{10692\alpha_x^2\eta^2 L^2}{\alpha_v(1-\lambda)^3}\frac{20}{\delta}$$

$$= \frac{2643062400\alpha_x^2\eta^3\gamma_x\gamma_y^2 L^6}{\alpha_v^2\alpha_y^2\mu^2\delta(1-\lambda)^6} + \frac{3387225600\alpha_x^2\eta^3\gamma_x\gamma_y^2 L^2}{\alpha_v^2\delta(1-\lambda)^6}\left(\frac{L^2}{\rho_x K} + \frac{100L^4}{3\rho_y\mu^2 K}\right)$$

$$\leq c_9$$

$$= \frac{11000\gamma_x\eta L^4}{\mu^2\delta(1-\lambda)^2} + \frac{8640\gamma_x\alpha_x^2\eta}{\delta(1-\lambda)^2}\left(\frac{L^2}{\rho_x K} + \frac{100L^4}{3\rho_y\mu^2 K}\right). \tag{137}$$

To solve this inequality, we set

$$\frac{2643062400\alpha_x^2\eta^3\gamma_x\gamma_y^2 L^6}{\alpha_v^2\alpha_y^2\mu^2\delta(1-\lambda)^6} \leq \frac{11000\gamma_x\eta L^4}{\mu^2\delta(1-\lambda)^2},$$

$$\frac{3387225600\alpha_x^2\eta^3\gamma_x\gamma_y^2 L^2}{\alpha_v^2\delta(1-\lambda)^6} \leq \frac{8640\gamma_x\alpha_x^2\eta}{\delta(1-\lambda)^2}. \tag{138}$$

Then, due to $\eta < 1$ and $\alpha_x \eta < 1$, we can obtain

$$\gamma_y \leq \min\left\{\frac{\alpha_v \alpha_y (1-\lambda)^2}{491L}, \frac{\alpha_v (1-\lambda)^2}{627L}\right\} . \tag{139}$$

For Eq. (72), we have

$$c_6 \frac{2861568\alpha_x^2 \alpha_v \eta^2 L^2}{\delta^2(1-\lambda)^3} \frac{20}{\delta}$$

$$= \frac{707379609600\alpha_x^2 \eta^3 \gamma_x \gamma_y^2 L^6}{\alpha_y^2 \mu^2 \delta^3(1-\lambda)^6} + \frac{906544742400\alpha_x^2 \eta^3 \gamma_x \gamma_y^2 L^2}{\delta^3(1-\lambda)^6}\left(\frac{L^2}{\rho_x K} + \frac{100L^4}{3\rho_y \mu^2 K}\right)$$

$$\leq c_9$$

$$= \frac{11000\gamma_x \eta L^4}{\mu^2 \delta(1-\lambda)^2} + \frac{8640\gamma_x \alpha_x^2 \eta}{\delta(1-\lambda)^2}\left(\frac{L^2}{\rho_x K} + \frac{100L^4}{3\rho_y \mu^2 K}\right) . \tag{140}$$

To solve this inequality, we set

$$\frac{707379609600\alpha_x^2 \eta^3 \gamma_x \gamma_y^2 L^6}{\alpha_y^2 \mu^2 \delta^3(1-\lambda)^6} \leq \frac{11000\gamma_x \eta L^4}{\mu^2 \delta(1-\lambda)^2} ,$$

$$\frac{906544742400\alpha_x^2 \eta^3 \gamma_x \gamma_y^2 L^2}{\delta^3(1-\lambda)^6} \leq \frac{8640\gamma_x \alpha_x^2 \eta}{\delta(1-\lambda)^2} . \tag{141}$$

Then, due to $\eta < 1$ and $\alpha_x \eta < 1$, we can obtain

$$\gamma_y \leq \min\left\{\frac{\alpha_y \delta(1-\lambda)^2}{8020L}, \frac{\delta(1-\lambda)^2}{10244L}\right\} . \tag{142}$$

For Eq. (73), we have

$$c_{10} \frac{51508224\alpha_x^2 \gamma_y^2 \alpha_v^2 \eta^4 L^2}{\delta^3(1-\lambda)^2} \frac{20}{\delta}$$

$$= \frac{11331809280000\alpha_v^2 \alpha_x^2 \eta^5 \gamma_x \gamma_y^2 L^6}{\mu^2 \delta^5(1-\lambda)^4}$$

$$+ \frac{8900621107200\alpha_v^2 \alpha_x^2 \alpha_y^2 \eta^5 \gamma_x \gamma_y^2 L^2}{\delta^5(1-\lambda)^4}\left(\frac{L^2}{\rho_x K} + \frac{100L^4}{3\rho_y \mu^2 K}\right)$$

$$\leq c_9$$

$$= \frac{11000\gamma_x \eta L^4}{\mu^2 \delta(1-\lambda)^2} + \frac{8640\gamma_x \alpha_x^2 \eta}{\delta(1-\lambda)^2}\left(\frac{L^2}{\rho_x K} + \frac{100L^4}{3\rho_y \mu^2 K}\right) . \tag{143}$$

To solve this inequality, we set

$$\frac{11331809280000\alpha_v^2 \alpha_x^2 \eta^5 \gamma_x \gamma_y^2 L^6}{\mu^2 \delta^5(1-\lambda)^4} \leq \frac{11000\gamma_x \eta L^4}{\mu^2 \delta(1-\lambda)^2} ,$$

$$\frac{8900621107200\alpha_v^2 \alpha_x^2 \alpha_y^2 \eta^5 \gamma_x \gamma_y^2 L^2}{\delta^5(1-\lambda)^4} \leq \frac{8640\gamma_x \alpha_x^2 \eta}{\delta(1-\lambda)^2} . \tag{144}$$

Then, due to $\eta < 1$, $\alpha_v < 1$, $\alpha_x \eta < 1$, $\alpha_y \eta < 1$, we can obtain

$$\gamma_y \leq \frac{\delta^2(1-\lambda)}{32097L} . \tag{145}$$

For Eq. (74), we have

$$c_{10} \frac{297432\alpha_x^2 \gamma_y^2 \eta^4 L^2}{\delta(1-\lambda)^2} \frac{20}{\delta}$$

$$
= \frac{65435040000\alpha_x^2\eta^5\gamma_x\gamma_y^2 L^6}{\mu^2\delta^3(1-\lambda)^4} + \frac{51396249600\alpha_x^2\alpha_y^2\eta^5\gamma_x\gamma_y^2 L^2}{\delta^3(1-\lambda)^4}\left(\frac{L^2}{\rho_x K} + \frac{100L^4}{3\rho_y\mu^2 K}\right)
$$

$$
\leq c_9
$$

$$
= \frac{11000\eta\gamma_x L^4}{\mu^2\delta(1-\lambda)^2} + \frac{8640\alpha_x^2\eta\gamma_x}{\delta(1-\lambda)^2}\left(\frac{L^2}{\rho_x K} + \frac{100L^4}{3\rho_y\mu^2 K}\right). \tag{146}
$$

To solve this inequality, we set

$$
\frac{65435040000\alpha_x^2\eta^5\gamma_x\gamma_y^2 L^6}{\mu^2\delta^3(1-\lambda)^4} \leq \frac{11000\eta\gamma_x L^4}{\mu^2\delta(1-\lambda)^2} ,
$$
$$
\frac{51396249600\alpha_x^2\alpha_y^2\eta^5\gamma_x\gamma_y^2 L^2}{\delta^3(1-\lambda)^4} \leq \frac{8640\alpha_x^2\eta\gamma_x}{\delta(1-\lambda)^2} . \tag{147}
$$

Then, due to $\eta < 1$, $\alpha_x\eta < 1$ and $\alpha_y\eta < 1$, we can obtain

$$
\gamma_y \leq \frac{\delta(1-\lambda)}{2439L} . \tag{148}
$$

For Eq. (84), we have

$$
c_5\frac{10692\alpha_y^2\eta^2 L^2}{\alpha_u(1-\lambda)^3}\frac{20}{\delta}
$$

$$
= \frac{2643062400\alpha_y^2\eta^3\gamma_x^3 L^6}{\alpha_u^2\alpha_x^2\mu^2\delta(1-\lambda)^6} + \frac{3387225600\alpha_y^2\eta^3\gamma_x^3 L^2}{\alpha_u^2\delta(1-\lambda)^6}\left(\frac{L^2}{\rho_x K} + \frac{100L^4}{3\rho_y\mu^2 K}\right)
$$

$$
\leq c_{10}
$$

$$
= \frac{11000\eta\gamma_x L^4}{\mu^2\delta(1-\lambda)^2} + \frac{8640\alpha_y^2\eta\gamma_x}{\delta(1-\lambda)^2}\left(\frac{L^2}{\rho_x K} + \frac{100L^4}{3\rho_y\mu^2 K}\right). \tag{149}
$$

To solve this inequality, we set

$$
\frac{2643062400\alpha_y^2\eta^3\gamma_x^3 L^6}{\alpha_u^2\alpha_x^2\mu^2\delta(1-\lambda)^6} \leq \frac{11000\eta\gamma_x L^4}{\mu^2\delta(1-\lambda)^2} ,
$$
$$
\frac{3387225600\alpha_y^2\eta^3\gamma_x^3 L^2}{\alpha_u^2\delta(1-\lambda)^6} \leq \frac{8640\alpha_y^2\eta\gamma_x}{\delta(1-\lambda)^2} . \tag{150}
$$

Then, due to $\eta < 1$ and $\alpha_y\eta < 1$, we can obtain

$$
\gamma_x \leq \min\left\{\frac{\alpha_u\alpha_x(1-\lambda)^2}{491L}, \frac{\alpha_u(1-\lambda)^2}{627L}\right\} . \tag{151}
$$

For Eq. (85), we have

$$
c_5\frac{2861568\alpha_y^2\alpha_u\eta^2 L^2}{\delta^2(1-\lambda)^3}\frac{20}{\delta}
$$

$$
= \frac{707379609600\alpha_y^2\eta^3\gamma_x^3 L^6}{\alpha_x^2\mu^2\delta^3(1-\lambda)^6} + \frac{906544742400\alpha_y^2\eta^3\gamma_x^3 L^2}{\delta^3(1-\lambda)^6}\left(\frac{L^2}{\rho_x K} + \frac{100L^4}{3\rho_y\mu^2 K}\right)
$$

$$
\leq c_{10}
$$

$$
= \frac{11000\eta\gamma_x L^4}{\mu^2\delta(1-\lambda)^2} + \frac{8640\alpha_y^2\eta\gamma_x}{\delta(1-\lambda)^2}\left(\frac{L^2}{\rho_x K} + \frac{100L^4}{3\rho_y\mu^2 K}\right). \tag{152}
$$

To solve this inequality, we set

$$
\frac{707379609600\alpha_y^2\eta^3\gamma_x^3 L^6}{\alpha_x^2\mu^2\delta^3(1-\lambda)^6} \leq \frac{11000\eta\gamma_x L^4}{\mu^2\delta(1-\lambda)^2} ,
$$

$$\frac{906544742400\alpha_y^2\eta^3\gamma_x^3L^2}{\delta^3(1-\lambda)^6} \leq \frac{8640\alpha_y^2\eta\gamma_x}{\delta(1-\lambda)^2} \ . \tag{153}$$

Then, due to $\eta < 1$ and $\alpha_y\eta < 1$, we can obtain

$$\gamma_x \leq \min\left\{\frac{\alpha_x\delta(1-\lambda)^2}{8020L}, \frac{\delta(1-\lambda)^2}{10244L}\right\} \ . \tag{154}$$

For Eq. (86), we have

$$
\begin{aligned}
& c_9 \frac{51508224\alpha_y^2\gamma_x^2\alpha_u^2\eta^4L^2}{\delta^3(1-\lambda)^2}\frac{20}{\delta} \\
& = \frac{11331809280000\alpha_u^2\alpha_y^2\eta^5\gamma_x^3L^6}{\mu^2\delta^5(1-\lambda)^4} + \frac{8900621107200\alpha_u^2\alpha_x^2\alpha_y^2\eta^5\gamma_x^3L^2}{\delta^5(1-\lambda)^4}\left(\frac{L^2}{\rho_xK} + \frac{100L^4}{3\rho_y\mu^2K}\right) \\
& \leq c_{10} \\
& = \frac{11000\eta\gamma_xL^4}{\mu^2\delta(1-\lambda)^2} + \frac{8640\alpha_y^2\eta\gamma_x}{\delta(1-\lambda)^2}\left(\frac{L^2}{\rho_xK} + \frac{100L^4}{3\rho_y\mu^2K}\right) \ .
\end{aligned}
\tag{155}
$$

To solve this inequality, we set

$$
\begin{aligned}
\frac{11331809280000\alpha_u^2\alpha_y^2\eta^5\gamma_x^3L^6}{\mu^2\delta^5(1-\lambda)^4} &\leq \frac{11000\eta\gamma_xL^4}{\mu^2\delta(1-\lambda)^2} \ , \\
\frac{8900621107200\alpha_u^2\alpha_x^2\alpha_y^2\eta^5\gamma_x^3L^2}{\delta^5(1-\lambda)^4} &\leq \frac{8640\alpha_y^2\eta\gamma_x}{\delta(1-\lambda)^2} \ .
\end{aligned}
\tag{156}
$$

Then, due to $\eta < 1$, $\alpha_u < 1$, $\alpha_x\eta < 1$ and $\alpha_y\eta < 1$, we can obtain

$$\gamma_x \leq \frac{\delta^2(1-\lambda)}{32097L} \ . \tag{157}$$

For Eq. (87), we have

$$
\begin{aligned}
& c_9 \frac{297432\alpha_y^2\gamma_x^2\eta^4L^2}{\delta(1-\lambda)^2}\frac{20}{\delta} \\
& = \frac{65435040000\alpha_y^2\eta^5\gamma_x^3L^6}{\mu^2\delta^3(1-\lambda)^4} + \frac{51396249600\alpha_x^2\alpha_y^2\eta^5\gamma_x^3L^2}{\delta^3(1-\lambda)^4}\left(\frac{L^2}{\rho_xK} + \frac{100L^4}{3\rho_y\mu^2K}\right) \\
& \leq c_{10} \\
& = \frac{11000\eta\gamma_xL^4}{\mu^2\delta(1-\lambda)^2} + \frac{8640\alpha_y^2\eta\gamma_x}{\delta(1-\lambda)^2}\left(\frac{L^2}{\rho_xK} + \frac{100L^4}{3\rho_y\mu^2K}\right) \ .
\end{aligned}
\tag{158}
$$

To solve this inequality, we set

$$
\begin{aligned}
\frac{65435040000\alpha_y^2\eta^5\gamma_x^3L^6}{\mu^2\delta^3(1-\lambda)^4} &\leq \frac{11000\eta\gamma_xL^4}{\mu^2\delta(1-\lambda)^2} \ , \\
\frac{51396249600\alpha_x^2\alpha_y^2\eta^5\gamma_x^3L^2}{\delta^3(1-\lambda)^4} &\leq \frac{8640\alpha_y^2\eta\gamma_x}{\delta(1-\lambda)^2} \ .
\end{aligned}
\tag{159}
$$

Then, due to $\eta < 1$, $\alpha_x\eta < 1$ and $\alpha_y\eta < 1$, we can obtain

$$\gamma_x \leq \frac{\delta(1-\lambda)}{2439L} \ . \tag{160}$$

For Eq. (88), we have

$$c_6 \frac{10692\alpha_y^2\eta^2L^2}{\alpha_v(1-\lambda)^3}\frac{20}{\delta}$$

$$= \frac{2643062400\eta^3\gamma_x\gamma_y^2 L^6}{\alpha_v^2\mu^2\delta(1-\lambda)^6} + \frac{3387225600\alpha_y^2\eta^3\gamma_x\gamma_y^2 L^2}{\alpha_v^2\delta(1-\lambda)^6}\left(\frac{L^2}{\rho_x K} + \frac{100L^4}{3\rho_y\mu^2 K}\right)$$

$$\leq c_{10}$$

$$= \frac{11000\eta\gamma_x L^4}{\mu^2\delta(1-\lambda)^2} + \frac{8640\alpha_y^2\eta\gamma_x}{\delta(1-\lambda)^2}\left(\frac{L^2}{\rho_x K} + \frac{100L^4}{3\rho_y\mu^2 K}\right). \tag{161}$$

To solve this inequality, we set

$$\frac{2643062400\eta^3\gamma_x\gamma_y^2 L^6}{\alpha_v^2\mu^2\delta(1-\lambda)^6} \leq \frac{11000\eta\gamma_x L^4}{\mu^2\delta(1-\lambda)^2},$$

$$\frac{3387225600\alpha_y^2\eta^3\gamma_x\gamma_y^2 L^2}{\alpha_v^2\delta(1-\lambda)^6} \leq \frac{8640\alpha_y^2\eta\gamma_x}{\delta(1-\lambda)^2}. \tag{162}$$

Then, due to $\eta < 1$, we can obtain

$$\gamma_y \leq \min\left\{\frac{\alpha_v(1-\lambda)^2}{491L}, \frac{\alpha_v(1-\lambda)^2}{627L}\right\}. \tag{163}$$

For Eq. (89), we have

$$c_6 \frac{2861568\alpha_y^2\alpha_v\eta^2 L^2}{\delta^2(1-\lambda)^3}\frac{20}{\delta}$$

$$= \frac{707379609600\eta^3\gamma_x\gamma_y^2 L^6}{\mu^2\delta^3(1-\lambda)^6} + \frac{906544742400\alpha_y^2\eta^3\gamma_x\gamma_y^2 L^2}{\delta^3(1-\lambda)^6}\left(\frac{L^2}{\rho_x K} + \frac{100L^4}{3\rho_y\mu^2 K}\right)$$

$$\leq c_{10}$$

$$= \frac{11000\eta\gamma_x L^4}{\mu^2\delta(1-\lambda)^2} + \frac{8640\alpha_y^2\eta\gamma_x}{\delta(1-\lambda)^2}\left(\frac{L^2}{\rho_x K} + \frac{100L^4}{3\rho_y\mu^2 K}\right). \tag{164}$$

To solve this inequality, we set

$$\frac{707379609600\eta^3\gamma_x\gamma_y^2 L^6}{\mu^2\delta^3(1-\lambda)^6} \leq \frac{11000\eta\gamma_x L^4}{\mu^2\delta(1-\lambda)^2},$$

$$\frac{906544742400\alpha_y^2\eta^3\gamma_x\gamma_y^2 L^2}{\delta^3(1-\lambda)^6} \leq \frac{8640\alpha_y^2\eta\gamma_x}{\delta(1-\lambda)^2}. \tag{165}$$

Then, due to $\eta < 1$, we can obtain

$$\gamma_y \leq \frac{\delta(1-\lambda)^2}{10244L}. \tag{166}$$

For Eq. (99), we have

$$c_9 \frac{41472\gamma_x^2\alpha_u^4\eta^2}{\delta^3}\frac{30}{\alpha_u(1-\lambda)}$$

$$= \frac{13685760000\alpha_u^3\eta^3\gamma_x^3 L^4}{\mu^2\delta^4(1-\lambda)^3} + \frac{10749542400\alpha_u^3\alpha_x^2\eta^3\gamma_x^3}{\delta^4(1-\lambda)^3}\left(\frac{L^2}{\rho_x K} + \frac{100L^4}{3\rho_y\mu^2 K}\right)$$

$$\leq c_5$$

$$= \frac{12360\eta\gamma_x^3 L^4}{\alpha_u\alpha_x^2\mu^2(1-\lambda)^3} + \frac{15840\eta\gamma_x^3}{\alpha_u(1-\lambda)^3}\left(\frac{L^2}{\rho_x K} + \frac{100L^4}{3\rho_y\mu^2 K}\right). \tag{167}$$

To solve this inequality, we set

$$\frac{13685760000\alpha_u^3\eta^3\gamma_x^3 L^4}{\mu^2\delta^4(1-\lambda)^3} \leq \frac{12360\eta\gamma_x^3 L^4}{\alpha_u\alpha_x^2\mu^2(1-\lambda)^3},$$

$$\frac{10749542400\alpha_u^3\alpha_x^2\eta^3\gamma_x^3}{\delta^4(1-\lambda)^3} \leq \frac{15840\eta\gamma_x^3}{\alpha_u(1-\lambda)^3} \ . \tag{168}$$

Then, due to $\eta < 1$ and $\alpha_u < 1$, we can obtain

$$\alpha_x \leq \frac{\delta^2}{1053} \ . \tag{169}$$

For Eq. (100), we have

$$
\begin{aligned}
& c_9 \frac{72\gamma_x^2\alpha_u^2\eta^2}{\delta}\frac{30}{\alpha_u(1-\lambda)} \\
&= \frac{23760000\alpha_u\eta^3\gamma_x^3L^4}{\mu^2\delta^2(1-\lambda)^3} + \frac{18662400\alpha_u\alpha_x^2\eta^3\gamma_x^3}{\delta^2(1-\lambda)^3}\Big(\frac{L^2}{\rho_x K} + \frac{100L^4}{3\rho_y\mu^2 K}\Big) \\
&\leq c_5 \\
&= \frac{12360\eta\gamma_x^3L^4}{\alpha_u\alpha_x^2\mu^2(1-\lambda)^3} + \frac{15840\eta\gamma_x^3}{\alpha_u(1-\lambda)^3}\Big(\frac{L^2}{\rho_x K} + \frac{100L^4}{3\rho_y\mu^2 K}\Big) \ .
\end{aligned}
\tag{170}
$$

To solve this inequality, we set

$$
\begin{aligned}
\frac{23760000\alpha_u\eta^3\gamma_x^3L^4}{\mu^2\delta^2(1-\lambda)^3} &\leq \frac{12360\eta\gamma_x^3L^4}{\alpha_u\alpha_x^2\mu^2(1-\lambda)^3} \ , \\
\frac{18662400\alpha_u\alpha_x^2\eta^3\gamma_x^3}{\delta^2(1-\lambda)^3} &\leq \frac{15840\eta\gamma_x^3}{\alpha_u(1-\lambda)^3} \ .
\end{aligned}
\tag{171}
$$

Then, due to $\alpha_u < 1$ and $\eta < 1$, we can obtain

$$\alpha_x \leq \frac{\delta}{44} \ . \tag{172}$$

For Eq. (101), we have

$$
\begin{aligned}
& c_9 \frac{792\eta^2\gamma_x^2}{\delta(1-\lambda)^2}\frac{30}{\alpha_u(1-\lambda)} \\
&= \frac{261360000\eta^3\gamma_x^3L^4}{\alpha_u\mu^2\delta^2(1-\lambda)^5} + \frac{205286400\alpha_x^2\eta^3\gamma_x^3}{\alpha_u\delta^2(1-\lambda)^5}\Big(\frac{L^2}{\rho_x K} + \frac{100L^4}{3\rho_y\mu^2 K}\Big) \\
&\leq c_5 \\
&= \frac{12360\eta\gamma_x^3L^4}{\alpha_u\alpha_x^2\mu^2(1-\lambda)^3} + \frac{15840\eta\gamma_x^3}{\alpha_u(1-\lambda)^3}\Big(\frac{L^2}{\rho_x K} + \frac{100L^4}{3\rho_y\mu^2 K}\Big) \ .
\end{aligned}
\tag{173}
$$

To solve this inequality, we set

$$
\begin{aligned}
\frac{261360000\eta^3\gamma_x^3L^4}{\alpha_u\mu^2\delta^2(1-\lambda)^5} &\leq \frac{12360\eta\gamma_x^3L^4}{\alpha_u\alpha_x^2\mu^2(1-\lambda)^3} \ , \\
\frac{205286400\alpha_x^2\eta^3\gamma_x^3}{\alpha_u\delta^2(1-\lambda)^5} &\leq \frac{15840\eta\gamma_x^3}{\alpha_u(1-\lambda)^3} \ .
\end{aligned}
\tag{174}
$$

Then, due to $\eta < 1$, we can obtain

$$\alpha_x \leq \frac{\delta(1-\lambda)}{146} \ . \tag{175}$$

For Eq. (102), we have

$$
c_9 \frac{62954496\alpha_u^2\eta^4\gamma_x^4L^2}{\delta^3(1-\lambda)^2}\frac{30}{\alpha_u(1-\lambda)}
$$

$$
= \frac{20774983680000\alpha_u\eta^5\gamma_x^5 L^6}{\mu^2\delta^4(1-\lambda)^5} + \frac{16317805363200\alpha_u\alpha_x^2\eta^5\gamma_x^5 L^2}{\delta^4(1-\lambda)^5}\Big(\frac{L^2}{\rho_x K} + \frac{100L^4}{3\rho_y\mu^2 K}\Big)
$$

$$
\leq c_5
$$

$$
= \frac{12360\eta\gamma_x^3 L^4}{\alpha_u\alpha_x^2\mu^2(1-\lambda)^3} + \frac{15840\eta\gamma_x^3}{\alpha_u(1-\lambda)^3}\Big(\frac{L^2}{\rho_x K} + \frac{100L^4}{3\rho_y\mu^2 K}\Big). \tag{176}
$$

To solve this inequality, we set

$$
\frac{20774983680000\alpha_u\eta^5\gamma_x^5 L^6}{\mu^2\delta^4(1-\lambda)^5} \leq \frac{12360\eta\gamma_x^3 L^4}{\alpha_u\alpha_x^2\mu^2(1-\lambda)^3} ,
$$

$$
\frac{16317805363200\alpha_u\alpha_x^2\eta^5\gamma_x^5 L^2}{\delta^4(1-\lambda)^5} \leq \frac{15840\eta\gamma_x^3}{\alpha_u(1-\lambda)^3} . \tag{177}
$$

Then, due to $\eta < 1$, $\alpha_u < 1$ and $\alpha_x\eta < 1$, we can obtain

$$
\gamma_x \leq \frac{\delta^2(1-\lambda)}{40998L} . \tag{178}
$$

For Eq. (103), we have

$$
c_9\frac{363528\eta^4\gamma_x^4 L^2}{\delta(1-\lambda)^2}\frac{30}{\alpha_u(1-\lambda)}
$$

$$
= \frac{119964240000\eta^5\gamma_x^5 L^6}{\alpha_u\mu^2\delta^2(1-\lambda)^5} + \frac{94226457600\alpha_x^2\eta^5\gamma_x^5 L^2}{\alpha_u\delta^2(1-\lambda)^5}\Big(\frac{L^2}{\rho_x K} + \frac{100L^4}{3\rho_y\mu^2 K}\Big)
$$

$$
\leq c_5
$$

$$
= \frac{12360\eta\gamma_x^3 L^4}{\alpha_u\alpha_x^2\mu^2(1-\lambda)^3} + \frac{15840\eta\gamma_x^3}{\alpha_u(1-\lambda)^3}\Big(\frac{L^2}{\rho_x K} + \frac{100L^4}{3\rho_y\mu^2 K}\Big). \tag{179}
$$

To solve this inequality, we set

$$
\frac{119964240000\eta^5\gamma_x^5 L^6}{\alpha_u\mu^2\delta^2(1-\lambda)^5} \leq \frac{12360\eta\gamma_x^3 L^4}{\alpha_u\alpha_x^2\mu^2(1-\lambda)^3}
$$

$$
\frac{94226457600\alpha_x^2\eta^5\gamma_x^5 L^2}{\alpha_u\delta^2(1-\lambda)^5} \leq \frac{15840\eta\gamma_x^3}{\alpha_u(1-\lambda)^3} . \tag{180}
$$

Then, due to $\eta < 1$ and $\alpha_x\eta < 1$, we can obtain

$$
\gamma_x \leq \frac{\delta(1-\lambda)}{3116L} . \tag{181}
$$

For Eq. (104), we have

$$
c_6\frac{13068\eta^2\gamma_x^2 L^2}{\alpha_v(1-\lambda)^3}\frac{30}{\alpha_u(1-\lambda)}
$$

$$
= \frac{4845614400\eta^3\gamma_x^3\gamma_y^2 L^6}{\alpha_u\alpha_v^2\alpha_y^2\mu^2(1-\lambda)^7} + \frac{6209913600\eta^3\gamma_x^3\gamma_y^2 L^2}{\alpha_u\alpha_v^2(1-\lambda)^7}\Big(\frac{L^2}{\rho_x K} + \frac{100L^4}{3\rho_y\mu^2 K}\Big)
$$

$$
\leq c_5
$$

$$
= \frac{12360\eta\gamma_x^3 L^4}{\alpha_u\alpha_x^2\mu^2(1-\lambda)^3} + \frac{15840\eta\gamma_x^3}{\alpha_u(1-\lambda)^3}\Big(\frac{L^2}{\rho_x K} + \frac{100L^4}{3\rho_y\mu^2 K}\Big). \tag{182}
$$

To solve this inequality, we set

$$
\frac{4845614400\eta^3\gamma_x^3\gamma_y^2 L^6}{\alpha_u\alpha_v^2\alpha_y^2\mu^2(1-\lambda)^7} \leq \frac{12360\eta\gamma_x^3 L^4}{\alpha_u\alpha_x^2\mu^2(1-\lambda)^3} ,
$$

$$\frac{6209913600\eta^3\gamma_x^3\gamma_y^2L^2}{\alpha_u\alpha_v^2(1-\lambda)^7} \leq \frac{15840\eta\gamma_x^3}{\alpha_u(1-\lambda)^3} \ . \tag{183}$$

Then, due to $\eta < 1$ and $\alpha_x\eta < 1$, we can obtain

$$\gamma_y \leq \min\left\{\frac{\alpha_v\alpha_y(1-\lambda)^2}{627L}, \frac{\alpha_v(1-\lambda)^2}{627L}\right\} \ . \tag{184}$$

For Eq. (105), we have

$$c_6\frac{3497472\alpha_v\eta^2\gamma_x^2L^2}{\delta^2(1-\lambda)^3}\frac{30}{\alpha_u(1-\lambda)}$$

$$= \frac{1296862617600\eta^3\gamma_x^3\gamma_y^2L^6}{\alpha_u\alpha_y^2\mu^2\delta^2(1-\lambda)^7} + \frac{1661998694400\eta^3\gamma_x^3\gamma_y^2L^2}{\alpha_u\delta^2(1-\lambda)^7}\left(\frac{L^2}{\rho_xK} + \frac{100L^4}{3\rho_y\mu^2K}\right)$$

$$\leq c_5$$

$$= \frac{12360\eta\gamma_x^3L^4}{\alpha_u\alpha_x^2\mu^2(1-\lambda)^3} + \frac{15840\eta\gamma_x^3}{\alpha_u(1-\lambda)^3}\left(\frac{L^2}{\rho_xK} + \frac{100L^4}{3\rho_y\mu^2K}\right) \ . \tag{185}$$

To solve this inequality, we set

$$\frac{1296862617600\eta^3\gamma_x^3\gamma_y^2L^6}{\alpha_u\alpha_y^2\mu^2\delta^2(1-\lambda)^7} \leq \frac{12360\eta\gamma_x^3L^4}{\alpha_u\alpha_x^2\mu^2(1-\lambda)^3} \ ,$$

$$\frac{1661998694400\eta^3\gamma_x^3\gamma_y^2L^2}{\alpha_u\delta^2(1-\lambda)^7} \leq \frac{15840\eta\gamma_x^3}{\alpha_u(1-\lambda)^3} \ . \tag{186}$$

Then, due to $\eta < 1$ and $\alpha_x\eta < 1$, we can obtain

$$\gamma_y \leq \min\left\{\frac{\alpha_y\delta(1-\lambda)^2}{10244L}, \frac{\delta(1-\lambda)^2}{10244L}\right\} \ . \tag{187}$$

For Eq. (106), we have

$$c_{10}\frac{62954496\alpha_v^2\eta^4\gamma_x^2\gamma_y^2L^2}{\delta^3(1-\lambda)^2}\frac{30}{\alpha_u(1-\lambda)}$$

$$= \frac{20774983680000\alpha_v^2\eta^5\gamma_x^3\gamma_y^2L^6}{\alpha_u\mu^2\delta^4(1-\lambda)^5} + \frac{16317805363200\alpha_v^2\alpha_y^2\eta^5\gamma_x^3\gamma_y^2L^2}{\alpha_u\delta^4(1-\lambda)^5}\left(\frac{L^2}{\rho_xK} + \frac{100L^4}{3\rho_y\mu^2K}\right)$$

$$\leq c_5$$

$$= \frac{12360\eta\gamma_x^3L^4}{\alpha_u\alpha_x^2\mu^2(1-\lambda)^3} + \frac{15840\eta\gamma_x^3}{\alpha_u(1-\lambda)^3}\left(\frac{L^2}{\rho_xK} + \frac{100L^4}{3\rho_y\mu^2K}\right) \ . \tag{188}$$

To solve this inequality, we set

$$\frac{20774983680000\alpha_v^2\eta^5\gamma_x^3\gamma_y^2L^6}{\alpha_u\mu^2\delta^4(1-\lambda)^5} \leq \frac{12360\eta\gamma_x^3L^4}{\alpha_u\alpha_x^2\mu^2(1-\lambda)^3} \ ,$$

$$\frac{16317805363200\alpha_v^2\alpha_y^2\eta^5\gamma_x^3\gamma_y^2L^2}{\alpha_u\delta^4(1-\lambda)^5} \leq \frac{15840\eta\gamma_x^3}{\alpha_u(1-\lambda)^3} \ . \tag{189}$$

Then, due to $\eta < 1$, $\alpha_v < 1$, $\alpha_x\eta < 1$ and $\alpha_y\eta < 1$, we can obtain

$$\gamma_y \leq \frac{\delta^2(1-\lambda)}{40998L} \ . \tag{190}$$

For Eq. (107), we have

$$c_{10}\frac{363528\eta^4\gamma_x^2\gamma_y^2L^2}{\delta(1-\lambda)^2}\frac{30}{\alpha_u(1-\lambda)}$$

$$
= \frac{119964240000\eta^5\gamma_x^3\gamma_y^2 L^6}{\alpha_u\mu^2\delta^2(1-\lambda)^5} + \frac{94226457600\alpha_y^2\eta^5\gamma_x^3\gamma_y^2 L^2}{\alpha_u\delta^2(1-\lambda)^5}\left(\frac{L^2}{\rho_x K} + \frac{100L^4}{3\rho_y\mu^2 K}\right)
$$
$$
\leq c_5
$$
$$
= \frac{12360\eta\gamma_x^3 L^4}{\alpha_u\alpha_x^2\mu^2(1-\lambda)^3} + \frac{15840\eta\gamma_x^3}{\alpha_u(1-\lambda)^3}\left(\frac{L^2}{\rho_x K} + \frac{100L^4}{3\rho_y\mu^2 K}\right). \tag{191}
$$

To solve this inequality, we set

$$
\frac{119964240000\eta^5\gamma_x^3\gamma_y^2 L^6}{\alpha_u\mu^2\delta^2(1-\lambda)^5} \leq \frac{12360\eta\gamma_x^3 L^4}{\alpha_u\alpha_x^2\mu^2(1-\lambda)^3},
$$
$$
\frac{94226457600\alpha_y^2\eta^5\gamma_x^3\gamma_y^2 L^2}{\alpha_u\delta^2(1-\lambda)^5} \leq \frac{15840\eta\gamma_x^3}{\alpha_u(1-\lambda)^3}. \tag{192}
$$

Then, due to $\eta < 1$, $\alpha_x\eta < 1$ and $\alpha_y\eta < 1$, we can obtain

$$
\gamma_y \leq \frac{\delta(1-\lambda)}{3116L}. \tag{193}
$$

For Eq. (117), we have

$$
c_{10}\frac{41472\gamma_y^2\alpha_v^4\eta^2}{\delta^3}\frac{30}{\alpha_v(1-\lambda)}
$$
$$
= \frac{13685760000\alpha_v^3\eta^3\gamma_x\gamma_y^2 L^4}{\mu^2\delta^4(1-\lambda)^3} + \frac{10749542400\alpha_v^3\alpha_y^2\eta^3\gamma_x\gamma_y^2}{\delta^4(1-\lambda)^3}\left(\frac{L^2}{\rho_x K} + \frac{100L^4}{3\rho_y\mu^2 K}\right)
$$
$$
\leq c_6
$$
$$
= \frac{12360\eta\gamma_x\gamma_y^2 L^4}{\alpha_v\alpha_y^2\mu^2(1-\lambda)^3} + \frac{15840\eta\gamma_x\gamma_y^2}{\alpha_v(1-\lambda)^3}\left(\frac{L^2}{\rho_x K} + \frac{100L^4}{3\rho_y\mu^2 K}\right). \tag{194}
$$

To solve this inequality, we set

$$
\frac{13685760000\alpha_v^3\eta^3\gamma_x\gamma_y^2 L^4}{\mu^2\delta^4(1-\lambda)^3} \leq \frac{12360\eta\gamma_x\gamma_y^2 L^4}{\alpha_v\alpha_y^2\mu^2(1-\lambda)^3},
$$
$$
\frac{10749542400\alpha_v^3\alpha_y^2\eta^3\gamma_x\gamma_y^2}{\delta^4(1-\lambda)^3} \leq \frac{15840\eta\gamma_x\gamma_y^2}{\alpha_v(1-\lambda)^3}. \tag{195}
$$

Then, due to $\eta < 1$ and $\alpha_v < 1$, we can obtain

$$
\alpha_y \leq \frac{\delta^2}{1053}. \tag{196}
$$

For Eq. (118), we have

$$
c_{10}\frac{72\gamma_y^2\alpha_v^2\eta^2}{\delta}\frac{30}{\alpha_v(1-\lambda)}
$$
$$
= \frac{23760000\alpha_v\eta^3\gamma_x\gamma_y^2 L^4}{\mu^2\delta^2(1-\lambda)^3} + \frac{18662400\alpha_v\alpha_y^2\eta^3\gamma_x\gamma_y^2}{\delta^2(1-\lambda)^3}\left(\frac{L^2}{\rho_x K} + \frac{100L^4}{3\rho_y\mu^2 K}\right)
$$
$$
\leq c_6
$$
$$
= \frac{12360\eta\gamma_x\gamma_y^2 L^4}{\alpha_v\alpha_y^2\mu^2(1-\lambda)^3} + \frac{15840\eta\gamma_x\gamma_y^2}{\alpha_v(1-\lambda)^3}\left(\frac{L^2}{\rho_x K} + \frac{100L^4}{3\rho_y\mu^2 K}\right). \tag{197}
$$

To solve this inequality, we set

$$
\frac{23760000\alpha_v\eta^3\gamma_x\gamma_y^2 L^4}{\mu^2\delta^2(1-\lambda)^3} \leq \frac{12360\eta\gamma_x\gamma_y^2 L^4}{\alpha_v\alpha_y^2\mu^2(1-\lambda)^3},
$$

$$\frac{18662400\alpha_v\alpha_y^2\eta^3\gamma_x\gamma_y^2}{\delta^2(1-\lambda)^3} \leq \frac{15840\eta\gamma_x\gamma_y^2}{\alpha_v(1-\lambda)^3} \ . \tag{198}$$

Then, due to $\eta < 1$ and $\alpha_v < 1$, we can obtain

$$\alpha_y \leq \frac{\delta}{44} \ . \tag{199}$$

For Eq. (119), we have

$$
\begin{aligned}
c_{10} &\frac{792\eta^2\gamma_y^2}{\delta(1-\lambda)^2}\frac{30}{\alpha_v(1-\lambda)} \\
&= \frac{261360000\eta^3\gamma_x\gamma_y^2L^4}{\alpha_v\mu^2\delta^2(1-\lambda)^5} + \frac{205286400\alpha_y^2\eta^3\gamma_x\gamma_y^2}{\alpha_v\delta^2(1-\lambda)^5}\left(\frac{L^2}{\rho_xK} + \frac{100L^4}{3\rho_y\mu^2K}\right) \\
&\leq c_6 \\
&= \frac{12360\eta\gamma_x\gamma_y^2L^4}{\alpha_v\alpha_y^2\mu^2(1-\lambda)^3} + \frac{15840\eta\gamma_x\gamma_y^2}{\alpha_v(1-\lambda)^3}\left(\frac{L^2}{\rho_xK} + \frac{100L^4}{3\rho_y\mu^2K}\right) \ .
\end{aligned}
\tag{200}
$$

To solve this inequality, we set

$$
\begin{aligned}
\frac{261360000\eta^3\gamma_x\gamma_y^2L^4}{\alpha_v\mu^2\delta^2(1-\lambda)^5} &\leq \frac{12360\eta\gamma_x\gamma_y^2L^4}{\alpha_v\alpha_y^2\mu^2(1-\lambda)^3} \ , \\
\frac{205286400\alpha_y^2\eta^3\gamma_x\gamma_y^2}{\alpha_v\delta^2(1-\lambda)^5} &\leq \frac{15840\eta\gamma_x\gamma_y^2}{\alpha_v(1-\lambda)^3} \ .
\end{aligned}
\tag{201}
$$

Then, due to $\eta < 1$, we can obtain

$$\alpha_y \leq \frac{\delta(1-\lambda)}{146} \ . \tag{202}$$

For Eq. (120), we have

$$
\begin{aligned}
c_{10} &\frac{62954496\alpha_v^2\eta^4\gamma_y^4L^2}{\delta^3(1-\lambda)^2}\frac{30}{\alpha_v(1-\lambda)} \\
&= \frac{20774983680000\alpha_v\eta^5\gamma_x\gamma_y^4L^6}{\mu^2\delta^4(1-\lambda)^5} + \frac{16317805363200\alpha_v\alpha_y^2\eta^5\gamma_x\gamma_y^4L^2}{\delta^4(1-\lambda)^5}\left(\frac{L^2}{\rho_xK} + \frac{100L^4}{3\rho_y\mu^2K}\right) \\
&\leq c_6 \\
&= \frac{12360\eta\gamma_x\gamma_y^2L^4}{\alpha_v\alpha_y^2\mu^2(1-\lambda)^3} + \frac{15840\eta\gamma_x\gamma_y^2}{\alpha_v(1-\lambda)^3}\left(\frac{L^2}{\rho_xK} + \frac{100L^4}{3\rho_y\mu^2K}\right) \ .
\end{aligned}
\tag{203}
$$

To solve this inequality, we set

$$
\begin{aligned}
\frac{20774983680000\alpha_v\eta^5\gamma_x\gamma_y^4L^6}{\mu^2\delta^4(1-\lambda)^5} &\leq \frac{12360\eta\gamma_x\gamma_y^2L^4}{\alpha_v\alpha_y^2\mu^2(1-\lambda)^3} \ , \\
\frac{16317805363200\alpha_v\alpha_y^2\eta^5\gamma_x\gamma_y^4L^2}{\delta^4(1-\lambda)^5} &\leq \frac{15840\eta\gamma_x\gamma_y^2}{\alpha_v(1-\lambda)^3} \ .
\end{aligned}
\tag{204}
$$

Then, due to $\eta < 1$, $\alpha_v < 1$, and $\alpha_y\eta < 1$, we can obtain

$$\gamma_y \leq \frac{\delta^2(1-\lambda)}{40998L} \ . \tag{205}$$

For Eq. (121), we have

$$
c_{10}\frac{363528\eta^4\gamma_y^4L^2}{\delta(1-\lambda)^2}\frac{30}{\alpha_v(1-\lambda)}
$$

$$= \frac{119964240000\eta^5\gamma_x\gamma_y^4 L^6}{\alpha_v\mu^2\delta^2(1-\lambda)^5} + \frac{94226457600\alpha_y^2\eta^5\gamma_x\gamma_y^4 L^2}{\alpha_v\delta^2(1-\lambda)^5}\left(\frac{L^2}{\rho_x K} + \frac{100L^4}{3\rho_y\mu^2 K}\right)$$

$$\leq c_6$$

$$= \frac{12360\eta\gamma_x\gamma_y^2 L^4}{\alpha_v\alpha_y^2\mu^2(1-\lambda)^3} + \frac{15840\eta\gamma_x\gamma_y^2}{\alpha_v(1-\lambda)^3}\left(\frac{L^2}{\rho_x K} + \frac{100L^4}{3\rho_y\mu^2 K}\right). \tag{206}$$

To solve this inequality, we set

$$\frac{119964240000\eta^5\gamma_x\gamma_y^4 L^6}{\alpha_v\mu^2\delta^2(1-\lambda)^5} \leq \frac{12360\eta\gamma_x\gamma_y^2 L^4}{\alpha_v\alpha_y^2\mu^2(1-\lambda)^3},$$

$$\frac{94226457600\alpha_y^2\eta^5\gamma_x\gamma_y^4 L^2}{\alpha_v\delta^2(1-\lambda)^5} \leq \frac{15840\eta\gamma_x\gamma_y^2}{\alpha_v(1-\lambda)^3}. \tag{207}$$

Then, due to $\eta < 1$ and $\alpha_y\eta < 1$, we can obtain

$$\gamma_y \leq \frac{\delta(1-\lambda)}{3116L}. \tag{208}$$

For Eq. (122), we have

$$c_5\frac{13068\eta^2\gamma_y^2 L^2}{\alpha_u(1-\lambda)^3}\frac{30}{\alpha_v(1-\lambda)}$$

$$= \frac{4845614400\eta^3\gamma_x^3\gamma_y^2 L^6}{\alpha_u^2\alpha_v\alpha_x^2\mu^2(1-\lambda)^7} + \frac{6209913600\eta^3\gamma_x^3\gamma_y^2 L^2}{\alpha_u^2\alpha_v(1-\lambda)^7}\left(\frac{L^2}{\rho_x K} + \frac{100L^4}{3\rho_y\mu^2 K}\right)$$

$$\leq c_6$$

$$= \frac{12360\eta\gamma_x\gamma_y^2 L^4}{\alpha_v\alpha_y^2\mu^2(1-\lambda)^3} + \frac{15840\eta\gamma_x\gamma_y^2}{\alpha_v(1-\lambda)^3}\left(\frac{L^2}{\rho_x K} + \frac{100L^4}{3\rho_y\mu^2 K}\right). \tag{209}$$

To solve this inequality, we set

$$\frac{4845614400\eta^3\gamma_x^3\gamma_y^2 L^6}{\alpha_u^2\alpha_v\alpha_x^2\mu^2(1-\lambda)^7} \leq \frac{12360\eta\gamma_x\gamma_y^2 L^4}{\alpha_v\alpha_y^2\mu^2(1-\lambda)^3},$$

$$\frac{6209913600\eta^3\gamma_x^3\gamma_y^2 L^2}{\alpha_u^2\alpha_v(1-\lambda)^7} \leq \frac{15840\eta\gamma_x\gamma_y^2}{\alpha_v(1-\lambda)^3}. \tag{210}$$

Then, due to $\eta < 1$ and $\alpha_y\eta < 1$, we can obtain

$$\gamma_x \leq \min\left\{\frac{\alpha_u\alpha_x(1-\lambda)^2}{627L}, \frac{\alpha_u(1-\lambda)^2}{627L}\right\}. \tag{211}$$

For Eq. (123), we have

$$c_5\frac{3497472\alpha_u\eta^2\gamma_y^2(1-\delta)L^2}{\delta^2(1-\lambda)^3}\frac{30}{\alpha_v(1-\lambda)}$$

$$= \frac{1296862617600\eta^3\gamma_x^3\gamma_y^2 L^6}{\alpha_v\alpha_x^2\mu^2\delta^2(1-\lambda)^7} + \frac{1661998694400\eta^3\gamma_x^3\gamma_y^2 L^2}{\alpha_v\delta^2(1-\lambda)^7}\left(\frac{L^2}{\rho_x K} + \frac{100L^4}{3\rho_y\mu^2 K}\right)$$

$$\leq c_6$$

$$= \frac{12360\eta\gamma_x\gamma_y^2 L^4}{\alpha_v\alpha_y^2\mu^2(1-\lambda)^3} + \frac{15840\eta\gamma_x\gamma_y^2}{\alpha_v(1-\lambda)^3}\left(\frac{L^2}{\rho_x K} + \frac{100L^4}{3\rho_y\mu^2 K}\right). \tag{212}$$

To solve this inequality, we set

$$\frac{1296862617600\eta^3\gamma_x^3\gamma_y^2 L^6}{\alpha_v\alpha_x^2\mu^2\delta^2(1-\lambda)^7} \leq \frac{12360\eta\gamma_x\gamma_y^2 L^4}{\alpha_v\alpha_y^2\mu^2(1-\lambda)^3},$$

$$\frac{1661998694400\eta^3\gamma_x^3\gamma_y^2L^2}{\alpha_v\delta^2(1-\lambda)^7} \leq \frac{15840\eta\gamma_x\gamma_y^2}{\alpha_v(1-\lambda)^3} \ . \tag{213}$$

Then, due to $\eta < 1$ and $\alpha_y\eta < 1$, we can obtain

$$\gamma_x \leq \min\left\{\frac{\alpha_x\delta(1-\lambda)^2}{10244L}, \frac{\delta(1-\lambda)^2}{10244L}\right\} \ . \tag{214}$$

For Eq. (124), we have

$$c_9\frac{62954496\alpha_u^2\eta^4\gamma_x^2\gamma_y^2L^2}{\delta^3(1-\lambda)^2}\frac{30}{\alpha_v(1-\lambda)}$$
$$= \frac{20774983680000\alpha_u^2\eta^5\gamma_x^3\gamma_y^2L^6}{\alpha_v\mu^2\delta^4(1-\lambda)^5} + \frac{16317805363200\alpha_u^2\alpha_x^2\eta^5\gamma_x^3\gamma_y^2L^2}{\alpha_v\delta^4(1-\lambda)^5}\left(\frac{L^2}{\rho_xK} + \frac{100L^4}{3\rho_y\mu^2K}\right)$$
$$\leq c_6$$
$$= \frac{12360\eta\gamma_x\gamma_y^2L^4}{\alpha_v\alpha_y^2\mu^2(1-\lambda)^3} + \frac{15840\eta\gamma_x\gamma_y^2}{\alpha_v(1-\lambda)^3}\left(\frac{L^2}{\rho_xK} + \frac{100L^4}{3\rho_y\mu^2K}\right) \ . \tag{215}$$

To solve this inequality, we set

$$\frac{20774983680000\alpha_u^2\eta^5\gamma_x^3\gamma_y^2L^6}{\alpha_v\mu^2\delta^4(1-\lambda)^5} \leq \frac{12360\eta\gamma_x\gamma_y^2L^4}{\alpha_v\alpha_y^2\mu^2(1-\lambda)^3} \ ,$$
$$\frac{16317805363200\alpha_u^2\alpha_x^2\eta^5\gamma_x^3\gamma_y^2L^2}{\alpha_v\delta^4(1-\lambda)^5} \leq \frac{15840\eta\gamma_x\gamma_y^2}{\alpha_v(1-\lambda)^3} \ . \tag{216}$$

Then, due to $\eta < 1$, $\alpha_u < 1$, $\alpha_x\eta < 1$, and $\alpha_y\eta < 1$, we can obtain

$$\gamma_x \leq \frac{\delta^2(1-\lambda)}{40998L} \ . \tag{217}$$

For Eq. (125), we have

$$c_9\frac{363528\eta^4\gamma_x^2\gamma_y^2L^2}{\delta(1-\lambda)^2}\frac{30}{\alpha_v(1-\lambda)}$$
$$= \frac{119964240000\eta^5\gamma_x^3\gamma_y^2L^6}{\alpha_v\mu^2\delta^2(1-\lambda)^5} + \frac{94226457600\alpha_x^2\eta^5\gamma_x^3\gamma_y^2L^2}{\alpha_v\delta^2(1-\lambda)^5}\left(\frac{L^2}{\rho_xK} + \frac{100L^4}{3\rho_y\mu^2K}\right)$$
$$\leq c_6$$
$$= \frac{12360\eta\gamma_x\gamma_y^2L^4}{\alpha_v\alpha_y^2\mu^2(1-\lambda)^3} + \frac{15840\eta\gamma_x\gamma_y^2}{\alpha_v(1-\lambda)^3}\left(\frac{L^2}{\rho_xK} + \frac{100L^4}{3\rho_y\mu^2K}\right) \ . \tag{218}$$

To solve this inequality, we set

$$\frac{119964240000\eta^5\gamma_x^3\gamma_y^2L^6}{\alpha_v\mu^2\delta^2(1-\lambda)^5} \leq \frac{12360\eta\gamma_x\gamma_y^2L^4}{\alpha_v\alpha_y^2\mu^2(1-\lambda)^3} \ ,$$
$$\frac{94226457600\alpha_x^2\eta^5\gamma_x^3\gamma_y^2L^2}{\alpha_v\delta^2(1-\lambda)^5} \leq \frac{15840\eta\gamma_x\gamma_y^2}{\alpha_v(1-\lambda)^3} \ . \tag{219}$$

Then, due to $\eta < 1$, $\alpha_x\eta < 1$, and $\alpha_y\eta < 1$, we can obtain

$$\gamma_x \leq \frac{\delta(1-\lambda)}{3116L} \ . \tag{220}$$

By now, all Eqs. (69-74), Eqs. (84-89), Eqs. (99-107), and Eqs. (117-125) have been solved in terms of the values of $\{c_5, c_6, c_9, c_{10}\}$. In what follows, we eliminate $\mathbb{E}\left[\|\bar{g}_t\|^2\right]$ and $\mathbb{E}\left[\|\bar{h}_t\|^2\right]$.

To eliminate $\mathbb{E}\left[\|\bar{g}_t\|^2\right]$, we enforce

$$-\frac{\gamma_x\eta}{4} + c_0\frac{25\eta\gamma_x^2L^2}{6\gamma_y\mu^3} + C_5\frac{9\gamma_x^2}{2\alpha_x^2}$$

$$\leq -\frac{\gamma_x\eta}{4} + \frac{200\eta\gamma_x^3L^4}{6\gamma_y^2\mu^4}$$

$$+ \frac{216\gamma_x^3\eta L^2}{\rho_x K} + \frac{21600\gamma_x^3\eta L^4}{3\rho_y\mu^2 K} + c_9\frac{324\gamma_x^2\eta^2}{\delta} + c_9\frac{25754112\gamma_x^4\alpha_u^2\eta^4L^2}{\delta^3} + c_9\frac{148716\gamma_x^4\eta^4L^2}{\delta}$$

$$+ c_5\frac{5346\gamma_x^2\eta^2L^2}{\alpha_u(1-\lambda)} + c_5\frac{1430784\gamma_x^2\alpha_u\eta^2L^2}{\delta^2(1-\lambda)}$$

$$+ c_6\frac{5346\gamma_x^2\eta^2L^2}{\alpha_v(1-\lambda)} + c_6\frac{1430784\gamma_x^2\alpha_v\eta^2L^2}{\delta^2(1-\lambda)}$$

$$+ c_{10}\frac{25754112\gamma_x^2\gamma_y^2\alpha_v^2\eta^4L^2}{\delta^3} + c_{10}\frac{148716\gamma_x^2\gamma_y^2\eta^4L^2}{\delta}$$

$$\leq 0 . \tag{221}$$

To solve this inequality, we set

$$\frac{200\eta\gamma_x^3L^4}{6\gamma_y^2\mu^4} \leq \frac{\gamma_x\eta}{60} , \tag{222}$$

$$\frac{216\gamma_x^3\eta L^2}{\rho_x K} \leq \frac{\gamma_x\eta}{60} , \tag{223}$$

$$\frac{21600\gamma_x^3\eta L^4}{3\rho_y\mu^2 K} \leq \frac{\gamma_x\eta}{60} , \tag{224}$$

$$c_9\frac{324\gamma_x^2\eta^2}{\delta} \leq \frac{\gamma_x\eta}{60} , \tag{225}$$

$$c_9\frac{25754112\gamma_x^4\alpha_u^2\eta^4L^2}{\delta^3} \leq \frac{\gamma_x\eta}{60} , \tag{226}$$

$$c_9\frac{148716\gamma_x^4\eta^4L^2}{\delta} \leq \frac{\gamma_x\eta}{60} , \tag{227}$$

$$c_5\frac{5346\gamma_x^2\eta^2L^2}{\alpha_u(1-\lambda)} \leq \frac{\gamma_x\eta}{60} , \tag{228}$$

$$c_5\frac{1430784\gamma_x^2\alpha_u\eta^2L^2}{\delta^2(1-\lambda)} \leq \frac{\gamma_x\eta}{60} , \tag{229}$$

$$c_6\frac{5346\gamma_x^2\eta^2L^2}{\alpha_v(1-\lambda)} \leq \frac{\gamma_x\eta}{60} , \tag{230}$$

$$c_6\frac{1430784\gamma_x^2\alpha_v\eta^2L^2}{\delta^2(1-\lambda)} \leq \frac{\gamma_x\eta}{60} , \tag{231}$$

$$c_{10}\frac{25754112\gamma_x^2\gamma_y^2\alpha_v^2\eta^4L^2}{\delta^3} \leq \frac{\gamma_x\eta}{60} , \tag{232}$$

$$c_{10}\frac{148716\gamma_x^2\gamma_y^2\eta^4L^2}{\delta} \leq \frac{\gamma_x\eta}{60} . \tag{233}$$

For Eq. (222), we can obtain

$$\gamma_x \leq \frac{\gamma_y\mu^2}{45L^2} . \tag{234}$$

For Eq. (223), we can obtain

$$\gamma_x \leq \frac{\sqrt{\rho_x K}}{114L} . \tag{235}$$

For Eq. (224), we can obtain

$$\gamma_x \leq \frac{\sqrt{\rho_y K}\mu}{658L^2} \ . \tag{236}$$

For Eq. (225), we have

$$
\begin{aligned}
&c_9 \frac{324\gamma_x^2\eta^2}{\delta} \\
&= \frac{11000\eta\gamma_x L^4}{\mu^2\delta(1-\lambda)^2}\frac{324\gamma_x^2\eta^2}{\delta} + \frac{8640\alpha_x^2\eta\gamma_x}{\delta(1-\lambda)^2}\frac{324\gamma_x^2\eta^2}{\delta}\left(\frac{L^2}{\rho_x K} + \frac{100L^4}{3\rho_y\mu^2 K}\right) \\
&\leq \frac{\gamma_x\eta}{60} \ .
\end{aligned} \tag{237}
$$

To solve this inequality, we set

$$
\begin{aligned}
\frac{11000\eta\gamma_x L^4}{\mu^2\delta(1-\lambda)^2}\frac{324\gamma_x^2\eta^2}{\delta} &\leq \frac{\gamma_x\eta}{180} \ , \\
\frac{8640\alpha_x^2\eta\gamma_x}{\delta(1-\lambda)^2}\frac{324\gamma_x^2\eta^2}{\delta}\frac{L^2}{\rho_x K} &\leq \frac{\gamma_x\eta}{180} \ , \\
\frac{8640\alpha_x^2\eta\gamma_x}{\delta(1-\lambda)^2}\frac{324\gamma_x^2\eta^2}{\delta}\frac{100L^4}{3\rho_y\mu^2 K} &\leq \frac{\gamma_x\eta}{180} \ .
\end{aligned} \tag{238}
$$

Then, due to $\eta < 1$ and $\alpha_x\eta < 1$, we can obtain

$$\gamma_x \leq \left\{ \frac{\mu\delta(1-\lambda)}{25329L^2}, \ \frac{\sqrt{\rho_x K}\delta(1-\lambda)}{22448L}, \ \frac{\sqrt{\rho_y K}\mu\delta(1-\lambda)}{129600L^2} \right\} \ . \tag{239}$$

For Eq. (226), we have

$$
\begin{aligned}
&c_9 \frac{25754112\gamma_x^4\alpha_u^2\eta^4 L^2}{\delta^3} \\
&= \frac{11000 * 25754112\alpha_u^2\eta^5\gamma_x^5 L^6}{\mu^2\delta^4(1-\lambda)^2} + \frac{8640 * 25754112\alpha_u^2\alpha_x^2\eta^5\gamma_x^5 L^4}{\rho_x\delta^4(1-\lambda)^2 K} \\
&\quad + \frac{8640 * 25754112 * 100\alpha_u^2\alpha_x^2\eta^5\gamma_x^5 L^6}{3\rho_y\mu^2\delta^4(1-\lambda)^2 K} \\
&\leq \frac{\gamma_x\eta}{60} \ .
\end{aligned} \tag{240}
$$

To solve this inequality, we set

$$
\begin{aligned}
\frac{11000 * 25754112\alpha_u^2\eta^5\gamma_x^5 L^6}{\mu^2\delta^4(1-\lambda)^2} &\leq \frac{\gamma_x\eta}{180} \ , \\
\frac{8640 * 25754112\alpha_u^2\alpha_x^2\eta^5\gamma_x^5 L^4}{\rho_x\delta^4(1-\lambda)^2 K} &\leq \frac{\gamma_x\eta}{180} \ , \\
\frac{8640 * 25754112 * 100\alpha_u^2\alpha_x^2\eta^5\gamma_x^5 L^6}{3\rho_y\mu^2\delta^4(1-\lambda)^2 K} &\leq \frac{\gamma_x\eta}{180} \ .
\end{aligned} \tag{241}
$$

Then, due to $\eta < 1$, $\alpha_u < 1$, and $\alpha_x\eta < 1$, we can obtain

$$\gamma_x \leq \left\{ \frac{\rho_y^{\frac{1}{4}}\mu^{\frac{1}{2}}\delta(1-\lambda)^{\frac{1}{2}}K^{\frac{1}{4}}}{6045L^{\frac{3}{2}}}, \ \frac{\rho_x^{\frac{1}{4}}\delta(1-\lambda)^{\frac{1}{2}}K^{\frac{1}{4}}}{2516L}, \ \frac{\mu^{\frac{1}{2}}\delta(1-\lambda)^{\frac{1}{2}}}{2673L^{\frac{3}{2}}} \right\} \ . \tag{242}$$

For Eq. (227), we have

$$
c_9 \frac{148716\gamma_x^4\eta^4 L^2}{\delta}
$$
$$
= \frac{11000 * 148716\eta^5\gamma_x^5 L^6}{\mu^2\delta^2(1-\lambda)^2} + \frac{8640 * 148716\alpha_x^2\eta^5\gamma_x^5 L^4}{\rho_x\delta^2(1-\lambda)^2 K}
$$
$$
+ \frac{8640 * 148716 * 100\alpha_x^2\eta^5\gamma_x^5 L^6}{3\rho_y\mu^2\delta^2(1-\lambda)^2 K}
$$
$$
\leq \frac{\gamma_x\eta}{60} . \tag{243}
$$

To solve this inequality, we set

$$
\frac{11000 * 148716\eta^5\gamma_x^5 L^6}{\mu^2\delta^2(1-\lambda)^2} \leq \frac{\gamma_x\eta}{180} ,
$$
$$
\frac{8640 * 148716\alpha_x^2\eta^5\gamma_x^5 L^4}{\rho_x\delta^2(1-\lambda)^2 K} \leq \frac{\gamma_x\eta}{180} ,
$$
$$
\frac{8640 * 148716 * 100\alpha_x^2\eta^5\gamma_x^5 L^6}{3\rho_y\mu^2\delta^2(1-\lambda)^2 K} \leq \frac{\gamma_x\eta}{180} . \tag{244}
$$

Then, due to $\eta < 1$ and $\alpha_x\eta < 1$, we can obtain

$$
\gamma_x \leq \left\{ \frac{\mu^{\frac{1}{2}}\delta^{\frac{1}{2}}(1-\lambda)^{\frac{1}{2}}}{737 L^{\frac{3}{2}}}, \frac{\rho_x^{\frac{1}{4}}\delta^{\frac{1}{2}}(1-\lambda)^{\frac{1}{2}}K^{\frac{1}{4}}}{694 L}, \frac{\rho_y^{\frac{1}{4}}\mu^{\frac{1}{2}}\delta^{\frac{1}{2}}(1-\lambda)^{\frac{1}{2}}K^{\frac{1}{4}}}{1667 L^{\frac{3}{2}}} \right\} . \tag{245}
$$

For Eq. (228), we have

$$
c_5 \frac{5346\gamma_x^2\eta^2 L^2}{\alpha_u(1-\lambda)}
$$
$$
= \frac{12360\eta\gamma_x^3 L^4}{\alpha_u\alpha_x^2\mu^2(1-\lambda)^3} \frac{5346\gamma_x^2\eta^2 L^2}{\alpha_u(1-\lambda)} + \frac{15840\eta\gamma_x^3}{\alpha_u(1-\lambda)^3} \frac{5346\gamma_x^2\eta^2 L^2}{\alpha_u(1-\lambda)} \left( \frac{L^2}{\rho_x K} + \frac{100 L^4}{3\rho_y\mu^2 K} \right)
$$
$$
\leq \frac{\gamma_x\eta}{60} . \tag{246}
$$

To solve this inequality, we set

$$
\frac{12360\eta\gamma_x^3 L^4}{\alpha_u\alpha_x^2\mu^2(1-\lambda)^3} \frac{5346\gamma_x^2\eta^2 L^2}{\alpha_u(1-\lambda)} \leq \frac{\gamma_x\eta}{180} ,
$$
$$
\frac{15840\eta\gamma_x^3}{\alpha_u(1-\lambda)^3} \frac{5346\gamma_x^2\eta^2 L^2}{\alpha_u(1-\lambda)} \frac{L^2}{\rho_x K} \leq \frac{\gamma_x\eta}{180} ,
$$
$$
\frac{15840\eta\gamma_x^3}{\alpha_u(1-\lambda)^3} \frac{5346\gamma_x^2\eta^2 L^2}{\alpha_u(1-\lambda)} \frac{100 L^4}{3\rho_y\mu^2 K} \leq \frac{\gamma_x\eta}{180} . \tag{247}
$$

Then, due to $\eta < 1$, we can obtain

$$
\gamma_x \leq \left\{ \frac{\alpha_u^{\frac{1}{2}}\alpha_x^{\frac{1}{2}}\mu^{\frac{1}{2}}(1-\lambda)}{331 L^{\frac{3}{2}}}, \frac{\alpha_u^{\frac{1}{2}}(1-\lambda)\rho_x^{\frac{1}{4}}K^{\frac{1}{4}}}{352 L}, \frac{\alpha_u^{\frac{1}{2}}\rho_y^{\frac{1}{4}}\mu^{\frac{1}{2}}(1-\lambda)K^{\frac{1}{4}}}{845 L^{\frac{3}{2}}} \right\} . \tag{248}
$$

For Eq. (229), we have

$$
c_5 \frac{5346\gamma_x^2\eta^2 L^2}{\alpha_u(1-\lambda)}
$$
$$
= \frac{12360\eta\gamma_x^3 L^4}{\alpha_u\alpha_x^2\mu^2(1-\lambda)^3} \frac{5346\gamma_x^2\eta^2 L^2}{\alpha_u(1-\lambda)} + \frac{15840\eta\gamma_x^3}{\alpha_u(1-\lambda)^3} \frac{5346\gamma_x^2\eta^2 L^2}{\alpha_u(1-\lambda)} \left( \frac{L^2}{\rho_x K} + \frac{100 L^4}{3\rho_y\mu^2 K} \right)
$$

$$\leq \frac{\gamma_x \eta}{60} \ . \tag{249}$$

To solve this inequality, we set

$$\frac{12360\eta\gamma_x^3 L^4}{\alpha_u \alpha_x^2 \mu^2 (1-\lambda)^3} \frac{5346\gamma_x^2 \eta^2 L^2}{\alpha_u (1-\lambda)} \leq \frac{\gamma_x \eta}{180} \ ,$$

$$\frac{15840\eta\gamma_x^3}{\alpha_u (1-\lambda)^3} \frac{5346\gamma_x^2 \eta^2 L^2}{\alpha_u (1-\lambda)} \frac{L^2}{\rho_x K} \leq \frac{\gamma_x \eta}{180} \ ,$$

$$\frac{15840\eta\gamma_x^3}{\alpha_u (1-\lambda)^3} \frac{5346\gamma_x^2 \eta^2 L^2}{\alpha_u (1-\lambda)} \frac{100L^4}{3\rho_y \mu^2 K} \leq \frac{\gamma_x \eta}{180} \ . \tag{250}$$

Then, due to $\eta < 1$, we can obtain

$$\gamma_x \leq \left\{ \frac{\alpha_u^{\frac{1}{2}} \alpha_x^{\frac{1}{2}} \mu^{\frac{1}{2}} (1-\lambda)}{331 L^{\frac{3}{2}}}, \ \frac{\alpha_u^{\frac{1}{2}} (1-\lambda) \rho_x^{\frac{1}{4}} K^{\frac{1}{4}}}{352 L}, \ \frac{\alpha_u^{\frac{1}{2}} \rho_y^{\frac{1}{4}} \mu^{\frac{1}{2}} (1-\lambda) K^{\frac{1}{4}}}{845 L^{\frac{3}{2}}} \right\} \ . \tag{251}$$

For Eq. (230), we have

$$c_6 \frac{5346\gamma_x^2 \eta^2 L^2}{\alpha_v (1-\lambda)}$$

$$= \frac{12360 * 5346\eta^3 \gamma_x^3 \gamma_y^2 L^6}{\alpha_v^2 \alpha_y^2 \mu^2 (1-\lambda)^4} + \frac{15840 * 5346\eta^3 \gamma_x^3 \gamma_y^2 L^4}{\alpha_v^2 \rho_x (1-\lambda)^4 K} + \frac{15840 * 5346 * 100\eta^3 \gamma_x^3 \gamma_y^2 L^6}{3\alpha_v^2 \rho_y \mu^2 (1-\lambda)^4 K}$$

$$\leq \frac{\gamma_x \eta}{60} \ . \tag{252}$$

To solve this inequality, we set

$$\frac{12360 * 5346\eta^3 \gamma_x^3 \gamma_y^2 L^6}{\alpha_v^2 \alpha_y^2 \mu^2 (1-\lambda)^4} \leq \frac{\gamma_x \eta}{180} \ ,$$

$$\frac{15840 * 5346\eta^3 \gamma_x^3 \gamma_y^2 L^4}{\alpha_v^2 \rho_x (1-\lambda)^4 K} \leq \frac{\gamma_x \eta}{180} \ ,$$

$$\frac{15840 * 5346 * 100\eta^3 \gamma_x^3 \gamma_y^2 L^6}{3\alpha_v^2 \rho_y \mu^2 (1-\lambda)^4 K} \leq \frac{\gamma_x \eta}{180} \ . \tag{253}$$

Then, due to $\gamma_x < \frac{(1-\lambda)}{L}$ and $\eta < 1$, we can obtain

$$\gamma_y \leq \left\{ \frac{\alpha_v \alpha_y \mu (1-\lambda)}{109059 L^2}, \ \frac{\alpha_v \rho_x^{\frac{1}{2}} (1-\lambda) K^{\frac{1}{2}}}{123461 L}, \ \frac{\alpha_v \rho_y^{\frac{1}{2}} \mu (1-\lambda) K^{\frac{1}{2}}}{712800 L^2} \right\} \ . \tag{254}$$

For Eq. (231), we have

$$c_6 \frac{1430784\gamma_x^2 \alpha_v \eta^2 L^2}{\delta^2 (1-\lambda)}$$

$$= \frac{12360 * 1430784\eta^3 \gamma_x^3 \gamma_y^2 L^6}{\alpha_y^2 \mu^2 \delta^2 (1-\lambda)^4} + \frac{15840 * 1430784\eta^3 \gamma_x^3 \gamma_y^2 L^4}{\rho_x \delta^2 (1-\lambda)^4 K}$$

$$+ \frac{15840 * 1430784 * 100\eta^3 \gamma_x^3 \gamma_y^2 L^6}{3\rho_y \mu^2 \delta^2 (1-\lambda)^4 K}$$

$$\leq \frac{\gamma_x \eta}{60} \ . \tag{255}$$

To solve this inequality, we set

$$\frac{12360 * 1430784\eta^3 \gamma_x^3 \gamma_y^2 L^6}{\alpha_y^2 \mu^2 \delta^2 (1-\lambda)^4} \leq \frac{\gamma_x \eta}{180} \ ,$$

$$\frac{15840 * 1430784\eta^3\gamma_x^3\gamma_y^2 L^4}{\rho_x \delta^2 (1-\lambda)^4 K} \leq \frac{\gamma_x \eta}{180},$$

$$\frac{15840 * 1430784 * 100\eta^3\gamma_x^3\gamma_y^2 L^6}{3\rho_y \mu^2 \delta^2 (1-\lambda)^4 K} \leq \frac{\gamma_x \eta}{180}. \tag{256}$$

Then, due to $\eta < 1$ and $\gamma_x < \frac{(1-\lambda)}{L}$, we can obtain

$$\gamma_y \leq \left\{ \frac{\alpha_y \mu \delta (1-\lambda)}{1784155 L^2}, \frac{\rho_x^{\frac{1}{2}} \delta (1-\lambda) K^{\frac{1}{2}}}{2019766 L}, \frac{\rho_y^{\frac{1}{2}} \mu \delta (1-\lambda) K^{\frac{1}{2}}}{11661120 L^2} \right\}. \tag{257}$$

For Eq. (232), we have

$$c_{10} \frac{25754112\gamma_x^2\gamma_y^2\alpha_v^2\eta^4 L^2}{\delta^3}$$

$$= \frac{11000 * 25754112\alpha_v^2\eta^5\gamma_x^3\gamma_y^2 L^6}{\mu^2 \delta^4 (1-\lambda)^2} + \frac{8640 * 25754112\alpha_v^2\alpha_y^2\eta^5\gamma_x^3\gamma_y^2 L^4}{\rho_x \delta^4 (1-\lambda)^2 K}$$

$$+ \frac{8640 * 25754112 * 100\alpha_v^2\alpha_y^2\eta^5\gamma_x^3\gamma_y^2 L^6}{3\rho_y \mu^2 \delta^4 (1-\lambda)^2 K}$$

$$\leq \frac{\gamma_x \eta}{60}. \tag{258}$$

To solve this inequality, we set

$$\frac{11000 * 25754112\alpha_v^2\eta^5\gamma_x^3\gamma_y^2 L^6}{\mu^2 \delta^4 (1-\lambda)^2} \leq \frac{\gamma_x \eta}{180},$$

$$\frac{8640 * 25754112\alpha_v^2\alpha_y^2\eta^5\gamma_x^3\gamma_y^2 L^4}{\rho_x \delta^4 (1-\lambda)^2 K} \leq \frac{\gamma_x \eta}{180},$$

$$\frac{8640 * 25754112 * 100\alpha_v^2\alpha_y^2\eta^5\gamma_x^3\gamma_y^2 L^6}{3\rho_y \mu^2 \delta^4 (1-\lambda)^2 K} \leq \frac{\gamma_x \eta}{180}. \tag{259}$$

Then, due to $\eta < 1$, $\alpha_v < 1$, $\gamma_x < \frac{(1-\lambda)}{L}$ and $\alpha_y \eta < 1$, we can obtain

$$\gamma_y \leq \left\{ \frac{\mu \delta^2}{7140949 L^2}, \frac{\rho_x^{\frac{1}{2}} \delta^2 K^{\frac{1}{2}}}{6328728 L}, \frac{\rho_y^{\frac{1}{2}} \mu \delta^2 K^{\frac{1}{2}}}{36538927 L^2} \right\}. \tag{260}$$

For Eq. (233), we have

$$c_{10} \frac{148716\gamma_x^2\gamma_y^2\eta^4 L^2}{\delta}$$

$$= \frac{11000 * 148716\eta^5\gamma_x^3\gamma_y^2 L^6}{\mu^2 \delta^2 (1-\lambda)^2} + \frac{8640 * 148716\alpha_y^2\eta^5\gamma_x^3\gamma_y^2 L^4}{\rho_x \delta^2 (1-\lambda)^2 K}$$

$$+ \frac{8640 * 148716 * 100\alpha_y^2\eta^5\gamma_x^3\gamma_y^2 L^6}{3\rho_y \mu^2 \delta^2 (1-\lambda)^2 K}$$

$$\leq \frac{\gamma_x \eta}{60}. \tag{261}$$

To solve this inequality, we set

$$\frac{11000 * 148716\eta^5\gamma_x^3\gamma_y^2 L^6}{\mu^2 \delta^2 (1-\lambda)^2} \leq \frac{\gamma_x \eta}{180},$$

$$\frac{8640 * 148716\alpha_y^2\eta^5\gamma_x^3\gamma_y^2 L^4}{\rho_x \delta^2 (1-\lambda)^2 K} \leq \frac{\gamma_x \eta}{180},$$

$$\frac{8640 * 148716 * 100\alpha_y^2\eta^5\gamma_x^3\gamma_y^2 L^6}{3\rho_y\mu^2\delta^2(1-\lambda)^2 K} \leq \frac{\gamma_x\eta}{180} . \tag{262}$$

Then, due to $\eta < 1$, $\gamma_x < \frac{(1-\lambda)}{L}$ and $\alpha_y\eta < 1$, we can obtain

$$\gamma_y \leq \left\{ \frac{\mu\delta}{542640L^2}, \frac{\rho_x^{\frac{1}{2}}\delta K^{\frac{1}{2}}}{480920L}, \frac{\rho_y^{\frac{1}{2}}\mu\delta K^{\frac{1}{2}}}{2776588L^2} \right\} . \tag{263}$$

To eliminate $\mathbb{E}\left[\|\bar{h}_t\|^2\right]$, we enforce

$$
\begin{aligned}
&- c_0\frac{3\eta\gamma_y^2}{4} + C_6\frac{9\gamma_y^2}{2\alpha_y^2} \\
\leq\ & -\frac{6\eta\gamma_x\gamma_y L^2}{\mu} + \frac{216\gamma_x\gamma_y^2\eta L^2}{\rho_x K} + \frac{21600\gamma_x\gamma_y^2\eta L^4}{3\rho_y\mu^2 K} \\
& + c_{10}\frac{324\gamma_y^2\eta^2}{\delta} + c_{10}\frac{25754112\gamma_y^4\alpha_v^2\eta^4 L^2}{\delta^3} + c_{10}\frac{148716\gamma_y^4\eta^4 L^2}{\delta} \\
& + c_5\frac{5346\gamma_y^2\eta^2 L^2}{\alpha_u(1-\lambda)} + c_5\frac{1430784\gamma_y^2\alpha_u\eta^2 L^2}{\delta^2(1-\lambda)} \\
& + c_9\frac{25754112\gamma_y^2\gamma_x^2\alpha_u^2\eta^4 L^2}{\delta^3} + c_9\frac{148716\gamma_y^2\gamma_x^2\eta^4 L^2}{\delta} \\
& + c_6\frac{5346\gamma_y^2\eta^2 L^2}{\alpha_v(1-\lambda)} + c_6\frac{1430784\gamma_y^2\alpha_v\eta^2 L^2}{\delta^2(1-\lambda)} \\
\leq\ & 0 .
\end{aligned}
\tag{264}
$$

To solve this inequality, we set

$$\frac{216\gamma_x\gamma_y^2\eta L^2}{\rho_x K} \leq \frac{\eta\gamma_x\gamma_y L^2}{2\mu} , \tag{265}$$

$$\frac{21600\gamma_x\gamma_y^2\eta L^4}{3\rho_y\mu^2 K} \leq \frac{\eta\gamma_x\gamma_y L^2}{2\mu} , \tag{266}$$

$$c_{10}\frac{324\gamma_y^2\eta^2}{\delta} \leq \frac{\eta\gamma_x\gamma_y L^2}{2\mu} , \tag{267}$$

$$c_{10}\frac{25754112\gamma_y^4\alpha_v^2\eta^4 L^2}{\delta^3} \leq \frac{\eta\gamma_x\gamma_y L^2}{2\mu} , \tag{268}$$

$$c_{10}\frac{148716\gamma_y^4\eta^4 L^2}{\delta} \leq \frac{\eta\gamma_x\gamma_y L^2}{2\mu} , \tag{269}$$

$$c_5\frac{5346\gamma_y^2\eta^2 L^2}{\alpha_u(1-\lambda)} \leq \frac{\eta\gamma_x\gamma_y L^2}{2\mu} , \tag{270}$$

$$c_5\frac{1430784\gamma_y^2\alpha_u\eta^2 L^2}{\delta^2(1-\lambda)} \leq \frac{\eta\gamma_x\gamma_y L^2}{2\mu} , \tag{271}$$

$$c_9\frac{25754112\gamma_y^2\gamma_x^2\alpha_u^2\eta^4 L^2}{\delta^3} \leq \frac{\eta\gamma_x\gamma_y L^2}{2\mu} , \tag{272}$$

$$c_9\frac{148716\gamma_y^2\gamma_x^2\eta^4 L^2}{\delta} \leq \frac{\eta\gamma_x\gamma_y L^2}{2\mu} , \tag{273}$$

$$c_6\frac{5346\gamma_y^2\eta^2 L^2}{\alpha_v(1-\lambda)} \leq \frac{\eta\gamma_x\gamma_y L^2}{2\mu} , \tag{274}$$

$$c_6\frac{1430784\gamma_y^2\alpha_v\eta^2 L^2}{\delta^2(1-\lambda)} \leq \frac{\eta\gamma_x\gamma_y L^2}{2\mu} . \tag{275}$$

For Eq. (265), we can obtain

$$\gamma_y \leq \frac{\rho_x K}{432\mu} . \tag{276}$$

For Eq. (266), we can obtain

$$\gamma_y \leq \frac{3\rho_y \mu K}{43200 L^2} . \tag{277}$$

For Eq. (267), we have

$$
\begin{aligned}
&c_{10} \frac{324\gamma_y^2 \eta^2}{\delta} \\
&= \frac{11000 * 324\eta^3 \gamma_x \gamma_y^2 L^4}{\mu^2 \delta^2 (1-\lambda)^2} + \frac{8640 * 324\alpha_y^2 \eta^3 \gamma_x \gamma_y^2 L^2}{\rho_x \delta^2 (1-\lambda)^2 K} + \frac{8640 * 324 * 100\alpha_y^2 \eta^3 \gamma_x \gamma_y^2 L^4}{3\rho_y \mu^2 \delta^2 (1-\lambda)^2 K} \\
&\leq \frac{\eta \gamma_x \gamma_y L^2}{2\mu} .
\end{aligned}
\tag{278}
$$

To solve this inequality, we set

$$
\begin{aligned}
\frac{11000 * 324\eta^3 \gamma_x \gamma_y^2 L^4}{\mu^2 \delta^2 (1-\lambda)^2} &\leq \frac{\eta \gamma_x \gamma_y L^2}{6\mu} , \\
\frac{8640 * 324\alpha_y^2 \eta^3 \gamma_x \gamma_y^2 L^2}{\rho_x \delta^2 (1-\lambda)^2 K} &\leq \frac{\eta \gamma_x \gamma_y L^2}{6\mu} , \\
\frac{8640 * 324 * 100\alpha_y^2 \eta^3 \gamma_x \gamma_y^2 L^4}{3\rho_y \mu^2 \delta^2 (1-\lambda)^2 K} &\leq \frac{\eta \gamma_x \gamma_y L^2}{6\mu} .
\end{aligned}
\tag{279}
$$

Then, due to $\eta < 1$ and $\alpha_y \eta < 1$, we can obtain

$$\gamma_y \leq \min\left\{ \frac{\mu\delta^2(1-\lambda)^2}{21384000 L^2}, \frac{\rho_x \delta^2 (1-\lambda)^2 K}{16796160\mu}, \frac{\rho_y \mu \delta^2 (1-\lambda)^2 K}{559872000 L^2} \right\} . \tag{280}$$

For Eq. (268), we have

$$
\begin{aligned}
&c_{10} \frac{25754112\gamma_y^4 \alpha_v^2 \eta^4 L^2}{\delta^3} \\
&= \frac{11000 * 25754112\alpha_v^2 \eta^5 \gamma_x \gamma_y^4 L^6}{\mu^2 \delta^4 (1-\lambda)^2} + \frac{8640 * 25754112\alpha_v^2 \alpha_y^2 \eta^5 \gamma_x \gamma_y^4 L^4}{\rho_x \delta^4 (1-\lambda)^2 K} \\
&\quad + \frac{8640 * 25754112 * 100\alpha_v^2 \alpha_y^2 \eta^5 \gamma_x \gamma_y^4 L^6}{3\rho_y \mu^2 \delta^4 (1-\lambda)^2 K} \\
&\leq \frac{\eta \gamma_x \gamma_y L^2}{2\mu} .
\end{aligned}
\tag{281}
$$

To solve this inequality, we set

$$
\begin{aligned}
\frac{11000 * 25754112\alpha_v^2 \eta^5 \gamma_x \gamma_y^4 L^6}{\mu^2 \delta^4 (1-\lambda)^2} &\leq \frac{\eta \gamma_x \gamma_y L^2}{6\mu} , \\
\frac{8640 * 25754112\alpha_v^2 \alpha_y^2 \eta^5 \gamma_x \gamma_y^4 L^4}{\rho_x \delta^4 (1-\lambda)^2 K} &\leq \frac{\eta \gamma_x \gamma_y L^2}{6\mu} , \\
\frac{8640 * 25754112 * 100\alpha_v^2 \alpha_y^2 \eta^5 \gamma_x \gamma_y^4 L^6}{3\rho_y \mu^2 \delta^4 (1-\lambda)^2 K} &\leq \frac{\eta \gamma_x \gamma_y L^2}{6\mu} .
\end{aligned}
\tag{282}
$$

Then, due to $\eta < 1$, $\alpha_v < 1$, and $\alpha_y \eta < 1$, we can obtain

$$\gamma_y \leq \min \left\{ \frac{\mu^{\frac{1}{3}} \delta^{\frac{4}{3}} (1-\lambda)^{\frac{2}{3}}}{11935 L^{\frac{4}{3}}}, \frac{\rho_x^{\frac{1}{3}} \delta^{\frac{4}{3}} (1-\lambda)^{\frac{2}{3}} K^{\frac{1}{3}}}{11012 \mu^{\frac{1}{3}} L^{\frac{2}{3}}}, \frac{\rho_y^{\frac{1}{3}} \mu^{\frac{1}{3}} \delta^{\frac{4}{3}} (1-\lambda)^{\frac{2}{3}} K^{\frac{1}{3}}}{35438 L^{\frac{4}{3}}} \right\} . \tag{283}$$

For Eq. (269), we have

$$\begin{aligned}
& c_{10} \frac{148716 \gamma_y^4 \eta^4 L^2}{\delta} \\
&= \frac{11000 * 148716 \eta^5 \gamma_x \gamma_y^4 L^6}{\mu^2 \delta^2 (1-\lambda)^2} + \frac{8640 * 148716 \alpha_y^2 \eta^5 \gamma_x \gamma_y^4 L^4}{\rho_x \delta^2 (1-\lambda)^2 K} \\
&\quad + \frac{8640 * 148716 * 100 \alpha_y^2 \eta^5 \gamma_x \gamma_y^4 L^6}{3 \rho_y \mu^2 \delta^2 (1-\lambda)^2 K} \\
&\leq \frac{\eta \gamma_x \gamma_y L^2}{2\mu} .
\end{aligned} \tag{284}$$

To solve this inequality, we set

$$\begin{aligned}
\frac{11000 * 148716 \eta^5 \gamma_x \gamma_y^4 L^6}{\mu^2 \delta^2 (1-\lambda)^2} &\leq \frac{\eta \gamma_x \gamma_y L^2}{6\mu} , \\
\frac{8640 * 148716 \alpha_y^2 \eta^5 \gamma_x \gamma_y^4 L^4}{\rho_x \delta^2 (1-\lambda)^2 K} &\leq \frac{\eta \gamma_x \gamma_y L^2}{6\mu} , \\
\frac{8640 * 148716 * 100 \alpha_y^2 \eta^5 \gamma_x \gamma_y^4 L^6}{3 \rho_y \mu^2 \delta^2 (1-\lambda)^2 K} &\leq \frac{\eta \gamma_x \gamma_y L^2}{6\mu} .
\end{aligned} \tag{285}$$

Then, due to $\eta < 1$, and $\alpha_y \eta < 1$, we can obtain

$$\gamma_y \leq \min \left\{ \frac{\mu^{\frac{1}{3}} \delta^{\frac{2}{3}} (1-\lambda)^{\frac{2}{3}}}{2142 L^{\frac{4}{3}}}, \frac{\rho_x^{\frac{1}{3}} \delta^{\frac{2}{3}} (1-\lambda)^{\frac{2}{3}} K^{\frac{1}{3}}}{1976 \mu^{\frac{1}{3}} L^{\frac{2}{3}}}, \frac{\rho_y^{\frac{1}{3}} \mu^{\frac{1}{3}} \delta^{\frac{2}{3}} (1-\lambda)^{\frac{2}{3}} K^{\frac{1}{3}}}{6358 L^{\frac{4}{3}}} \right\} . \tag{286}$$

For Eq. (270), we have

$$\begin{aligned}
& c_5 \frac{5346 \gamma_y^2 \eta^2 L^2}{\alpha_u (1-\lambda)} \\
&= \frac{12360 * 5346 \eta^3 \gamma_x^3 \gamma_y^2 L^6}{\alpha_u^2 \alpha_x^2 \mu^2 (1-\lambda)^4} + \frac{15840 * 5346 \eta^3 \gamma_x^3 \gamma_y^2 L^4}{\alpha_u^2 \rho_x (1-\lambda)^4 K} + \frac{15840 * 5346 * 100 \eta^3 \gamma_x^3 \gamma_y^2 L^6}{3 \alpha_u^2 \rho_y \mu^2 (1-\lambda)^4 K} \\
&\leq \frac{\eta \gamma_x \gamma_y L^2}{2\mu} .
\end{aligned} \tag{287}$$

To solve this inequality, we set

$$\begin{aligned}
\frac{12360 * 5346 \eta^3 \gamma_x^3 \gamma_y^2 L^6}{\alpha_u^2 \alpha_x^2 \mu^2 (1-\lambda)^4} &\leq \frac{\eta \gamma_x \gamma_y L^2}{6\mu} , \\
\frac{15840 * 5346 \eta^3 \gamma_x^3 \gamma_y^2 L^4}{\alpha_u^2 \rho_x (1-\lambda)^4 K} &\leq \frac{\eta \gamma_x \gamma_y L^2}{6\mu} , \\
\frac{15840 * 5346 * 100 \eta^3 \gamma_x^3 \gamma_y^2 L^6}{3 \alpha_u^2 \rho_y \mu^2 (1-\lambda)^4 K} &\leq \frac{\eta \gamma_x \gamma_y L^2}{6\mu} .
\end{aligned} \tag{288}$$

Then, due to $\gamma_x < \frac{\alpha_x (1-\lambda)^2}{L}$ and $\eta < 1$, we can obtain

$$\gamma_y \leq \min \left\{ \frac{\alpha_u^2 \mu}{396459360 L^2}, \frac{\alpha_u^2 \rho_x K}{508083840 \mu \alpha_x^2}, \frac{\alpha_u^2 \rho_y \mu K}{16936128000 \alpha_x^2 L^2} \right\} . \tag{289}$$

For Eq. (271), we have

$$
c_5 \frac{1430784\gamma_y^2\alpha_u\eta^2L^2}{\delta^2(1-\lambda)}
$$
$$
= \frac{12360*1430784\eta^3\gamma_x^3\gamma_y^2L^6}{\alpha_x^2\mu^2\delta^2(1-\lambda)^4} + \frac{15840*1430784\eta^3\gamma_x^3\gamma_y^2L^4}{\rho_x\delta^2(1-\lambda)^4K}
$$
$$
+ \frac{15840*1430784*100\eta^3\gamma_x^3\gamma_y^2L^6}{3\rho_y\mu^2\delta^2(1-\lambda)^4K}
$$
$$
\leq \frac{\eta\gamma_x\gamma_yL^2}{2\mu} . \tag{290}
$$

To solve this inequality, we set

$$
\frac{12360*1430784\eta^3\gamma_x^3\gamma_y^2L^6}{\alpha_x^2\mu^2\delta^2(1-\lambda)^4} \leq \frac{\eta\gamma_x\gamma_yL^2}{6\mu} ,
$$
$$
\frac{15840*1430784\eta^3\gamma_x^3\gamma_y^2L^4}{\rho_x\delta^2(1-\lambda)^4K} \leq \frac{\eta\gamma_x\gamma_yL^2}{6\mu} ,
$$
$$
\frac{15840*1430784*100\eta^3\gamma_x^3\gamma_y^2L^6}{3\rho_y\mu^2\delta^2(1-\lambda)^4K} \leq \frac{\eta\gamma_x\gamma_yL^2}{6\mu} . \tag{291}
$$

Then, due to $\gamma_x < \frac{\alpha_x\delta(1-\lambda)^2}{L}$, and $\eta < 1$, we can obtain

$$
\gamma_y \leq \min\left\{ \frac{\mu}{106106941440L^2}, \frac{\rho_xK}{135981711360\mu\alpha_x^2}, \frac{\rho_y\mu K}{4532723712000\alpha_x^2L^2} \right\} . \tag{292}
$$

For Eq. (272), we have

$$
c_9 \frac{25754112\gamma_y^2\gamma_x^2\alpha_u^2\eta^4L^2}{\delta^3}
$$
$$
= \frac{11000*25754112\alpha_u^2\eta^5\gamma_x^3\gamma_y^2L^6}{\mu^2\delta^4(1-\lambda)^2} + \frac{8640*25754112\alpha_u^2\alpha_x^2\eta^5\gamma_x^3\gamma_y^2L^4}{\rho_x\delta^4(1-\lambda)^2K}
$$
$$
+ \frac{8640*25754112*100\alpha_u^2\alpha_x^2\eta^5\gamma_x^3\gamma_y^2L^6}{3\rho_y\mu^2\delta^4(1-\lambda)^2K}
$$
$$
\leq \frac{\eta\gamma_x\gamma_yL^2}{2\mu} . \tag{293}
$$

To solve this inequality, we set

$$
\frac{11000*25754112\alpha_u^2\eta^5\gamma_x^3\gamma_y^2L^6}{\mu^2\delta^4(1-\lambda)^2} \leq \frac{\eta\gamma_x\gamma_yL^2}{6\mu} ,
$$
$$
\frac{8640*25754112\alpha_u^2\alpha_x^2\eta^5\gamma_x^3\gamma_y^2L^4}{\rho_x\delta^4(1-\lambda)^2K} \leq \frac{\eta\gamma_x\gamma_yL^2}{6\mu} ,
$$
$$
\frac{8640*25754112*100\alpha_u^2\alpha_x^2\eta^5\gamma_x^3\gamma_y^2L^6}{3\rho_y\mu^2\delta^4(1-\lambda)^2K} \leq \frac{\eta\gamma_x\gamma_yL^2}{6\mu} . \tag{294}
$$

Then, due to $\gamma_x < \frac{\alpha_x\delta(1-\lambda)}{L}$, $\alpha_u < 1$, $\eta < 1$, and $\alpha_x\eta < 1$, we can obtain

$$
\gamma_y \leq \min\left\{ \frac{\mu\delta^2}{1699771392000L^2}, \frac{\rho_x\delta^2K}{1335093166080\mu}, \frac{\rho_y\mu\delta^2K}{44503105536000L^2} \right\} . \tag{295}
$$

For Eq. (273), we have

$$
c_9 \frac{148716\gamma_y^2\gamma_x^2\eta^4L^2}{\delta}
$$

$$
\begin{aligned}
&= \frac{11000*148716\eta^5\gamma_x^3\gamma_y^2 L^6}{\mu^2\delta^2(1-\lambda)^2} + \frac{8640*148716\alpha_x^2\eta^5\gamma_x^3\gamma_y^2 L^4}{\rho_x\delta^2(1-\lambda)^2 K} \\
&\quad + \frac{8640*148716*100\alpha_x^2\eta^5\gamma_x^3\gamma_y^2 L^6}{3\rho_y\mu^2\delta^2(1-\lambda)^2 K} \\
&\leq \frac{\eta\gamma_x\gamma_y L^2}{2\mu} \ .
\end{aligned} \tag{296}
$$

To solve this inequality, we set

$$
\begin{aligned}
\frac{11000*148716\eta^5\gamma_x^3\gamma_y^2 L^6}{\mu^2\delta^2(1-\lambda)^2} &\leq \frac{\eta\gamma_x\gamma_y L^2}{6\mu} \ , \\
\frac{8640*148716\alpha_x^2\eta^5\gamma_x^3\gamma_y^2 L^4}{\rho_x\delta^2(1-\lambda)^2 K} &\leq \frac{\eta\gamma_x\gamma_y L^2}{6\mu} \ , \\
\frac{8640*148716*100\alpha_x^2\eta^5\gamma_x^3\gamma_y^2 L^6}{3\rho_y\mu^2\delta^2(1-\lambda)^2 K} &\leq \frac{\eta\gamma_x\gamma_y L^2}{6\mu} \ .
\end{aligned} \tag{297}
$$

Then, due to $\gamma_x < \frac{(1-\lambda)}{L}$, $\eta < 1$, and $\alpha_x\eta < 1$, we can obtain

$$
\gamma_y \leq \min\left\{ \frac{\mu\delta^2}{9815256000L^2}, \frac{\rho_x\delta^2 K}{7709437440\mu}, \frac{\rho_y\mu\delta^2 K}{256981248000L^2} \right\} \ . \tag{298}
$$

For Eq. (274), we have

$$
\begin{aligned}
&c_6\frac{5346\gamma_y^2\eta^2 L^2}{\alpha_v(1-\lambda)} \\
&= \frac{12360*5346\eta^3\gamma_x\gamma_y^4 L^6}{\alpha_v^2\alpha_y^2\mu^2(1-\lambda)^4} + \frac{15840*5346\eta^3\gamma_x\gamma_y^4 L^4}{\alpha_v^2\rho_x(1-\lambda)^4 K} + \frac{15840*5346*100\eta^3\gamma_x\gamma_y^4 L^6}{3\alpha_v^2\rho_y\mu^2(1-\lambda)^4 K} \\
&\leq \frac{\eta\gamma_x\gamma_y L^2}{2\mu} \ .
\end{aligned} \tag{299}
$$

To solve this inequality, we set

$$
\begin{aligned}
\frac{12360*5346\eta^3\gamma_x\gamma_y^4 L^6}{\alpha_v^2\alpha_y^2\mu^2(1-\lambda)^4} &\leq \frac{\eta\gamma_x\gamma_y L^2}{6\mu} \ , \\
\frac{15840*5346\eta^3\gamma_x\gamma_y^4 L^4}{\alpha_v^2\rho_x(1-\lambda)^4 K} &\leq \frac{\eta\gamma_x\gamma_y L^2}{6\mu} \ , \\
\frac{15840*5346*100\eta^3\gamma_x\gamma_y^4 L^6}{3\alpha_v^2\rho_y\mu^2(1-\lambda)^4 K} &\leq \frac{\eta\gamma_x\gamma_y L^2}{6\mu} \ .
\end{aligned} \tag{300}
$$

Then, due to $\eta < 1$, we can obtain

$$
\gamma_y \leq \min\left\{ \frac{\alpha_v^{\frac{2}{3}}\alpha_y^{\frac{2}{3}}\mu^{\frac{1}{3}}(1-\lambda)^{\frac{4}{3}}}{735L^{\frac{4}{3}}}, \frac{\alpha_v^{\frac{2}{3}}\rho_x^{\frac{1}{3}}(1-\lambda)^{\frac{4}{3}}K^{\frac{1}{3}}}{798\mu^{\frac{1}{3}}L^{\frac{2}{3}}}, \frac{\alpha_v^{\frac{2}{3}}\rho_y^{\frac{1}{3}}\mu^{\frac{1}{3}}(1-\lambda)^{\frac{4}{3}}K^{\frac{1}{3}}}{2569L^{\frac{4}{3}}} \right\} \ . \tag{301}
$$

For Eq. (275), we have

$$
\begin{aligned}
&c_6\frac{1430784\gamma_y^2\alpha_v\eta^2 L^2}{\delta^2(1-\lambda)} \\
&= \frac{12360*1430784\eta^3\gamma_x\gamma_y^4 L^6}{\alpha_y^2\mu^2\delta^2(1-\lambda)^4} + \frac{15840*1430784\eta^3\gamma_x\gamma_y^4 L^4}{\rho_x\delta^2(1-\lambda)^4 K}
\end{aligned}
$$

$$+ \frac{15840 * 1430784 * 100\eta^3\gamma_x\gamma_y^4 L^6}{3\rho_y\mu^2\delta^2(1-\lambda)^4 K}$$

$$\leq \frac{\eta\gamma_x\gamma_y L^2}{2\mu} . \tag{302}$$

To solve this inequality, we set

$$\frac{12360 * 1430784\eta^3\gamma_x\gamma_y^4 L^6}{\alpha_y^2\mu^2\delta^2(1-\lambda)^4} \leq \frac{\eta\gamma_x\gamma_y L^2}{6\mu} ,$$

$$\frac{15840 * 1430784\eta^3\gamma_x\gamma_y^4 L^4}{\rho_x\delta^2(1-\lambda)^4 K} \leq \frac{\eta\gamma_x\gamma_y L^2}{6\mu} ,$$

$$\frac{15840 * 1430784 * 100\eta^3\gamma_x\gamma_y^4 L^6}{3\rho_y\mu^2\delta^2(1-\lambda)^4 K} \leq \frac{\eta\gamma_x\gamma_y L^2}{6\mu} . \tag{303}$$

Then, due to $\eta < 1$, we can obtain

$$\gamma_y \leq \min\left\{ \frac{\alpha_y^{\frac{2}{3}}\mu^{\frac{1}{3}}\delta^{\frac{2}{3}}(1-\lambda)^{\frac{4}{3}}}{4735 L^{\frac{4}{3}}}, \frac{\rho_x^{\frac{1}{3}}\delta^{\frac{2}{3}}(1-\lambda)^{\frac{4}{3}}K^{\frac{1}{3}}}{5143\mu^{\frac{1}{3}}L^{\frac{2}{3}}}, \frac{\rho_y^{\frac{1}{3}}\mu^{\frac{1}{3}}\delta^{\frac{2}{3}}(1-\lambda)^{\frac{4}{3}}K^{\frac{1}{3}}}{16550 L^{\frac{4}{3}}} \right\} . \tag{304}$$

In summary, by setting

$$c_0 = \frac{8\gamma_x L^2}{\gamma_y\mu} , \quad c_1 = \frac{2\gamma_x}{\rho_x\eta} , \quad c_2 = \frac{200\gamma_x L^2}{3\rho_y\eta\mu^2} ,$$

$$c_3 = \frac{4\gamma_x L^2}{\alpha_x(1-\lambda)} + \frac{400\gamma_x L^4}{3\alpha_x\mu^2(1-\lambda)} + C_5\frac{2}{\alpha_x\eta(1-\lambda)} ,$$

$$c_4 = \frac{4\gamma_x L^2}{\alpha_y(1-\lambda)} + \frac{400\gamma_x L^4}{3\alpha_y\mu^2(1-\lambda)} + C_6\frac{2}{\alpha_y\eta(1-\lambda)} ,$$

$$c_7 = c_5\frac{64\alpha_u}{\delta(1-\lambda)} + c_9\frac{1152\gamma_x^2\alpha_u^2\eta^2}{\delta^2} ,$$

$$c_8 = c_6\frac{64\alpha_v}{\delta(1-\lambda)} + c_{10}\frac{1152\gamma_y^2\alpha_v^2\eta^2}{\delta^2} ,$$

$$c_{11} = c_5\frac{33\eta^2\rho_x}{\alpha_u(1-\lambda)} + c_5\frac{8832\alpha_u\eta^2\rho_x}{\delta^2(1-\lambda)} + c_9\frac{158976\gamma_x^2\alpha_u^2\eta^4\rho_x}{\delta^3} + c_9\frac{918\gamma_x^2\eta^4\rho_x}{\delta} ,$$

$$c_{12} = c_6\frac{33\eta^2\rho_y}{\alpha_v(1-\lambda)} + c_6\frac{8832\alpha_v\eta^2\rho_y}{\delta^2(1-\lambda)} + c_{10}\frac{158976\gamma_y^2\alpha_v^2\eta^4\rho_y}{\delta^3} + c_{10}\frac{918\gamma_y^2\eta^4\rho_y}{\delta} ,$$

$$c_5 = \frac{12360\eta\gamma_x^3 L^4}{\alpha_u\alpha_x^2\mu^2(1-\lambda)^3} + \frac{15840\eta\gamma_x^3}{\alpha_u(1-\lambda)^3}\left(\frac{L^2}{\rho_x K} + \frac{100L^4}{3\rho_y\mu^2 K}\right) ,$$

$$c_6 = \frac{12360\eta\gamma_x\gamma_y^2 L^4}{\alpha_v\alpha_y^2\mu^2(1-\lambda)^3} + \frac{15840\eta\gamma_x\gamma_y^2}{\alpha_v(1-\lambda)^3}\left(\frac{L^2}{\rho_x K} + \frac{100L^4}{3\rho_y\mu^2 K}\right) ,$$

$$c_9 = \frac{11000\eta\gamma_x L^4}{\mu^2\delta(1-\lambda)^2} + \frac{8640\alpha_x^2\eta\gamma_x}{\delta(1-\lambda)^2}\left(\frac{L^2}{\rho_x K} + \frac{100L^4}{3\rho_y\mu^2 K}\right) ,$$

$$c_{10} = \frac{11000\eta\gamma_x L^4}{\mu^2\delta(1-\lambda)^2} + \frac{8640\alpha_y^2\eta\gamma_x}{\delta(1-\lambda)^2}\left(\frac{L^2}{\rho_x K} + \frac{100L^4}{3\rho_y\mu^2 K}\right) ,$$

$$C_5 = \frac{48\gamma_x\alpha_x^2\eta L^2}{\rho_x K} + \frac{4800\gamma_x\alpha_x^2\eta L^4}{3\rho_y\mu^2 K} + c_9\frac{72\alpha_x^2\eta^2}{\delta}$$

$$+ c_5\frac{1188\alpha_x^2\eta^2 L^2}{\alpha_u(1-\lambda)} + c_5\frac{317952\alpha_x^2\alpha_u\eta^2 L^2}{\delta^2(1-\lambda)}$$

$$+ c_9\frac{5723136\alpha_x^2\gamma_x^2\alpha_u^2\eta^4 L^2}{\delta^3} + c_9\frac{33048\alpha_x^2\gamma_x^2\eta^4 L^2}{\delta}$$

$$+ c_6 \frac{1188\alpha_x^2\eta^2 L^2}{\alpha_v(1-\lambda)} + c_6 \frac{317952\alpha_x^2\alpha_v\eta^2 L^2}{\delta^2(1-\lambda)}$$
$$+ c_{10} \frac{5723136\alpha_x^2\gamma_y^2\alpha_v^2\eta^4 L^2}{\delta^3} + c_{10} \frac{33048\alpha_x^2\gamma_y^2\eta^4 L^2}{\delta} ,$$

$$C_6 = \frac{48\gamma_x\alpha_y^2\eta L^2}{\rho_x K} + \frac{4800\gamma_x\alpha_y^2\eta L^4}{3\rho_y\mu^2 K} + c_{10} \frac{72\alpha_y^2\eta^2}{\delta}$$
$$+ c_5 \frac{1188\alpha_y^2\eta^2 L^2}{\alpha_u(1-\lambda)} + c_5 \frac{317952\alpha_y^2\alpha_u\eta^2 L^2}{\delta^2(1-\lambda)}$$
$$+ c_9 \frac{5723136\alpha_y^2\gamma_x^2\alpha_u^2\eta^4 L^2}{\delta^3} + c_9 \frac{33048\alpha_y^2\gamma_x^2\eta^4 L^2}{\delta}$$
$$+ c_6 \frac{1188\alpha_y^2\eta^2 L^2}{\alpha_v(1-\lambda)} + c_6 \frac{317952\alpha_y^2\alpha_v\eta^2 L^2}{\delta^2(1-\lambda)}$$
$$+ c_{10} \frac{5723136\alpha_y^2\gamma_y^2\alpha_v^2\eta^4 L^2}{\delta^3} + c_{10} \frac{33048\alpha_y^2\gamma_y^2\eta^4 L^2}{\delta} , \tag{305}$$

and

$$\gamma_x \leq \min \left\{ \frac{\delta^2(1-\lambda)}{40998L}, \frac{\alpha_u(1-\lambda)^2}{627L}, \frac{\delta(1-\lambda)^2}{10244L}, \frac{\alpha_u\alpha_x(1-\lambda)^2}{491L}, \frac{\alpha_x\delta(1-\lambda)^2}{10244L}, \frac{\alpha_u\alpha_x(1-\lambda)^2}{627L}, \right.$$
$$\frac{\gamma_y\mu^2}{45L^2}, \frac{\mu\delta(1-\lambda)}{25329L^2}, \frac{\sqrt{\rho_x K}\delta(1-\lambda)}{22448L}, \frac{\sqrt{\rho_y K}\mu\delta(1-\lambda)}{129600L^2},$$
$$\frac{\mu^{\frac{1}{2}}\delta(1-\lambda)^{\frac{1}{2}}\rho_y^{\frac{1}{4}}K^{\frac{1}{4}}}{6045L^{\frac{3}{2}}}, \frac{\delta(1-\lambda)^{\frac{1}{2}}\rho_x^{\frac{1}{4}}K^{\frac{1}{4}}}{2516L}, \frac{\mu^{\frac{1}{2}}\delta(1-\lambda)^{\frac{1}{2}}}{2673L^{\frac{3}{2}}}, \frac{\mu^{\frac{1}{2}}\delta^{\frac{1}{2}}(1-\lambda)^{\frac{1}{2}}}{737L^{\frac{3}{2}}},$$
$$\frac{\delta^{\frac{1}{2}}(1-\lambda)^{\frac{1}{2}}\rho_x^{\frac{1}{4}}K^{\frac{1}{4}}}{694L}, \frac{\mu^{\frac{1}{2}}\delta^{\frac{1}{2}}(1-\lambda)^{\frac{1}{2}}\rho_y^{\frac{1}{4}}K^{\frac{1}{4}}}{1667L^{\frac{3}{2}}}, \frac{\alpha_u^{\frac{1}{2}}\alpha_x^{\frac{1}{2}}\mu^{\frac{1}{2}}(1-\lambda)}{331L^{\frac{3}{2}}},$$
$$\left. \frac{\alpha_u^{\frac{1}{2}}(1-\lambda)\rho_x^{\frac{1}{4}}K^{\frac{1}{4}}}{352L}, \frac{\alpha_u^{\frac{1}{2}}\mu^{\frac{1}{2}}(1-\lambda)\rho_y^{\frac{1}{4}}K^{\frac{1}{4}}}{845L^{\frac{3}{2}}}, \frac{\alpha_u^{\frac{1}{2}}\alpha_x^{\frac{1}{2}}\mu^{\frac{1}{2}}(1-\lambda)}{331L^{\frac{3}{2}}}, \frac{\alpha_u^{\frac{1}{2}}(1-\lambda)\rho_x^{\frac{1}{4}}K^{\frac{1}{4}}}{352L}, \frac{\alpha_u^{\frac{1}{2}}\mu^{\frac{1}{2}}(1-\lambda)\rho_y^{\frac{1}{4}}K^{\frac{1}{4}}}{845L^{\frac{3}{2}}} \right\}, \tag{306}$$

and

$$\gamma_y \leq \min \left\{ \frac{1}{6L}, \frac{\delta^2(1-\lambda)}{40998L}, \frac{\delta(1-\lambda)^2}{10244L}, \frac{\alpha_v\alpha_y(1-\lambda)^2}{627L}, \frac{\alpha_v(1-\lambda)^2}{627L}, \frac{\alpha_y\delta(1-\lambda)^2}{10244L}, \frac{\alpha_v\alpha_y\mu(1-\lambda)}{109059L^2}, \right.$$
$$\frac{\alpha_v(1-\lambda)\rho_x^{\frac{1}{2}}K^{\frac{1}{2}}}{123461L}, \frac{\alpha_v\mu(1-\lambda)\rho_y^{\frac{1}{2}}K^{\frac{1}{2}}}{712800L^2}, \frac{\alpha_y\mu\delta(1-\lambda)}{1784155L^2}, \frac{\delta(1-\lambda)\rho_x^{\frac{1}{2}}K^{\frac{1}{2}}}{2019766L}, \frac{\mu\delta(1-\lambda)\rho_y^{\frac{1}{2}}K^{\frac{1}{2}}}{11661120L^2},$$
$$\frac{\mu\delta^2}{7140949L^2}, \frac{\delta^2\rho_x^{\frac{1}{2}}K^{\frac{1}{2}}}{6328728L}, \frac{\mu\delta^2\rho_y^{\frac{1}{2}}K^{\frac{1}{2}}}{36538927L^2}, \frac{\delta\rho_x^{\frac{1}{2}}K^{\frac{1}{2}}}{480920L}, \frac{\mu\delta\rho_y^{\frac{1}{2}}K^{\frac{1}{2}}}{2776588L^2}, \frac{\rho_x K}{432\mu}, \frac{3\mu\rho_y K}{43200L^2},$$
$$\frac{\mu\delta^2(1-\lambda)^2}{21384000L^2}, \frac{\delta^2(1-\lambda)^2\rho_x K}{16796160\mu}, \frac{\mu\delta^2(1-\lambda)^2\rho_y K}{559872000L^2},$$
$$\frac{\mu^{\frac{1}{3}}\delta^{\frac{4}{3}}(1-\lambda)^{\frac{2}{3}}}{11935L^{\frac{4}{3}}}, \frac{\delta^{\frac{4}{3}}(1-\lambda)^{\frac{2}{3}}\rho_x^{\frac{1}{3}}K^{\frac{1}{3}}}{11012\mu^{\frac{1}{3}}L^{\frac{2}{3}}}, \frac{\mu^{\frac{1}{3}}\delta^{\frac{4}{3}}(1-\lambda)^{\frac{2}{3}}\rho_y^{\frac{1}{3}}K^{\frac{1}{3}}}{35438L^{\frac{4}{3}}},$$
$$\frac{\alpha_u^2\mu}{396459360L^2}, \frac{\alpha_u^2\rho_x K}{508083840\mu\alpha_x^2}, \frac{\alpha_u^2\mu\rho_y K}{16936128000\alpha_x^2 L^2},$$
$$\frac{\mu}{106106941440L^2}, \frac{\rho_x K}{135981711360\mu\alpha_x^2}, \frac{\mu\rho_y K}{4532723712000\alpha_x^2 L^2},$$
$$\frac{\mu\delta^2}{1699771392000L^2}, \frac{\delta^2\rho_x K}{1335093166080\mu}, \frac{\mu\delta^2\rho_y K}{44503105536000L^2},$$
$$\left. \frac{\alpha_v^{\frac{2}{3}}\alpha_y^{\frac{2}{3}}\mu^{\frac{1}{3}}(1-\lambda)^{\frac{4}{3}}}{735L^{\frac{4}{3}}}, \frac{\alpha_v^{\frac{2}{3}}(1-\lambda)^{\frac{4}{3}}\rho_x^{\frac{1}{3}}K^{\frac{1}{3}}}{798\mu^{\frac{1}{3}}L^{\frac{2}{3}}}, \frac{\alpha_v^{\frac{2}{3}}\mu^{\frac{1}{3}}(1-\lambda)^{\frac{4}{3}}\rho_y^{\frac{1}{3}}K^{\frac{1}{3}}}{2569L^{\frac{4}{3}}}, \right.$$

$$\left. \frac{\alpha_y^{\frac{2}{3}}\mu^{\frac{1}{3}}\delta^{\frac{2}{3}}(1-\lambda)^{\frac{4}{3}}}{4735L^{\frac{4}{3}}}, \frac{\delta^{\frac{2}{3}}(1-\lambda)^{\frac{4}{3}}\rho_x^{\frac{1}{3}}K^{\frac{1}{3}}}{5143\mu^{\frac{1}{3}}L^{\frac{2}{3}}}, \frac{\mu^{\frac{1}{3}}\delta^{\frac{2}{3}}(1-\lambda)^{\frac{4}{3}}\rho_y^{\frac{1}{3}}K^{\frac{1}{3}}}{16550L^{\frac{4}{3}}} \right\}, \tag{307}$$

and

$$\alpha_x \le \min\left\{ \frac{\delta^2}{1053}, \frac{\delta(1-\lambda)}{146} \right\},$$
$$\alpha_y \le \min\left\{ \frac{\delta^2}{1053}, \frac{\delta(1-\lambda)}{146} \right\},$$
$$\alpha_u \le \frac{\delta(1-\lambda)}{263},$$
$$\alpha_v \le \frac{\delta(1-\lambda)}{263}, \tag{308}$$

and

$$\eta \le \min\left\{ 1, \frac{1}{\sqrt{\rho_x}}, \frac{1}{\sqrt{\rho_y}}, \frac{1}{\alpha_x}, \frac{1}{\alpha_y}, \frac{1}{2\gamma_x L_F} \right\}, \quad \rho_x > 0, \quad \rho_y > 0, \tag{309}$$

we can obtain

$$P_{t+1} - P_t$$
$$\le -\frac{\gamma_x \eta}{2}\mathbb{E}\left[ \|\nabla F(\bar{x}_t)\|^2 \right]$$
$$+ c_1 \frac{2\rho_x^2\eta^4\sigma^2}{K} + c_2 \frac{2\rho_y^2\eta^4\sigma^2}{K} + c_{11}2\rho_x^2\eta^4\sigma^2 + c_{12}2\rho_y^2\eta^4\sigma^2 + C_3 3\rho_x^2\eta^4\sigma^2 + C_4 3\rho_y^2\eta^4\sigma^2$$
$$\le -\frac{\gamma_x \eta}{2}\mathbb{E}\left[ \|\nabla F(\bar{x}_t)\|^2 \right]$$
$$+ \frac{2\gamma_x}{\rho_x\eta}\frac{2\rho_x^2\eta^4\sigma^2}{K} + \frac{200\gamma_x L^2}{3\rho_y\eta\mu^2}\frac{2\rho_y^2\eta^4\sigma^2}{K} + c_{11}2\rho_x^2\eta^4\sigma^2 + c_{12}2\rho_y^2\eta^4\sigma^2 + c_{11}\rho_x\eta^2\sigma^2 + c_{12}\rho_y\eta^2\sigma^2, \tag{310}$$

where the last step holds due to $C_3 \le c_{11}\frac{1}{3\rho_x\eta^2}$ and $C_4 \le c_{12}\frac{1}{3\rho_x\eta^2}$ in Eq. (51) and Eq. (53).

By summing over $t$ from $0$ to $T-1$, we can obtain

$$\frac{1}{T}\sum_{t=0}^{T-1}\mathbb{E}\left[ \|\nabla F(\bar{x}_t)\|^2 \right]$$
$$\le \frac{2(P_0 - F(x_*))}{\gamma_x\eta T} + \frac{8\rho_x\eta^2\sigma^2}{K} + \frac{800\rho_y\eta^2 L^2\sigma^2}{3\mu^2 K}$$
$$+ c_{11}(2\rho_x\eta^2 + 1)\frac{2\rho_x\eta\sigma^2}{\gamma_x} + c_{12}(2\rho_y\eta^2 + 1)\frac{2\rho_y\eta\sigma^2}{\gamma_x}. \tag{311}$$

Because $\rho_x\eta^2 < 1$, $\rho_y\eta^2 < 1$, $\delta < 1$, $\alpha_u < 1$, and $\alpha_v < 1$, we can obtain

$$c_{11} = c_5\frac{33\eta^2\rho_x}{\alpha_u(1-\lambda)} + c_5\frac{8832\alpha_u\eta^2\rho_x}{\delta^2(1-\lambda)} + c_9\frac{158976\gamma_x^2\alpha_u^2\eta^4\rho_x}{\delta^3} + c_9\frac{918\gamma_x^2\eta^4\rho_x}{\delta}$$
$$\le c_5\eta^2\rho_x\frac{33}{\alpha_u\delta^2(1-\lambda)} + c_5\eta^2\rho_x\frac{8832\alpha_u}{\alpha_u\delta^2(1-\lambda)} + c_9\gamma_x^2\eta^4\rho_x\frac{158976\alpha_u^2}{\delta^3} + c_9\gamma_x^2\eta^4\rho_x\frac{918}{\delta^3}$$
$$\le c_5\eta^2\rho_x\frac{8865}{\alpha_u\delta^2(1-\lambda)} + c_9\gamma_x^2\eta^4\rho_x\frac{159894}{\delta^3},$$
$$c_{12} = c_6\frac{33\eta^2\rho_y}{\alpha_v(1-\lambda)} + c_6\frac{8832\alpha_v\eta^2\rho_y}{\delta^2(1-\lambda)} + c_{10}\frac{158976\gamma_y^2\alpha_v^2\eta^4\rho_y}{\delta^3} + c_{10}\frac{918\gamma_y^2\eta^4\rho_y}{\delta}$$
$$\le c_6\eta^2\rho_y\frac{33}{\alpha_v\delta^2(1-\lambda)} + c_6\eta^2\rho_y\frac{8832\alpha_v}{\alpha_v\delta^2(1-\lambda)} + c_{10}\gamma_y^2\eta^4\rho_y\frac{158976\alpha_v^2}{\delta^3} + c_{10}\gamma_y^2\eta^4\rho_y\frac{918}{\delta^3}$$

$$\leq c_6\eta^2\rho_y\frac{8865}{\alpha_v\delta^2(1-\lambda)} + c_{10}\gamma_y^2\eta^4\rho_y\frac{159894}{\delta^3} \ , \tag{312}$$

and then we can obtain

$$\frac{1}{T}\sum_{t=0}^{T-1}\mathbb{E}\Big[\|\nabla F(\bar{x}_t)\|^2\Big]$$

$$\leq \frac{2(P_0 - F(x_*))}{\gamma_x\eta T} + \frac{8\rho_x\eta^2\sigma^2}{K} + \frac{800\rho_y\eta^2 L^2\sigma^2}{3\mu^2 K} + c_{11}\frac{6\rho_x\eta\sigma^2}{\gamma_x} + c_{12}\frac{6\rho_y\eta\sigma^2}{\gamma_x}$$

$$\leq \frac{2(P_0 - F(x_*))}{\gamma_x\eta T} + \frac{8\rho_x\eta^2\sigma^2}{K} + \frac{800\rho_y\eta^2 L^2\sigma^2}{3\mu^2 K}$$

$$+ \left(c_5\eta^2\rho_x\frac{8865}{\alpha_u\delta^2(1-\lambda)} + c_9\gamma_x^2\eta^4\rho_x\frac{159894}{\delta^3}\right)\frac{6\rho_x\eta\sigma^2}{\gamma_x}$$

$$+ \left(c_6\eta^2\rho_y\frac{8865}{\alpha_v\delta^2(1-\lambda)} + c_{10}\gamma_y^2\eta^4\rho_y\frac{159894}{\delta^3}\right)\frac{6\rho_y\eta\sigma^2}{\gamma_x}$$

$$\leq \frac{2(P_0 - F(x_*))}{\gamma_x\eta T} + \frac{8\rho_x\eta^2\sigma^2}{K} + \frac{800\rho_y\eta^2 L^2\sigma^2}{3\mu^2 K}$$

$$+ \left(\frac{12360 L^4}{\alpha_u\alpha_x^2\mu^2(1-\lambda)^3} + \frac{15840}{\alpha_u(1-\lambda)^3}\Big(\frac{L^2}{\rho_x K} + \frac{100 L^4}{3\rho_y\mu^2 K}\Big)\right)\frac{8865\times 6\gamma_x^2\rho_x^2\eta^4\sigma^2}{\alpha_u\delta^2(1-\lambda)}$$

$$+ \left(\frac{11000 L^4}{\mu^2\delta(1-\lambda)^2} + \frac{8640\alpha_x^2}{\delta(1-\lambda)^2}\Big(\frac{L^2}{\rho_x K} + \frac{100 L^4}{3\rho_y\mu^2 K}\Big)\right)\frac{159894\times 6\gamma_x^2\eta^6\rho_x^2\sigma^2}{\delta^3}$$

$$+ \left(\frac{12360 L^4}{\alpha_v\alpha_y^2\mu^2(1-\lambda)^3} + \frac{15840}{\alpha_v(1-\lambda)^3}\Big(\frac{L^2}{\rho_x K} + \frac{100 L^4}{3\rho_y\mu^2 K}\Big)\right)\frac{8865\times 6\gamma_y^2\rho_y^2\eta^4\sigma^2}{\alpha_v\delta^2(1-\lambda)}$$

$$+ \left(\frac{11000 L^4}{\mu^2\delta(1-\lambda)^2} + \frac{8640\alpha_y^2}{\delta(1-\lambda)^2}\Big(\frac{L^2}{\rho_x K} + \frac{100 L^4}{3\rho_y\mu^2 K}\Big)\right)\frac{159894\times 6\gamma_y^2\eta^6\rho_y^2\sigma^2}{\delta^3} \ . \tag{313}$$

When $t = 0$, we have $x_0^{(k)} = x_0$ and $y_0^{(k)} = y_0$, then

$$P_0 = \mathbb{E}[F(\bar{x}_0)] + c_0\mathbb{E}[\|\bar{y}_0 - y^*(\bar{x}_0)\|^2]$$

$$+ c_1\mathbb{E}\Big[\Big\|\frac{1}{K}\sum_{k=1}^K g_0^{(k)} - \frac{1}{K}\sum_{k=1}^K \nabla_x f^{(k)}(x_0^{(k)}, y_0^{(k)})\Big\|^2\Big] + c_2\mathbb{E}\Big[\Big\|\frac{1}{K}\sum_{k=1}^K h_0^{(k)} - \frac{1}{K}\sum_{k=1}^K \nabla_y f^{(k)}(x_0^{(k)}, y_0^{(k)})\Big\|^2\Big]$$

$$+ c_5\frac{1}{K}\mathbb{E}[\|U_0 - \bar{U}_0\|_F^2] + c_6\frac{1}{K}\mathbb{E}[\|V_0 - \bar{V}_0\|_F^2] + c_7\frac{1}{K}\mathbb{E}[\|\hat{U}_0 - U_0\|_F^2] + c_8\frac{1}{K}\mathbb{E}[\|\hat{V}_0 - V_0\|_F^2]$$

$$+ c_9\frac{1}{K}\mathbb{E}[\|\tilde{X}_1 - \hat{X}_0\|_F^2] + c_{10}\frac{1}{K}\mathbb{E}[\|\tilde{Y}_1 - \hat{Y}_0\|_F^2]$$

$$+ c_{11}\frac{1}{K}\sum_{k=1}^K\mathbb{E}\Big[\Big\|g_0^{(k)} - \nabla_x f^{(k)}(x_0^{(k)}, y_0^{(k)})\Big\|^2\Big] + c_{12}\frac{1}{K}\sum_{k=1}^K\mathbb{E}\Big[\Big\|h_0^{(k)} - \nabla_y f^{(k)}(x_0^{(k)}, y_0^{(k)})\Big\|^2\Big]$$

$$= \mathbb{E}[F(\bar{x}_0)] + c_0\mathbb{E}[\|\bar{y}_0 - y^*(\bar{x}_0)\|^2]$$

$$+ c_1\mathbb{E}\Big[\Big\|\frac{1}{K}\sum_{k=1}^K g_0^{(k)} - \frac{1}{K}\sum_{k=1}^K \nabla_x f^{(k)}(x_0^{(k)}, y_0^{(k)})\Big\|^2\Big] + c_2\mathbb{E}\Big[\Big\|\frac{1}{K}\sum_{k=1}^K h_0^{(k)} - \frac{1}{K}\sum_{k=1}^K \nabla_y f^{(k)}(x_0^{(k)}, y_0^{(k)})\Big\|^2\Big]$$

$$+ c_5\frac{1}{K}\mathbb{E}[\|U_0 - \bar{U}_0\|_F^2] + c_6\frac{1}{K}\mathbb{E}[\|V_0 - \bar{V}_0\|_F^2]$$

$$+ (c_5\frac{64\alpha_u}{\delta(1-\lambda)} + c_9\frac{1152\gamma_x^2\alpha_u^2\eta^2}{\delta^2})\frac{1}{K}\mathbb{E}[\|\hat{U}_0 - U_0\|_F^2]$$

$$+ (c_6\frac{64\alpha_v}{\delta(1-\lambda)} + c_{10}\frac{1152\gamma_y^2\alpha_v^2\eta^2}{\delta^2})\frac{1}{K}\mathbb{E}[\|\hat{V}_0 - V_0\|_F^2]$$

$$+ c_9 \frac{1}{K}\mathbb{E}[\|\tilde{X}_1 - \hat{X}_0\|_F^2] + c_{10}\frac{1}{K}\mathbb{E}[\|\tilde{Y}_1 - \hat{Y}_0\|_F^2]$$

$$+ (c_5\frac{33\eta^2\rho_x}{\alpha_u(1-\lambda)} + c_5\frac{8832\alpha_u\eta^2\rho_x}{\delta^2(1-\lambda)} + c_9\frac{158976\gamma_x^2\alpha_u^2\eta^4\rho_x}{\delta^3} + c_9\frac{918\gamma_x^2\eta^4\rho_x}{\delta})\frac{1}{K}\sum_{k=1}^K \mathbb{E}\left[\left\|g_0^{(k)} - \nabla_x f^{(k)}(x_0^{(k)}, y_0^{(k)})\right\|^2\right]$$

$$+ (c_6\frac{33\eta^2\rho_y}{\alpha_v(1-\lambda)} + c_6\frac{8832\alpha_v\eta^2\rho_y}{\delta^2(1-\lambda)} + c_{10}\frac{158976\gamma_y^2\alpha_v^2\eta^4\rho_y}{\delta^3} + c_{10}\frac{918\gamma_y^2\eta^4\rho_y}{\delta})\frac{1}{K}\sum_{k=1}^K \mathbb{E}\left[\left\|h_0^{(k)} - \nabla_y f^{(k)}(x_0^{(k)}, y_0^{(k)})\right\|^2\right] .$$

$$\tag{314}$$

Because the initial batch size is $B$, we have

$$P_0 \leq \mathbb{E}[F(\bar{x}_0)] + c_0\mathbb{E}[\|\bar{y}_0 - y^*(\bar{x}_0)\|^2] + c_1\frac{\sigma^2}{B} + c_2\frac{\sigma^2}{B}$$

$$+ c_5(6\frac{\sigma^2}{B} + 6\frac{1}{K}\sum_{k=1}^K \mathbb{E}[\|\nabla_x f^{(k)}(x_0, y_0)\|^2]) + c_6(6\frac{\sigma^2}{B} + 6\frac{1}{K}\sum_{k=1}^K \mathbb{E}[\|\nabla_y f^{(k)}(x_0, y_0)\|^2])$$

$$+ (c_5\frac{64\alpha_u}{\delta(1-\lambda)} + c_9\frac{1152\gamma_x^2\alpha_u^2\eta^2}{\delta^2})(\frac{\sigma^2}{B} + \frac{1}{K}\sum_{k=1}^K \mathbb{E}[\|\nabla_x f^{(k)}(x_0, y_0)\|^2])$$

$$+ (c_6\frac{64\alpha_v}{\delta(1-\lambda)} + c_{10}\frac{1152\gamma_y^2\alpha_v^2\eta^2}{\delta^2})(\frac{\sigma^2}{B} + \frac{1}{K}\sum_{k=1}^K \mathbb{E}[\|\nabla_y f^{(k)}(x_0, y_0)\|^2])$$

$$+ c_9(\gamma_x^2\eta^2\frac{\sigma^2}{B} + \gamma_x^2\eta^2\frac{1}{K}\sum_{k=1}^K \mathbb{E}[\|\nabla_x f^{(k)}(x_0, y_0)\|^2]) + c_{10}(\gamma_y^2\eta^2\frac{\sigma^2}{B} + \gamma_y^2\eta^2\frac{1}{K}\sum_{k=1}^K \mathbb{E}[\|\nabla_y f^{(k)}(x_0, y_0)\|^2])$$

$$+ (c_5\frac{33\eta^2\rho_x}{\alpha_u(1-\lambda)} + c_5\frac{8832\alpha_u\eta^2\rho_x}{\delta^2(1-\lambda)} + c_9\frac{158976\gamma_x^2\alpha_u^2\eta^4\rho_x}{\delta^3} + c_9\frac{918\gamma_x^2\eta^4\rho_x}{\delta})\frac{\sigma^2}{B}$$

$$+ (c_6\frac{33\eta^2\rho_y}{\alpha_v(1-\lambda)} + c_6\frac{8832\alpha_v\eta^2\rho_y}{\delta^2(1-\lambda)} + c_{10}\frac{158976\gamma_y^2\alpha_v^2\eta^4\rho_y}{\delta^3} + c_{10}\frac{918\gamma_y^2\eta^4\rho_y}{\delta})\frac{\sigma^2}{B}$$

$$= \mathbb{E}[F(\bar{x}_0)] + c_0\mathbb{E}[\|\bar{y}_0 - y^*(\bar{x}_0)\|^2] + c_1\frac{\sigma^2}{B} + c_2\frac{\sigma^2}{B}$$

$$+ c_5(6 + \frac{64\alpha_u}{\delta(1-\lambda)} + \frac{8865\eta^2\rho_x}{\alpha_u\delta^2(1-\lambda)})\frac{\sigma^2}{B} + c_6(6 + \frac{64\alpha_v}{\delta(1-\lambda)} + \frac{8865\eta^2\rho_y}{\alpha_v\delta^2(1-\lambda)})\frac{\sigma^2}{B}$$

$$+ c_9(\frac{1152\gamma_x^2\alpha_u^2\eta^2}{\delta^2} + \gamma_x^2\eta^2 + \frac{159894\gamma_x^2\eta^4\rho_x}{\delta^3})\frac{\sigma^2}{B} + c_{10}(\frac{1152\gamma_y^2\alpha_v^2\eta^2}{\delta^2} + \gamma_y^2\eta^2 + \frac{159894\gamma_y^2\eta^4\rho_y}{\delta^3})\frac{\sigma^2}{B}$$

$$+ c_5(6 + \frac{64\alpha_u}{\delta(1-\lambda)})\frac{1}{K}\sum_{k=1}^K \mathbb{E}[\|\nabla_x f^{(k)}(x_0, y_0)\|^2]) + c_9(\frac{1152\gamma_x^2\alpha_u^2\eta^2}{\delta^2} + \gamma_x^2\eta^2)\frac{1}{K}\sum_{k=1}^K \mathbb{E}[\|\nabla_x f^{(k)}(x_0, y_0)\|^2])$$

$$+ c_6(6 + \frac{64\alpha_v}{\delta(1-\lambda)})\frac{1}{K}\sum_{k=1}^K \mathbb{E}[\|\nabla_y f^{(k)}(x_0, y_0)\|^2]) + c_{10}(\frac{1152\gamma_y^2\alpha_v^2\eta^2}{\delta^2} + \gamma_y^2\eta^2)\frac{1}{K}\sum_{k=1}^K \mathbb{E}[\|\nabla_y f^{(k)}(x_0, y_0)\|^2]$$

$$\leq \mathbb{E}[F(\bar{x}_0)] + c_0\mathbb{E}[\|\bar{y}_0 - y^*(\bar{x}_0)\|^2] + c_1\frac{\sigma^2}{B} + c_2\frac{\sigma^2}{B}$$

$$+ c_5\frac{8935}{\alpha_u\delta^2(1-\lambda)}\frac{\sigma^2}{B} + c_6\frac{8935}{\alpha_v\delta^2(1-\lambda)}\frac{\sigma^2}{B} + c_9\frac{161047\gamma_x^2\eta^2}{\delta^3}\frac{\sigma^2}{B} + c_{10}\frac{161047\gamma_y^2\eta^2}{\delta^3}\frac{\sigma^2}{B}$$

$$+ \left(c_5\frac{70}{\delta(1-\lambda)} + c_9\frac{1153\gamma_x^2\eta^2}{\delta^2}\right)\frac{1}{K}\sum_{k=1}^K \mathbb{E}[\|\nabla_x f^{(k)}(x_0, y_0)\|^2])$$

$$+ \left(c_6\frac{70}{\delta(1-\lambda)} + c_{10}\frac{1153\gamma_y^2\eta^2}{\delta^2}\right)\frac{1}{K}\sum_{k=1}^K \mathbb{E}[\|\nabla_y f^{(k)}(x_0, y_0)\|^2]) ,$$

$$\tag{315}$$

where the last step holds due to $\delta < 1$, $1 - \lambda < 1$, $\alpha_u < 1$, and $\alpha_v < 1$.

Finally, we can obtain

$$
\frac{1}{T}\sum_{t=0}^{T-1}\mathbb{E}\Big[\|\nabla F(\bar{x}_t)\|^2\Big]
$$

$$
\leq \frac{2(P_0 - F(x_*))}{\gamma_x \eta T} + \frac{8\rho_x \eta^2 \sigma^2}{K} + \frac{800\rho_y \eta^2 L^2 \sigma^2}{3\mu^2 K}
$$

$$
+ \left(\frac{12360L^4}{\alpha_u \alpha_x^2 \mu^2 (1-\lambda)^3} + \frac{15840}{\alpha_u (1-\lambda)^3}\Big(\frac{L^2}{\rho_x K} + \frac{100L^4}{3\rho_y \mu^2 K}\Big)\right)\frac{8865 \times 6\gamma_x^2 \rho_x^2 \eta^4 \sigma^2}{\alpha_u \delta^2 (1-\lambda)}
$$

$$
+ \left(\frac{11000L^4}{\mu^2 \delta (1-\lambda)^2} + \frac{8640\alpha_x^2}{\delta (1-\lambda)^2}\Big(\frac{L^2}{\rho_x K} + \frac{100L^4}{3\rho_y \mu^2 K}\Big)\right)\frac{159894 \times 6\gamma_x^2 \eta^6 \rho_x^2 \sigma^2}{\delta^3}
$$

$$
+ \left(\frac{12360L^4}{\alpha_v \alpha_y^2 \mu^2 (1-\lambda)^3} + \frac{15840}{\alpha_v (1-\lambda)^3}\Big(\frac{L^2}{\rho_x K} + \frac{100L^4}{3\rho_y \mu^2 K}\Big)\right)\frac{8865 \times 6\gamma_y^2 \rho_y^2 \eta^4 \sigma^2}{\alpha_v \delta^2 (1-\lambda)}
$$

$$
+ \left(\frac{11000L^4}{\mu^2 \delta (1-\lambda)^2} + \frac{8640\alpha_y^2}{\delta (1-\lambda)^2}\Big(\frac{L^2}{\rho_x K} + \frac{100L^4}{3\rho_y \mu^2 K}\Big)\right)\frac{159894 \times 6\gamma_y^2 \eta^6 \rho_y^2 \sigma^2}{\delta^3}
$$

$$
\leq \frac{2(F(x_0) - F(x_*))}{\gamma_x \eta T} + \frac{16L^2}{\eta \gamma_y \mu T}\mathbb{E}[\|\bar{y}_0 - y^*(\bar{x}_0)\|^2] + \frac{4}{\rho_x \eta^2 T}\frac{\sigma^2}{B} + \frac{400L^2}{3\rho_y \eta^2 \mu^2 T}\frac{\sigma^2}{B}
$$

$$
+ \left(\frac{220873200\gamma_x^2 L^4}{\alpha_u^2 \alpha_x^2 \mu^2 \delta^2 (1-\lambda)^4 T} + \frac{283060800\gamma_x^2}{\alpha_u^2 \delta^2 (1-\lambda)^4 T}\Big(\frac{L^2}{\rho_x K} + \frac{100L^4}{3\rho_y \mu^2 K}\Big)\right)\frac{\sigma^2}{B}
$$

$$
+ \left(\frac{220873200\gamma_y^2 L^4}{\alpha_v^2 \alpha_y^2 \mu^2 \delta^2 (1-\lambda)^4 T} + \frac{283060800\gamma_y^2}{\alpha_v^2 \delta^2 (1-\lambda)^4 T}\Big(\frac{L^2}{\rho_x K} + \frac{100L^4}{3\rho_y \mu^2 K}\Big)\right)\frac{\sigma^2}{B}
$$

$$
+ \left(\frac{3543034000\eta^2 \gamma_x^2 L^4}{\mu^2 \delta^4 (1-\lambda)^2 T} + \frac{2782892160\alpha_x^2 \eta^2 \gamma_x^2}{\delta^4 (1-\lambda)^2 T}\Big(\frac{L^2}{\rho_x K} + \frac{100L^4}{3\rho_y \mu^2 K}\Big)\right)\frac{\sigma^2}{B}
$$

$$
+ \left(\frac{3543034000\eta^2 \gamma_y^2 L^4}{\mu^2 \delta^4 (1-\lambda)^2 T} + \frac{2782892160\alpha_y^2 \eta^2 \gamma_y^2}{\delta^4 (1-\lambda)^2 T}\Big(\frac{L^2}{\rho_x K} + \frac{100L^4}{3\rho_y \mu^2 K}\Big)\right)\frac{\sigma^2}{B}
$$

$$
+ \left(\frac{1730400\gamma_x^2 L^4}{\alpha_u \alpha_x^2 \mu^2 \delta (1-\lambda)^4 T} + \frac{2217600\gamma_x^2}{\alpha_u \delta (1-\lambda)^4 T}\Big(\frac{L^2}{\rho_x K} + \frac{100L^4}{3\rho_y \mu^2 K}\Big) + \frac{25366000\eta^2 \gamma_x^2 L^4}{\mu^2 \delta^3 (1-\lambda)^2 T}\right.
$$

$$
\left. + \frac{19923840\alpha_x^2 \eta^2 \gamma_x^2}{\delta^3 (1-\lambda)^2 T}\Big(\frac{L^2}{\rho_x K} + \frac{100L^4}{3\rho_y \mu^2 K}\Big)\right)\frac{1}{K}\sum_{k=1}^{K}\mathbb{E}[\|\nabla_x f^{(k)}(x_0, y_0)\|^2]
$$

$$
+ \left(\frac{1730400\gamma_y^2 L^4}{\alpha_v \alpha_y^2 \mu^2 \delta (1-\lambda)^4 T} + \frac{2217600\gamma_y^2}{\alpha_v \delta (1-\lambda)^4 T}\Big(\frac{L^2}{\rho_x K} + \frac{100L^4}{3\rho_y \mu^2 K}\Big) + \frac{25366000\eta^2 \gamma_y^2 L^4}{\mu^2 \delta^3 (1-\lambda)^2 T}\right.
$$

$$
\left. + \frac{19923840\alpha_y^2 \eta^2 \gamma_y^2}{\delta^3 (1-\lambda)^2 T}\Big(\frac{L^2}{\rho_x K} + \frac{100L^4}{3\rho_y \mu^2 K}\Big)\right)\frac{1}{K}\sum_{k=1}^{K}\mathbb{E}[\|\nabla_y f^{(k)}(x_0, y_0)\|^2]
$$

$$
+ \frac{8\rho_x \eta^2 \sigma^2}{K} + \frac{800\rho_y \eta^2 L^2 \sigma^2}{3\mu^2 K}
$$

$$
+ \left(\frac{12360L^4}{\alpha_u \alpha_x^2 \mu^2 (1-\lambda)^3} + \frac{15840}{\alpha_u (1-\lambda)^3}\Big(\frac{L^2}{\rho_x K} + \frac{100L^4}{3\rho_y \mu^2 K}\Big)\right)\frac{8865 \times 6\gamma_x^2 \rho_x^2 \eta^4 \sigma^2}{\alpha_u \delta^2 (1-\lambda)}
$$

$$
+ \left(\frac{11000L^4}{\mu^2 \delta (1-\lambda)^2} + \frac{8640\alpha_x^2}{\delta (1-\lambda)^2}\Big(\frac{L^2}{\rho_x K} + \frac{100L^4}{3\rho_y \mu^2 K}\Big)\right)\frac{159894 \times 6\gamma_x^2 \eta^6 \rho_x^2 \sigma^2}{\delta^3}
$$

$$
+ \left(\frac{12360L^4}{\alpha_v \alpha_y^2 \mu^2 (1-\lambda)^3} + \frac{15840}{\alpha_v (1-\lambda)^3}\Big(\frac{L^2}{\rho_x K} + \frac{100L^4}{3\rho_y \mu^2 K}\Big)\right)\frac{8865 \times 6\gamma_y^2 \rho_y^2 \eta^4 \sigma^2}{\alpha_v \delta^2 (1-\lambda)}
$$

$$
+ \left(\frac{11000L^4}{\mu^2 \delta (1-\lambda)^2} + \frac{8640\alpha_y^2}{\delta (1-\lambda)^2}\Big(\frac{L^2}{\rho_x K} + \frac{100L^4}{3\rho_y \mu^2 K}\Big)\right)\frac{159894 \times 6\gamma_y^2 \eta^6 \rho_y^2 \sigma^2}{\delta^3} \, . \tag{316}
$$

Then, it is easy to obtain

$$
\begin{aligned}
\frac{1}{T} &\sum_{t=0}^{T-1} \mathbb{E}\Big[\|\nabla F(\bar{x}_t)\|^2\Big] \\
&\leq O\left(\frac{1}{\gamma_x \eta T}\right)(F(x_0) - F(x_*)) + O\left(\frac{\kappa L}{\eta \gamma_y T}\right)\mathbb{E}[\|y_0 - y^*(x_0)\|^2] \\
&+ O\left(\frac{1}{TB}\left(\frac{\gamma_x^2 \kappa^2 L^2}{\alpha_u^2 \alpha_x^2 \delta^2 (1-\lambda)^4} + \frac{\gamma_y^2 \kappa^2 L^2}{\alpha_v^2 \alpha_y^2 \delta^2 (1-\lambda)^4} + \frac{\eta^2 \gamma_x^2 \kappa^2 L^2}{\delta^4 (1-\lambda)^2} + \frac{\eta^2 \gamma_y^2 \kappa^2 L^2}{\delta^4 (1-\lambda)^2}\right)\right)\sigma^2 \\
&+ O\left(\frac{1}{TB}\left(\frac{\gamma_x^2}{\alpha_u^2 \delta^2 (1-\lambda)^4} + \frac{\gamma_y^2}{\alpha_v^2 \delta^2 (1-\lambda)^4} + \frac{\alpha_x^2 \eta^2 \gamma_x^2}{\delta^4 (1-\lambda)^2} + \frac{\alpha_y^2 \eta^2 \gamma_y^2}{\delta^4 (1-\lambda)^2}\right)\left(\frac{L^2}{\rho_x K} + \frac{\kappa^2 L^2}{\rho_y K}\right)\right)\sigma^2 \\
&+ O\left(\frac{1}{T}\left(\frac{\gamma_x^2 \kappa^2 L^2}{\alpha_u \alpha_x^2 \delta (1-\lambda)^4} + \frac{\eta^2 \gamma_x^2 \kappa^2 L^2}{\delta^3 (1-\lambda)^2}\right)\right. \\
&\quad \left. + \frac{1}{T}\left(\frac{\gamma_x^2}{\alpha_u \delta (1-\lambda)^4} + \frac{\alpha_x^2 \eta^2 \gamma_x^2}{\delta^3 (1-\lambda)^2}\right)\left(\frac{L^2}{\rho_x K} + \frac{\kappa^2 L^2}{\rho_y K}\right)\right)\frac{1}{K}\sum_{k=1}^{K}\mathbb{E}[\|\nabla_x f^{(k)}(x_0, y_0)\|^2] \\
&+ O\left(\frac{1}{T}\left(\frac{\gamma_y^2 \kappa^2 L^2}{\alpha_v \alpha_y^2 \delta (1-\lambda)^4} + \frac{\eta^2 \gamma_y^2 \kappa^2 L^2}{\delta^3 (1-\lambda)^2}\right)\right. \\
&\quad \left. + \frac{1}{T}\left(\frac{\gamma_y^2}{\alpha_v \delta (1-\lambda)^4} + \frac{\alpha_y^2 \eta^2 \gamma_y^2}{\delta^3 (1-\lambda)^2}\right)\left(\frac{L^2}{\rho_x K} + \frac{L^4}{\rho_y \mu^2 K}\right)\right)\frac{1}{K}\sum_{k=1}^{K}\mathbb{E}[\|\nabla_y f^{(k)}(x_0, y_0)\|^2] \\
&+ O\left(\frac{\gamma_x^2 \rho_x^2 \eta^4 \kappa^2 L^2}{\alpha_u^2 \alpha_x^2 \delta^2 (1-\lambda)^4} + \frac{\gamma_x^2 \eta^6 \rho_x^2 \kappa^2 L^2}{\delta^4 (1-\lambda)^2} + \frac{\gamma_y^2 \rho_y^2 \eta^4 \kappa^2 L^2}{\alpha_v^2 \alpha_y^2 \delta^2 (1-\lambda)^4} + \frac{\gamma_y^2 \eta^6 \rho_y^2 \kappa^2 L^2}{\delta^4 (1-\lambda)^2}\right)\sigma^2 \\
&+ O\left(\left(\frac{\gamma_x^2 \rho_x^2 \eta^4}{\alpha_u^2 \delta^2 (1-\lambda)^4} + \frac{\alpha_x^2 \gamma_x^2 \eta^6 \rho_x^2}{\delta^4 (1-\lambda)^2} + \frac{\gamma_y^2 \rho_y^2 \eta^4}{\alpha_v^2 \delta^2 (1-\lambda)^4} + \frac{\alpha_y^2 \gamma_y^2 \eta^6 \rho_y^2}{\delta^4 (1-\lambda)^2}\right)\left(\frac{L^2}{\rho_x K} + \frac{\kappa^2 L^2}{\rho_y K}\right)\right)\sigma^2 \\
&+ O\left(\frac{1}{\rho_x \eta^2 TB}\right)\sigma^2 + O\left(\frac{\kappa^2}{\rho_y \eta^2 TB}\right)\sigma^2 + O\left(\frac{\rho_x \eta^2}{K}\right)\sigma^2 + O\left(\frac{\rho_y \eta^2 \kappa^2}{K}\right)\sigma^2 .
\end{aligned}
\tag{317}
$$

Based on Eq. (308), we can know

$$
\begin{aligned}
\alpha_x &= O\left(\delta^2(1-\lambda)\right) , & \alpha_y &= O\left(\delta^2(1-\lambda)\right) , \\
\alpha_u &= O\left(\delta(1-\lambda)\right) , & \alpha_v &= O\left(\delta(1-\lambda)\right) .
\end{aligned}
\tag{318}
$$

In addition, according to Eq. (309), we can set

$$
\rho_x = O\left(\frac{1}{K}\right) , \qquad\qquad \rho_y = O\left(\frac{1}{K}\right) .
\tag{319}
$$

Then, based on Eqs. (306-307), we can know that

$$
\gamma_x = O\left(\frac{\delta^3(1-\lambda)^4}{\kappa^3}\right) , \qquad\qquad \gamma_y = O\left(\frac{\delta^3(1-\lambda)^4}{\kappa}\right) .
\tag{320}
$$

Based on these upper bounds, by setting $\eta = O\left(\frac{K\epsilon}{\kappa}\right)$, $B = \frac{1}{\epsilon}$, and $T = O\left(\frac{\kappa^4}{K(1-\lambda)^4 \delta^3 \epsilon^3}\right)$, we can obtain

$$
\frac{1}{T}\sum_{t=0}^{T-1}\mathbb{E}\Big[\|\nabla F(\bar{x}_t)\|^2\Big] \leq O(\epsilon^2) .
\tag{321}
$$

$\square$

