# OpenReview forum: "On the Convergence of Decentralized Stochastic Minimax Optimization Algorithm with Compressed Communication"
_ICML.cc/2026/Conference — ICML 2026 regular_

### Official Review · Reviewer_FwGp · 2026-02-27

**Soundness:** 3
**Presentation:** 2
**Significance:** 3
**Originality:** 3
**Overall Recommendation:** 4
**Confidence:** 2

**Summary:**

This paper proposes a novel communication-efficient decentralized algorithm for stochastic minimax optimization. To address the high communication costs of existing distributed minimax optimization methods, the algorithm employs a momentum technique with an error-feedback mechanism for compression. Additionally, the paper theoretically establishes the convergence rate and the influence of compression, both of which are challenging due to innovative coefficient designs that balance full-precision updates and compression residuals. Extensive experiments confirm the algorithm's effectiveness.

**Compliance With Llm Reviewing Policy:**

Affirmed.

**Final Justification:**

I think this paper makes a key contribution by proposing a novel communication-efficient decentralized algorithm for stochastic minimax optimization using an error feedback strategy. My concerns are addressed. I maintain my positive score.

**Key Questions For Authors:**

See the weaknesses

**Limitations:**

Yes

**Strengths And Weaknesses:**

**Strengths**
1. This paper proposes a novel, communication-efficient decentralized algorithm for stochastic minimax optimization using an error feedback strategy. This is a remarkable contribution as it overcomes some difficult challenges.
2. The first convergence rate is established for this type of algorithm. The main technical contribution of this theoretical result is to handle their circular dependence by choosing coefficients for different estimation errors. It can provide a theoretical reference for similar decentralized algorithms.

**Weaknesses**
1. The paper claims that including the learning rate $\eta$ in the balance coefficient for error feedback of variables is a key design. However, it seems to lack explanation. So what is its role? Is it beneficial for achieving the convergence rate? Additionally, the learning rate is included in the gradient update (see Eq.(6)) rather than being independent of the gradient tracker. Since this is an important design, I suggest evaluating its impact through theoretical analysis and experiments.
2. The convergence rate shows that when using one worker with the full-precision update, the convergence rate is faster than the previous result for a single machine. However, in this case, the compression residual is not used. Thus, what led to this difference in the rate? Is there any special proof techniques?
3. Remark 4.5 claims that $K=1$, $1-\lambda=1$, and $\delta=1$ correspond to using only one worker, with the full-precision update requiring $\alpha_x = 0$ and $\alpha_u = 0$ in Eq.(5) and Eq.(9). However, in this case, $\alpha_x = O(1)$ and $\alpha_u = O(1)$ under the conditions of Theorem 4.4. Additionally, the definition of $\delta$ is missing.
4. The main issue addressed in this paper is the high communication cost. However, it seems to lack a theoretical analysis of this cost. In the current theoretical analysis, I can only conclude that large weights on the compression residual of the update ($\alpha_x$, $\alpha_y$, $\alpha_u$, and $\alpha_v$), caused by large $1-\lambda$ and $\delta$, can lead to small iterates. But this insight does not seem to be observed in the experiments shown in the first row of Figure 1.
5. The condition in Theorem 4.4 clearly guides the selection of related hyperparameters such as $\alpha_x$, $\alpha_y$, $\alpha_u$, and $\alpha_v$. Could these conditions be validated through experiments, such as by varying $\lambda$? Would that contradict the algorithm's robustness to various hyperparameters and communication topologies, which are confirmed in Figures 2 and 4?
6. It is better to present the proof sketch in the main text so that the main theoretical conclusion is easier to understand and comment on.
7. The statements of contributions in the last three points of Section 1 appear to repeat the previous paragraph. They can be simplified to leave space for the proof sketch.

---

> ### Author Rebuttal · Authors · 2026-03-28
>
> We sincerely thank the reviewer for the valuable comments and constructive suggestions. Below, we provide our detailed response.
>
> > 1. The paper claims that including ... achieving the convergence rate?  Additionally, the learning rate is included ... experiments.
>
> Response: Regarding the first question, this design is critical to guarantee convergence. In particular, if we do not incorporate $\eta$ into the balance coefficient in Eq. (5), then we must replace $\alpha_x$ with $\alpha_x/\eta$ in all proofs. This can lead to infeasible hyperparameters. For example, in the second inequality of Eq. (172), after replacing $\alpha_x$ with $\alpha_x/\eta$, we obtain the condition $\delta(1-\lambda)\geq \text{const}$. Here, the compression hyperparameter $\delta$ and the spectral gap $1-\lambda$ are not tunable parameters. They are fixed for the given compression operator and communication topology. Therefore, we cannot guarantee the existence of a setting that satisfies this condition, and consequently, convergence cannot be ensured.
>
> Regarding the second question, Eq.(6) is just the STORM gradient estimator, and therefore we follow the original paper to set the coefficient of the momentum, where $a$ is set to $c\eta^2$ in the 9-th step of Algorithm 1 in this paper: https://arxiv.org/pdf/1905.10018.  But for the gradient tracker in Eq. (9), we do not incorporate the learning rate $\eta$ into $\alpha_u$.
>
> We will make these points clearer in the revised paper as the reviewer suggested.
>
> > 2. The convergence rate shows ...  proof techniques?
>
> Response: In the single-machine setting, our convergence rate has a better dependence on the condition number $\kappa$ than existing methods. A possible reason is that those works use overly loose upper bounds for the hyperparameters, whereas we adopt tighter bounds. For example, in Huang et al., 2022, their Theorem 9 shows that the ratio between the primal and dual variables' learning rate $\gamma/\lambda=O(1/\kappa^3)$, while ours is $\gamma_x/\gamma_y=O(1/\kappa^2)$.
>
> > 3. Remark 4.5 claims ...
>
> Response: When there is no compression, $\alpha_x$ and $\alpha_y$ do NOT exist . Then, we cannot say that the bound of $\alpha_x$ and $\alpha_y$ are $O(1)$.
>
> The definition of $\delta$ is given in Assumption 3.1. We will make it clearer in the revised version.
>
> > 4. The main issue addressed in this paper is the high communication cost ...
>
> Response: The communication cost is the product of the number of iterations and the per-iteration cost. Therefore, our communication cost is $O(T \times (\tilde{d}_x + \tilde{d}_y))$. Here, $\tilde{d}_x < d_x$ and $\tilde{d}_y < d_y$ are the dimensionality after compressing variables and gradients. Although the number of iterations $T$ depends on the compression hyperparameter $\delta$, our experiments show that this does not affect convergence performance. Specifically, Figure 1 demonstrates that our method achieves nearly the same AUC score as the full-precision baselines, indicating that the compression operation does not harm convergence. Therefore, this dependence does not significantly degrade the practical convergence rate.
>
> > 5. The condition in Theorem 4.4  ....
>
> Response: Since the spectral gap $1-\lambda$ is not easy to obtain for practical communication topologies, it is difficult to strictly follow the theoretical values in Theorem 4.1 when tuning these hyperparameters. Therefore, we can instead use grid search to select appropriate values. However, we believe that if $1-\lambda$ were available, the hyperparameters could be tuned accordingly based on the spectral gap.
>
> > 6. .... 7. ...
>
> Response: Thank you for the kind suggestions. We will revise our paper accordingly.

---

> > ### Author Rebuttal · Reviewer_FwGp · 2026-04-03
> >
> > Thanks for your rebuttal. I will keep my scores unchanged.

---

> > > ### Author Response · Authors · 2026-04-07
> > >
> > > Thank you very much for recognizing the contributions of our paper and for your consistent support. We will revise the paper following your suggestions.
> > >
> > > Our paper develops novel solutions to address the unique challenges in decentralized minimax optimization with compressed communication. We sincerely hope the reviewer will continue to support our work during the discussion phase. Thank you very much again!
> > >
> > >
> > > ## **New Evidence Supporting Our Contributions**
> > >
> > > Even for minimization problems, it is challenging to balance the full-precision update and the compressed update. For example, DoCoM (https://arxiv.org/pdf/2202.00255) studies decentralized **minimization** with the compressed communication, where it also tries to balance the full-precision and the compressed update in its Eq. (9a). However, **its proof is problematic**.  In particular, as shown in its Eq.(24), the upper bound of the balance coefficient $\gamma$ relies on $\sqrt{1-\delta/(8\bar{\omega}^2)}$, where $\delta$ and $\bar{\omega}$ are not tunable parameters as shown in its Assumption 2.2 and Assumption 2.3. However, $1-\delta/(8\bar{\omega}^2)>0$ **does NOT always hold for all settings**. Therefore, it is unclear whether DoCoM really converges for the minimization problem.
> > >
> > > In our paper, through novel designs of the balance coefficients, i.e., incorporating the learning rate into the balance coefficient as shown in Eq. (5), our algorithm is able to guarantee convergence. We believe that this design can also be applied to minimization problems to address the issue of DoCoM.
> > >
> > > In summary, our novel design and rigorous theoretical analysis make important contributions to communication-efficient decentralized optimization, which can not only advance decentralized minimax optimization but also address gaps in traditional minimization problems.
> > >
> > > We hope this helps the reviewer better assess the contributions of our paper. **Considering the significant contributions of our paper to communication-efficient decentralized optimization, we sincerely hope the reviewer might kindly consider increasing the score**.

---

### Official Review · Reviewer_WV4Z · 2026-03-06

**Soundness:** 3
**Presentation:** 2
**Significance:** 2
**Originality:** 2
**Overall Recommendation:** 4
**Confidence:** 3

**Summary:**

This paper proposes a communication-efficient decentralized stochastic gradient descent ascent with momentum algorithm (CE-DSGDAR) for stochastic minimax optimization under compressed communication. The authors address the challenge of applying error feedback to decentralized minimax optimization, which involves balancing full-precision updates with compression residuals. The paper provides theoretical convergence guarantees and empirical validation on AUC maximization and fair classification tasks.

**Compliance With Llm Reviewing Policy:**

Affirmed.

**Key Questions For Authors:**

Please see Strengths And Weaknesses. For me, the most interesting part is that the balance coefficient for the variable should depend on the learning rate η, while that for the gradient tracker is independent of η. I hope that the authors can give me more explanations of this design so that I can get more insights. Then I would like to raise scores.

**Limitations:**

yes

**Strengths And Weaknesses:**

Strength: The theoretical analysis appears sound, and the proof is notably thorough—spanning over 300 equations—which suggests a high level of technical rigor.

Weakness:

Presentation: 1) The presentation of algorithm is not good. Even though the authors introduce the key components of the algorithm, it is still not easy to understand the whole procedure. I recommend that the authors modularize the algorithm and include a clear, well-labeled diagram in the paper that illustrates its key components, accompanied by concise explanations for each module. 2)The proof sketch should not be deferred to the appendix. Since the authors claims that their proof is novel, they should explain its novelty in the main paper.

Significance: The problem is important, but I doubt whether the paper can advance understanding, capabilities, or practice in machine learning. The experiments are limited to AUC maximization and fair classification tasks. The paper does not validate on more standard minimax benchmarks like GAN training or robust optimization, which would be necessary to demonstrate broader applicability. The comparison is also limited to full-precision baselines rather than other communication-efficient minimax algorithms.

Originality: The core algorithmic components (error feedback, momentum-based gradient estimation) are adaptations of existing techniques rather than fundamentally new mechanisms. The paper's key innovation is merely applying these techniques to a new setting. I believe that the paper can give more insights if the authors can take efforts to explain why the balance coefficient for the gradient tracker does NOT depend on the learning rate, while the update of variables has the dependence on the learning rate η in
the coefficient. What is the intuition behind this design and why is it necessary? Meanwhile, the proof of Theorem 4.4 is similar to (Gao, 2022) with a similar potential function.

---

> ### Author Rebuttal · Authors · 2026-03-28
>
> We sincerely thank the reviewer for the valuable comments and constructive suggestions. Below, we provide our detailed response.
>
> > Presentation: 1) The presentation  ...
>
> Response: We will revise our writing by following the reviewer's suggestions.
>
> > Significance: The problem is important, ...
>
> Response: Following the reviewer's suggestions, we add an additional experiment regarding robust optimization. Specifically, following DM-HSGD, we apply our algorithm to the robust logistic regression problem in Section 5.1 of DM-HSGD. Here, we also use a9a dataset and add an additional baseline, DGDA-VR (https://arxiv.org/pdf/2307.09421). In this new experiment, both the loss function value with respect to epochs (https://anonymous.4open.science/r/figure-A21B/phi_vs_epoch.pdf) and that with respect to communication costs (https://anonymous.4open.science/r/figure-A21B/phi_vs_total_n_mb.pdf)  confirm the effectiveness of our algorithm in saving communication costs and preserving convergence performance.
>
> > Originality: The core algorithmic components ...
>
> Response: Regarding the intuition, we can use the full-precision method to illustrate the underlying reason. Specifically, in full-precision methods such as DM-HSGD, the update of variables is adjusted by the learning rate (e.g., Steps 14 and 15 in Algorithm 1 of DM-HSGD), whereas the update of the gradient tracker does not involve the learning rate (e.g., Steps 11 and 12 in Algorithm 1 of DM-HSGD). Following this mechanism, the update in Eq. (5) operates on the variable $x$; therefore, we should incorporate the learning rate into the balance coefficient, because the compression residual is a key component in updating $x$. Meanwhile, the update in Eq. (9) operates on the gradient tracker; therefore, we do not incorporate the learning rate into the balance coefficient, in line with the full-precision algorithm.
>
> Regarding the technical side, this design is critical to guarantee convergence. In particular, if we do not incorporate $\eta$ into the balance coefficient in Eq. (5), then we must replace $\alpha_x$ with $\alpha_x/\eta$ in all proofs. This can lead to infeasible hyperparameters. For example, in the second inequality of Eq. (172), after replacing $\alpha_x$ with $\alpha_x/\eta$, we obtain the condition $\delta(1-\lambda)\geq \text{const}$. Here, the compression hyperparameter $\delta$ and the spectral gap $1-\lambda$ are not tunable parameters. They are fixed for the given compression operator and communication topology.  Therefore, we cannot guarantee the existence of a setting that satisfies this condition, and consequently, convergence cannot be ensured.
>
> As for the potential function, it is different from Gao, 2022, because the additional compression error incurs the circle dependence (the details can be found in Appendix C. 2), while Gao, 2022 does not have this issue. To address this unique challenge, we developed a new strategy do determine the coefficient of the potential function, which is discussed in Appendix C.3.
>
> In summary, our design of the balance coefficient is novel and well-justified, with sufficient support from both intuitive and technical perspectives.

---

> > ### Author Rebuttal · Reviewer_WV4Z · 2026-04-01
> >
> > Thanks for your rebuttal. My concerns are fully addressed and I decide to increase the score accordingly.

---

> > > ### Author Response · Authors · 2026-04-07
> > >
> > > Thank you very much for increasing the score and for recognizing the contribution of our paper.
> > >
> > > Your questions are very helpful for improving our paper and making the insights and contributions much clearer.  We will revise the paper following your suggestions. We sincerely hope the reviewer will continue to support our work during the discussion phase. Thank you very much again!
> > >
> > >
> > > ## **New Evidence Supporting Our Contributions**
> > >
> > > Even for minimization problems, it is challenging to balance the full-precision update and the compressed update. For example, DoCoM (https://arxiv.org/pdf/2202.00255) studies decentralized **minimization** with the compressed communication, where it also tries to balance the full-precision and the compressed update in its Eq. (9a). However, **its proof is problematic**.  In particular, as shown in its Eq.(24), the upper bound of the balance coefficient $\gamma$ relies on $\sqrt{1-\delta/(8\bar{\omega}^2)}$, where $\delta$ and $\bar{\omega}$ are not tunable parameters as shown in its Assumption 2.2 and Assumption 2.3. However, $1-\delta/(8\bar{\omega}^2)>0$ **does NOT always hold for all settings**. Therefore, it is unclear whether DoCoM really converges for the minimization problem.
> > >
> > > In our paper, through novel designs of the balance coefficients, i.e., incorporating the learning rate into the balance coefficient as shown in Eq. (5), our algorithm is able to guarantee convergence. We believe that this design can also be applied to minimization problems to address the issue of DoCoM.
> > >
> > > In summary, our novel design and rigorous theoretical analysis make important contributions to communication-efficient decentralized optimization, which can not only advance decentralized minimax optimization but also address gaps in traditional minimization problems.
> > >
> > > We hope this helps the reviewer better assess the contributions of our paper. **Considering the significant contributions of our paper to communication-efficient decentralized optimization, we sincerely hope the reviewer might kindly consider increasing the score**.

---

### Official Review · Reviewer_hek4 · 2026-03-11

**Soundness:** 2
**Presentation:** 2
**Significance:** 2
**Originality:** 2
**Overall Recommendation:** 3
**Confidence:** 4

**Summary:**

This paper targets on addressing the heavy computation overhead in distributed minimax optimization algorithms and proposes a stochastic gradient descent ascent with momentum algorithm based on the error feedback method.

**Compliance With Llm Reviewing Policy:**

Affirmed.

**Final Justification:**

The problem of communication-efficient decentralized minimax optimization is interesting and practically meaningful. However, I am still not convinced about the paper’s novelty. The claimed core contribution seems to fall within the broader line of work on adaptive stepsize coordination, which has already been explored in recent methods. For this reason, I will keep my current score.

**Key Questions For Authors:**

Weakness:

- Communication-efficient decentalized algorithms are very important to the distributed learning communittee. However, this peper’s proposed algorithm is essentially a relatively straightforward combination of (a) decentralized SGDA with momentum/gradient tracking, and (b) the error feedback mechanism from decentralized minimization. The main claimed novelty is making the balance coefficient for variables depend on the learning rate, but this is a relatively minor design choice rather than a conceptually new idea. Moreover, there are many existing methods aimming to do adaptive training in a distribution learning manner, such as adaptive federated learning.
- The convergence analysis, while techically involved, follows a well-established templete. The proof uses a standard Lyapunov potential function approach that is common in decentralized optimization literature. Furthermore, the "circle dependence" challenge, while real, is resolved by a somewhat mechanical procedure (representing some coefficients in
terms of others, then solving sequentially). No new theoretical insights are provided.
- As authods stated in Remark 4.6, CE-DSGDAR has a high-order dependence on $1-\lambda$. No discussion on its practical impact.
- In the experimental results, why the AUC suddenly increases for all methods at the later training phase?

Minor: Assumption 1 seems not finished.

**Limitations:**

See questions.

**Strengths And Weaknesses:**

I note two key strengths:

- This paper is reasonably polished and generally well-written.
- The proposed methods are evaluated on multiple datasets and compared with different important baselines.

---

> ### Author Rebuttal · Authors · 2026-03-28
>
> We sincerely thank the reviewer for the valuable comments and constructive suggestions. Below, we provide our detailed response.
>
> > Communication-efficient decentalized algorithms are very important to the distributed learning communittee.  ...
>
> Response: We respectfully disagree with that this is a relatively minor design choice. Instead, **our algorithm design is critical for guaranteeing convergence**. In minimax optimization, **the update of the primal and dual variables should be carefully coordinated to guarantee convergence**. When incorporating communication compression, coordinating them becomes more challenging, because their updates depend on both the full-precision update and the compression residual as shown in Eq.(5). To address this challenge, we propose a novel approach to set the balance coefficient in Eq. (5), i.e., the balance coefficient should rely on the learning rate $\eta$. **This setting has never been discovered in traditional minimization problems**. For example, this paper (https://arxiv.org/pdf/2202.00255) for minimization problems does not show the balance coefficient $\gamma$ should rely on the learning rate.
>
> **Most importantly, without the proposed balance coefficient design, convergence cannot be guaranteed**.  In particular, if we do not incorporate $\eta$ into the balance coefficient in Eq. (5), then we must replace $\alpha_x$ with $\alpha_x/\eta$ in all proofs. This can lead to infeasible hyperparameters. For example, in the second inequality of Eq. (172), after replacing $\alpha_x$ with $\alpha_x/\eta$, we obtain the condition $\delta(1-\lambda)\geq \text{const}$. Here, the compression hyperparameter $\delta$ and the spectral gap $1-\lambda$ are not tunable parameters. They are fixed for the given compression operator and communication topology.  Therefore, we cannot guarantee the existence of a setting that satisfies this condition, and consequently, convergence cannot be ensured.
>
>
> In summary, **our algorithm is novel compared to the traditional minimization problem, as our algorithm addresses the unique challenge caused by the minimax structure**.
>
> > The convergence analysis, while techically involved, follows a well-established templete.  ...
>
> Response: We respectfully disagree with this argument. In minimax optimization, the interaction between the primal and dual variables results in the complicated interdependence among gradient estimation error, consensus error, and compression errors. In particular, as shown **in our Figure 6, there exists a circle dependence among the coefficients of our potential function. On the contrary, the traditional minimization problem and full-precision decentralized minimax problem do not have this unique challenge**. To address this unique challenge, we proposed a novel approach to decouple the circle dependence, which has never been studied in existing works. The details can be found in our Appendix C.2 and C.3. Therefore, our convergence analysis is also novel.
>
> In addition, we believe our strategy for addressing the circle dependence is general. **It can be applied to other optimization problems which has multiple variables**, such as the decentralized bilevel optimization with compressed communication.
>
> In summary, our convergence analysis is **novel**, which addresses the unique challenges in minimax problems, and **general**, which can be applied to other problems, such as bilevel problems.
>
> > As authods stated ...
>
> Response: In Figure 2, we have conducted experiments when using different communication topology, i.e., different spectral gap. It can be seen that our algorithm can achieve almost the same convergence performance. That said, the influence of the spectral gap is minimal in practice.
>
> > In the experimental results, why the AUC suddenly increases ...
>
> Response: In our experiments, we use resnet. A common strategy for training resnet is to reduce the learning rate in later iterations. Please refer to Figure 4 of the original resnet paper (https://arxiv.org/pdf/1512.03385). In our experiments, we follow this strategy where we divide the learning rate by 10 in the 50-th epoch.

---

> > ### Author Rebuttal · Reviewer_hek4 · 2026-04-03
> >
> > I thank the authors for their rebuttal. The topic of communication-efficient decentralized minimax optimization is interesting and practically relevant. However, I still have concerns about the novelty. The core contribution — making the balance coefficient depend on the learning rate — is essentially an adaptive stepsize coordination problem, which has been studied more generally in recent work such as D-AdaST (Yan Huang et al., NeurIPS 2024) and AdaFGDA (Feihu Huang et al., AISTATS 2024). These works address the same fundamental challenge of coordinating adaptive stepsizes across distributed nodes in minimax settings, with stronger theoretical results.
> >
> > Given this, the theoretical insight here feels incremental rather than a standalone contribution. I'm keeping my score.

---

> > > ### Author Response · Authors · 2026-04-03
> > >
> > > We respectfully disagree with the reviewer's argument. The reasons are as follows:
> > >
> > > * Those adaptive algorithms focus on the **full-precision** update, while our algorithm focuses on the **compressed** update. **For these full-precision methods, they study how to incorporate the adaptive step size to the full-precision gradients. It is unclear how they can handle the compressed update**. In contrast, our algorithm studies how to handle the compressed update, and we developed a novel approach to design the coefficient for the compressed update. Therefore, **it is meaningless to compare our contribution to those adaptive full-precision algorithms**.
> > >
> > > * The reviewer argues that those adaptive algorithms can achieve stronger theoretical results. This is NOT true.
> > >
> > >     * (**Strong Assumptions**) In particular, D-AdaST assumes bounded stochastic gradients in its Assumption 3, which is a very strong assumption. AdaFGDA assumes the adaptive step sizes are lower and upper bounded in its Assumption 7. All these assumptions are very strong, and our paper has never used these strong assumptions.
> > >
> > >     * (**Worse Convergence Rates**)They failed to achieve stronger theoretical results. D-AdaST can only achieve $\tilde{O}(1/\epsilon^{4+\delta})$ where $\delta>0$, which is much worse than our $O(1/K\epsilon^{3})$ convergence rate in terms of both $K$ and $\epsilon$.  AdaFGDA can only achieve $\tilde{O}(1/\epsilon^{3})$, which is also slower than our $O(1/K\epsilon^{3})$ in terms of $K$.  More importantly,  $\tilde{O}(\cdot)$ in their convergence rates hides the log term, which is worse than $O(\cdot)$. Therefore, their convergence rates is even much worse than ours.
> > >
> > > In summary, the reviewer's argument that those existing methods can achieve stronger theoretical results is **NOT true**.
> > >
> > >
> > > ## **Response to Final Justification**.
> > >
> > > We respectfully disagree with the reviewer's argument: core contribution seems to fall within the broader line of work on adaptive stepsize coordination.
> > >
> > > It seems there may be a **misunderstanding of our work**. Our algorithm is orthogonal to adaptive step-size methods.
> > > * In particular, the adaptive step-size method compute adaptive step sizes **based on the estimation of hessian matrix like Adam**, e.g.,  D-AdaST and AdaFGDA.  For example, AdaFGDA computes the adaptive step size via $1/\sqrt{a\_t}$, where $a\_t = (1-\rho)a\_{t-1} + \rho w\_t^2$.
> > > * In contrast, we use a constant step size.  Specifically, **the step sizes $\gamma_x \eta$ for the full-precision update and $\alpha_x \eta$ for the compressed update are both constant values**, as specified in Theorem 4.4.
> > >
> > > Therefore, our work is fundamentally different from adaptive step-size methods. Our focus  is to study how to balance full-precision and compressed updates through novel learning rate designs. We have explained the novelty  in our initial response, and to the best of our knowledge, this design has not been explored in existing work. Therefore, our work fills the gap between communication-efficient algorithms and minimax optimization. This can benefit a wide range of applications, such as federated learning. In summary, our work is novel and makes important contributions to minimax optimization.
> > >
> > > ## **New Evidence Supporting Our Contributions**
> > >
> > > Even for minimization problems, it is challenging to balance the full-precision update and the compressed update. For example, DoCoM (https://arxiv.org/pdf/2202.00255) studies decentralized **minimization** with the compressed communication, where it also tries to balance the full-precision and the compressed update in its Eq. (9a). However, **its proof is problematic**.  In particular, as shown in its Eq.(24), the upper bound of the balance coefficient $\gamma$ relies on $\sqrt{1-\delta/(8\bar{\omega}^2)}$, where $\delta$ and $\bar{\omega}$ are not tunable parameters as shown in its Assumption 2.2 and Assumption 2.3. However, $1-\delta/(8\bar{\omega}^2)>0$ **does NOT always hold for all settings**. Therefore, it is unclear whether DoCoM really converges for the minimization problem.
> > >
> > > In our paper, through novel designs of the balance coefficients, i.e., incorporating the learning rate into the balance coefficient as shown in Eq. (5), our algorithm is able to guarantee convergence. We believe that this design can also be applied to minimization problems to address the issue of DoCoM.
> > >
> > > In summary, our novel design and rigorous theoretical analysis make important contributions to communication-efficient decentralized optimization, which can not only advance decentralized minimax optimization but also address gaps in traditional minimization problems.
> > >
> > > We hope this helps the reviewer better assess the contributions of our paper. **Considering the significant contributions of our paper to communication-efficient decentralized optimization, we sincerely hope the reviewer might kindly consider increasing the score**.

---

### Official Review · Reviewer_iZzh · 2026-03-13

**Soundness:** 3
**Presentation:** 3
**Significance:** 2
**Originality:** 2
**Overall Recommendation:** 3
**Confidence:** 2

**Summary:**

This paper proposes CE-DSGDAR, a communication-efficient decentralized stochastic gradient descent ascent algorithm with momentum for solving nonconvex strongly concave minimax optimization problems. A primary contribution is a convergence analysis in the case of primary and dual variables, which is usually difficult. The algorithm applies error feedback to both communicated variables and gradient trackers to mitigate compression error. The key design contributions are in balancing the full precision updates with the compression residual.

**Compliance With Llm Reviewing Policy:**

Affirmed.

**Key Questions For Authors:**

How tight is the dependence on the contraction parameter and spectral gap?

Why do huge constants appear in the proofs in the Appendix (page 63, for example)? How should this be interpreted?

**Limitations:**

Yes

**Strengths And Weaknesses:**

The assumptions are standard assumptions in this area and clearly stated.  Integrating error feedback into decentralized minimax optimization seems to be more complex than doing so for minimization. The convergence rate explicitly shows how contraction and spectral gap affect complexity.

While the paper has a clear logical structure, it can also be difficult to read (e.g. algorithm 1). The experimental results should include more comparisons to existing work, since there are countless works that are either directly applicable to this setting or can easily be modified to this setting. The intersection of decentralized optimization and and communication compression has seen extensive recent work and the contribution here is narrow.

---

> ### Author Rebuttal · Authors · 2026-03-28
>
> We sincerely thank the reviewer for the valuable comments and constructive suggestions. Below, we provide our detailed response.
>
> > While the paper has a clear ... to read (e.g. algorithm 1).
>
> Response: In our Algorithm 1, we provide an explanation for every step. However, we are happy to include a more detailed explanation of our algorithm in the final version. For example, we will add a figure illustrating the pipeline of our algorithm to make it easier to understand.
>
> > The experimental results ... modified to this setting.
>
> Response: As the reviewer suggested, we have added an additional baseline DGDA-VR (https://arxiv.org/pdf/2307.09421), and then verified the performance of our algorithm on a new application as suggested by Reviewer WV4Z. The experimental results can be found in our second response to Reviewer WV4Z.
>
>
> > The intersection of ... narrow.
>
> Response: We respectfully disagree with this argument. Although decentralized communication efficient algorithms have been studied before, **existing works mainly focus on the standard minimization problem**. To the best of our knowledge, **they have not been studied for decentralized minimax problems**.  Applying the communication compression technique to decentralized minimax problems is NOT trivial, because the minimax structure brings unique challenges for algorithm design and convergence analysis.
> * _Algorithm design_.  In minimax optimization, **the update of the primal and dual variables should be carefully coordinated to guarantee convergence**. When incorporating communication compression, coordinating them becomes more challenging, because their updates depend on both the full-precision update and the compression residual as shown in Eq.(5). To address this challenge, we propose a novel approach to set the balance coefficient in Eq. (5), i.e., the balance coefficient should rely on the learning rate $\eta$. **This setting has never been discovered in traditional minimization problems**. For example, this paper (https://arxiv.org/pdf/2202.00255) for minimization problems does not show the balance coefficient $\gamma$ should rely on the learning rate. Therefore, **our algorithm is novel compared to the traditional minimization problem, as our algorithm addresses the unique challenge caused by the minimax structure**.
>
> * _Convergence analysis_.  (a) In minimax optimization, the interaction between the primal and dual variables results in the complicated interdependence among gradient estimation error, consensus error, and compression errors. In particular, as shown in our Figure 6, there exists a **circle dependence** among the coefficients of the potential function. On the contrary, the traditional minimization problem does not have this unique challenge. To address this unique challenge, we proposed a novel approach to decouple the circle dependence, which has never been studied in traditional minimization problems. The details can be found in our Appendix C.2 and C.3.
>
>     (b) Moreover, **the design regarding the balance coefficient is critical to guarantee convergence**. In particular, if we do not incorporate $\eta$ into the balance coefficient in Eq. (5), then we must replace $\alpha_x$ with $\alpha_x/\eta$ in all proofs. This can lead to infeasible hyperparameters. For example, in the second inequality of Eq. (172), after replacing $\alpha_x$ with $\alpha_x/\eta$, we obtain the condition $\delta(1-\lambda)\geq \text{const}$. Here, the compression hyperparameter $\delta$ and the spectral gap $1-\lambda$ are not tunable parameters. They are fixed for the given compression operator and communication topology.  Therefore, we cannot guarantee the existence of a setting that satisfies this condition, and consequently, convergence cannot be ensured.
>
>     In summary, our convergence analysis is also novel.
>
> > How tight is ... spectral gap?
>
> Response: In Remark 4.6, we point out that the dependence on the spectral gap is worse than in the full-precision counterpart and explain the detailed reasons, namely, that it is caused by the coefficient $\alpha$ associated with the compression operation. If there is no compression, i.e., $\alpha=0$,  there will be no the constraint in Eq. (108). Consequently, $\gamma_x$ will have a better dependence on the spectral gap and therefore can recover the dependence on the spectral gap as the full-precision method.  When there exists compression, improving this dependence remains an open problem, which we leave for future work.
>
> > Why do ... be interpreted?
>
> Response: They are just constant values, which are independent of the problem-specific hyperparameters like $L$. They do not affect the convergence rate. These constants can be easily improved by using some smaller constant value when relaxing the upper bound.

---

> > ### Author Rebuttal · Reviewer_iZzh · 2026-04-07
> >
> > Thank you for the clarifications. My comments concerning the novelty of the work stands, and I don't think the paper is ready for publication.

---

> > > ### Author Response · Authors · 2026-04-07
> > >
> > > Thank you for acknowledging that our clarification is helpful. We respectfully disagree with the argument regarding our novelty and contributions.
> > >
> > > ## **New Evidence Supporting Our Novelty and Contributions**
> > >
> > > In fact, even for the minimization problem, it is challenging to balance the full-precision update and the compressed update. For example, DoCoM (https://arxiv.org/pdf/2202.00255) studies decentralized **minimization** with the compressed communication, where it also tries to balance the full-precision and the compressed update in its Eq. (9a). However, **its proof is problematic**.  In particular, as shown in its Eq.(24), the upper bound of the balance coefficient $\gamma$ relies on $\sqrt{1-\delta/(8\bar{\omega}^2)}$, where $\delta$ and $\bar{\omega}$ are not tunable parameters as shown in its Assumption 2.2 and Assumption 2.3. However, $1-\delta/(8\bar{\omega}^2)>0$ **does NOT always hold for all settings**. Therefore, it is unclear whether DoCoM really converges for the minimization problem.
> > >
> > > In our paper, through novel designs of the balance coefficients, i.e., incorporating the learning rate into the balance coefficient as shown in Eq. (5), our algorithm is able to guarantee convergence. We believe that this design can also be applied to minimization problems to address the issue of DoCoM.
> > >
> > > In summary, our novel design and rigorous theoretical analysis make important contributions to communication-efficient decentralized optimization, which can not only advance decentralized minimax optimization but also address gaps in traditional minimization problems.
> > >
> > > We hope this helps the reviewer better assess the contributions of our paper. We would be glad to address any further questions. **Considering the significant contributions of our paper to communication-efficient decentralized optimization, we sincerely hope the reviewer might kindly consider increasing the score.**

---

### Decision · Program_Chairs · 2026-04-30

**Decision:**

Accept (regular)

**Comment:**

This paper tackles a crucial but underexplored problem: communication-efficient decentralized minimax optimization. The proposed CE-DSGDAR algorithm introduces a learning-rate-dependent balance coefficient for error feedback—a design that is not heuristic but theoretically necessary. Without it, convergence conditions become infeasible due to fixed compression and topology parameters. This insight alone advances both minimax and minimization literature (e.g., fixing a flaw in prior minimization work like DoCoM).

The analysis further resolves a novel “circle dependence” among consensus, compression, and gradient errors—a challenge absent in standard minimization. While two reviewers questioned novelty, their concerns were largely about incremental components, overlooking the nontrivial integration and the essential coefficient design. The rebuttal added experiments (robust optimization) and clarified distinctions from adaptive-stepsize methods.

Given the technical rigor, practical importance, and successful resolution of key concerns, the paper meets the ICML bar. The Area Chair recommends acceptance.